# Calving event size measurements and statistics of Eqip Sermia, Greenland, from terrestrial radar interferometry

Andrea Walter[1,2], Martin P. Lüthi[1], Andreas Vieli[1]

5    [1]Institute of Geography, University of Zurich, Zurich, Switzerland
[2]Laboratory of Hydraulics, Hydrology and Glaciology, ETH Zurich, Zurich, Switzerland

*Correspondence to*: Andrea Walter (andrea.walter@geo.uzh.ch)

**Abstract.** Calving is a crucial process for the recently observed dynamic mass loss changes of the Greenland ice sheet. Despite its importance for global sea level change, major limitations in understanding the process of calving remain. This study presents high resolution calving event data and statistics recorded with a terrestrial radar interferometer at the front of Eqip Sermia, a marine terminating outlet glacier in Greenland. The derived digital elevation models with a spatial resolution of several meters recorded at one-minute intervals were processed to provide source areas and volumes of 906 individual calving events during a 6-day period. The calving front can be divided into sectors ending in shallow and deep water with different calving statistics and styles. For the shallow sector, characterised by an inclined and very high front, calving events are more frequent and larger than for the vertical ice cliff of the deep sector. We suggest that the calving volume deficiency of 90% relative to the estimated ice flux in our observations of the deep sector is removed by oceanic melt, subaquatic calving and small aerial calving events. Assuming a similar ice thickness for both sectors implies that subaqueous mass loss must be substantial for this sector with a contribution of up to 65% to the frontal mass loss. The size distribution of the shallow sector is represented by a log-normal model, while for the deep sector the log-normal and power law model fit well, but none of them is significantly better. Variations in calving activity and style between the sectors seem to be controlled by the bed topography and the front geometry. Within the short observation period no simple relationship between environmental forcings and calving frequency or event volume could be detected.

## 1 Introduction

Over the past decade rapid retreat, thinning and flow acceleration of many outlet glaciers contributed substantially to the observed increasing mass loss of the Greenland ice sheet (Moon et al., 2012; Enderlin et al., 2014; King et al., 2018) and consequently to global sea level rise (Rignot et al., 2011; IPCC, 2014). These dynamic changes seem to be related to a general warming of air temperature and water masses around Greenland (Straneo et al., 2013). Several studies have shown a high sensitivity of outlet glaciers to environmental forcings (Holland et al., 2008; Howat et al., 2010; Carr et al., 2017), while the fjord topography is an important control for the dynamic behaviour of the outlet glaciers (Warren, 1991; Catania et al., 2018). However, major limitations in understanding and predicting the dynamics of outlet glaciers remain, e.g. a complex link between atmospheric forcing and calving activity and insufficient resolution in models and observations. The detailed relationship between climate and dynamic changes is still poorly understood (McFadden et al., 2011; Vieli and Nick, 2011; Straneo et al., 2013).

Calving is a crucial process for the dynamic behaviour of tidewater glaciers, but the detailed mechanisms and relation to environmental forcing are not well understood (Joughin et al. 2004; Thomas, 2004; Nick et al., 2009). Calving rates are generally a function of the stress state at the terminus. When stresses exceed the strength of the ice, fractures form and propagate, until blocks of ice separate from the front. Mechanisms causing fractures to propagate are: 1) spatial gradients in the glacier velocity, 2) changes in frontal geometry (front position, height), 3) undercutting of the glacier front by melting at or below the water line and 4) buoyancy forces (Pralong and Funk, 2005; Benn et al., 2007). Direct and continuous observations

of the calving process are difficult and therefore the underlying mechanisms are observationally under-constrained. Most existing studies investigated the calving process on longer time scales by considering time averaged calving rates or fluxes. Available studies on individual calving events focus mostly on discontinuous (Warren et al., 1995; O'Neel et al., 2003) or indirect measurements (O'Neel et al., 2010; Walter et al., 2010; Bartholomaus et al., 2012; Glowacki et al., 2015). Several studies investigating the process of ice break-off over short time scales show that the process of calving has a very high temporal and spatial variability and that the observed calving size distribution for grounded tidewater glaciers is following a power law (Chapuis and Tetzlaff, 2014; Åström et al., 2014; Pętlicki and Kinnard, 2016). However, these investigations focus mostly on time averaged estimates of volumes, discontinuous datasets, indirect measurements or a combination thereof and thus lack continuous direct observations of the calving event size. For an accurate representation of the calving process in current flow models and to link calving activity with potential environmental forcings more detailed observations with high temporal and spatial resolution are necessary.

During the last 20 years observational data for monitoring calving glaciers were mainly obtained through satellites at a sampling frequency that is not suitable to observe individual calving events. Other more in-situ based approaches such as terrestrial photogrammetry using time-lapse cameras (dependent on weather and daylight) (Vallot et al., 2019) and drone data (limited temporal resolution) (Jouvet et al., 2017) also show severe limitations regarding the observation of the highly variable calving process. Promising results were obtained with seismic monitoring of calving (Amundson et al., 2012; Walter et al., 2013; Bartholomaus et al., 2015, Köhler et al., 2016, Köhler et al., 2019) and maximum wave amplitudes as a proxy for calving fluxes (Minowa et al., 2018), but those methods cannot quantify calving event volumes directly. Terrestrial laser scanning allows to measure the volume of individual calving events (Pętlicki and Kinnard, 2016), but requires suitable meteorological conditions and lacks the temporal resolution to detect individual calving events. Terrestrial radar interferometers can overcome most of the mentioned limitations and have been used to study the effects of tidal forcing on the front of an outlet glacier (Voytenko et al., 2015), to investigate calving rate and velocity (Rolstad and Norland, 2009), to determine calving event frequency (Chapuis et al., 2010), velocity variations and grounding line motion (Xie et al., 2018), pro-glacial mélange thickness (Xie et al., 2019), glacier's response to calving (Cassotto et al., 2018) or to estimate the volume of a single large calving event (Lüthi and Vieli, 2016).

This study aims at investigating the calving process and event statistics by using a terrestrial radar interferometer (TRI). For this purpose, the calving front of the tidewater outlet glacier Eqip Sermia in Greenland was investigated with a TRI at one-minute intervals during a 6-day field campaign in 2016. The resulting high resolution time-series of individual calving event volumes and related source areas allow us to investigate the relationship between calving front geometry, calving flux and environmental forcings such as tides or air temperature.

## 2 Study area and data acquisition methods

### 2.1 Study area

Eqip Sermia (69.81° N, 50.20° W) is an ocean terminating outlet glacier located at the western margin of the Greenland ice sheet. Observations of the glacier front position, surface elevation and flow speed are available at decadal resolution since 1912 and show a doubling of discharge and accelerated retreat within the last two decades (Lüthi et al., 2016). Between 1912 and 2006 velocities between 2.5 and 5 m d$^{-1}$ were observed, whereas today the glacier front velocities measured over the observation period in 2016 are reaching up to 16 m d$^{-1}$. After a rapid retreat starting in 2010, the calving front position stabilized during the last five years.

The calving front has a width of 3.2 km and a height above the water line between 50 and 170 m. The entire front is grounded but the water depth in the northern half is very shallow (0 – 20 m, termed 'shallow sector' from now on) and locally the bedrock protrudes above the water. In the southern sector the water depth is 70 to 100 m (subsequently termed 'deep sector'). Directly at the calving front no depth sounding data are available and the given depth estimates are extrapolated from bathymetric surveys in the proximity of the current front position (Rignot et al., 2015; Lüthi et al., 2016). The difference in bed topography between the deep sector and the shallow sector is also visible in the bathymetry from BedMachine v3 (Fig. S8; Morlighem et al., 2017). Related to the contrast in water depth, the geometry of the front is distinctly different between the two frontal sectors. In the deep southern sector the front is vertical and the frontal cliff height lower than in the shallow northern sector where the front is inclined (Figs. 1 and 2).

### 2.2 Terrestrial radar interferometer

A terrestrial radar interferometer (TRI, Gamma GPRI) was installed on bedrock 150 m above sea level across the bay of Eqip Sermia at 4.5 km distance (69.7523° N / 50.2520° W; Figs. 1 and 2) with the line-of-sight in flow direction of the glacier. The measurements were repeated at one-minute intervals from 19 August 2016, 18:40 UTC to 27 August 2016, 10:30 UTC. This allowed to produce an almost continuous record of velocity and elevation change over 7.65 days with a 1.53 days break (22 August 2016, 00:55 UTC to 23 August 2016, 13:00 UTC) due to an instrument failure.

The Gamma GPRI is a real-aperture radar interferometer featuring one transmitting and two receiving antennas. Acquisitions are obtained by antenna rotation along the vertical on a precision astronomical mount. Consecutive interferograms from one of the receiving antennas are used to calculate the velocity. The two receiving antennas facilitate reconstruction of the topography. The radar interferometer operates at a wavelength of $\lambda = 17.4$ mm (Ku-Band, 17.2 GHz). The range resolution is approximately 0.75 m, while the azimuth resolution is 0.1 degrees corresponding to 7 m at a slant range of 4.5 km (Werner et al., 2008a).

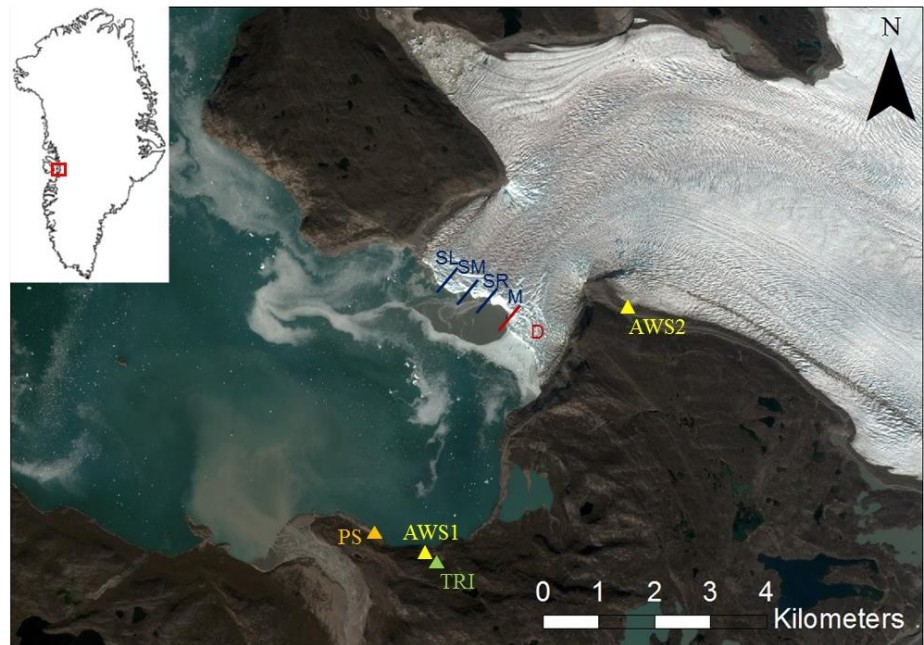

**Figure 1: Overview of Eqip Sermia and measurement sites. The positions of the terrestrial radar interferometer (TRI), the pressure sensor (PS) and the two weather stations (AWS) are indicated by triangles. The deep and shallow calving front sectors are marked with red and blue lines. Background: Sentinel-2A scene from 3 August 2016 (from ESA Copernicus Science Hub: https://scihub.copernicus.eu).**

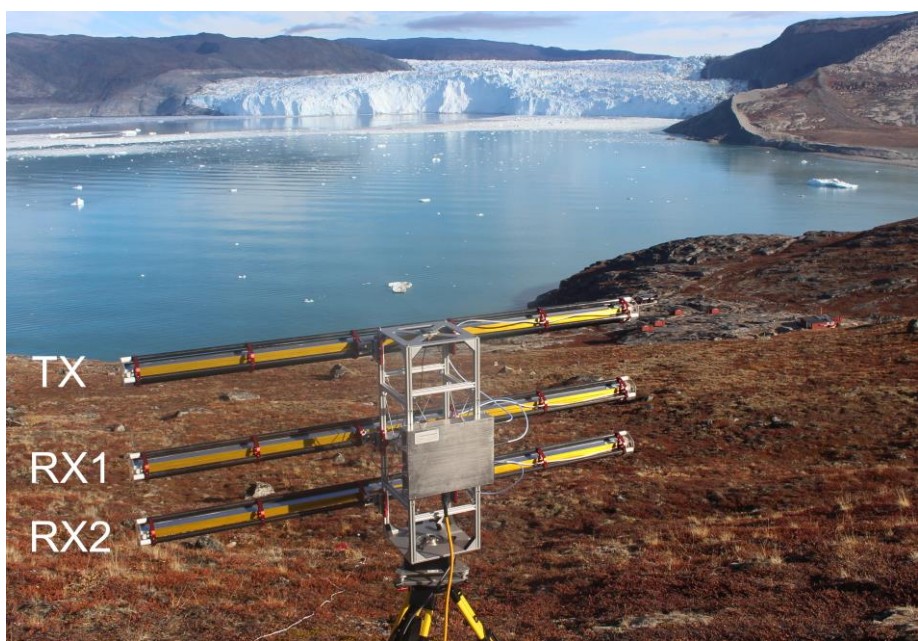

**Figure 2: The terrestrial radar interferometer (TRI) located opposite of the front of Eqip Sermia at a distance of 4.5 km (image: M. Lüthi, 2016). The TRI has one transmitting (TX) and two receiving antennas (RX1, RX2).**

## 2.3 Environmental data

Two automatic weather stations (AWS) with Decagon Em50 data loggers were installed at the sites indicated in Figure 1 and collected data in one hour intervals during the entire field campaign. AWS2 located next to the ice edge at 362 m a.s.l. (69.79442° N / 50.16115° W) measured air temperature and relative humidity (VP-3 humidity temperature and vapor pressure sensor) and wind (DS-2 sonic anemometer). AWS1 near the TRI at 60 m a.s.l. (69.75556° N / 50.25301° W) measured additionally incoming shortwave radiation (PYR solar radiation sensor) and precipitation (ECRN-100 high-resolution rain gauge). The meteorological conditions at the ice edge (AWS2) are influenced by the ice sheet while at AWS1 next to the TRI it is more representative for the weather conditions at the shore of the fjord.

Tides and waves induced by calving were recorded in the fjord with a RBRsolo pressure sensor (PS; Fig. 1) at a sampling rate of two seconds. The pressure sensor was installed at the shore at a distance of 4.5 km from the ice front (69.75731° N / 50.26490° W, Fig.1). To protect the sensor from floating ice and moving rocks it was fixed in a metal pipe that was attached to a rock at the shore by a steel cable.

## 3 Data processing methods

### 3.1 TRI data processing

The GPRI transmits the radar signal from antenna TX and records it by the two receiver antennas RX1 and RX2, which enables spatial interferometry (Fig. 2). To reconstruct topography, interferograms were produced using a standard workflow following Caduff et al. (2015) using the Gamma software stack. The resulting interferograms were unwrapped, using stable features on bedrock as reference. Following Strozzi et al. (2012), the unwrapped phases were then converted to topography z:

$$z = \frac{\lambda}{2\pi} \frac{R}{B} \phi + \frac{B}{2} - \left(\frac{\lambda}{2\pi}\right)^2 \frac{\phi^2}{2B},$$

where $\lambda$ = 17.4 mm is the wavelength, R the range to a point on the ground, B = 0.25 m the baseline between the two receiving antennas, and $\phi$ the measured interferometric phase. To correct for systematic error sources, which can be caused by errors in the reference heights and instrumental geometry, baseline errors and errors caused by a not perfectly vertical mounting of the three antennas (Strozzi et al., 2012), a correction factor was calculated. This was done by comparing the calculated digital elevation models (DEMs) with the Arctic DEM (Porter et al., 2018) and choosing control points on stable terrain at different distances from the radar. The resulting correction factor was multiplied with the calculated topography to minimize absolute uncertainty in the height estimates. To reduce noise from atmospheric disturbances 10 consecutive elevation models were stacked. This noise is mainly due to phase shifts in the interferogram induced through changes in air pressure, temperature and humidity (Goldstein, 1995). The final elevation models have a resolution of 3.75 m in range and about 8 m in azimuth direction at the glacier front and were obtained at 10 min intervals over the whole campaign.

The accuracy of the so obtained DEMs was evaluated by comparing them on stable terrain with the Arctic DEM as a reference DEM. The variability between the calculated TRI elevation models on stable terrain was investigated by looking at the DEM differences over time and space.

In a next step consecutive stacked elevation models were subtracted. The negative height changes at the glacier front were
interpreted as calving events. Due to the stacking, calving events within 10 minutes are merged together. The aerial extent of individual calving events were extracted from the calculated height changes with the watershed segmentation method from the scikit-image package (van der Walt et al., 2014) with a height change of 15 m as starting points for the calving events and 5 m as threshold. This threshold corresponds to the maximum variability of the height between elevation models on stable terrain outside the glacier. Height changes of less than 5 m are considered as noise and filtered out. Additionally, calving events
smaller than 10 adjacent pixels and with a bounding box width smaller than 3 pixels (11.25 m) were excluded as noise. Thus, only calving events with both, $\geq$ 10 adjacent pixels and a bounding box width larger than 3 pixels, were extracted. Due to the asymmetric grid, events extended in range direction are more likely to be filtered out with the 10 pixel filter than wide ones. As noise has mostly an irregular shape, calving events smaller than 40 pixels also needed to fulfil the condition (number of pixels $\cdot$ 1.6) $\geq$ (number of pixels in bounding box). This condition is subsequently termed shape condition. When applying
these filtering thresholds, the signal-to-noise ratio is higher on stable terrain than without filtering. To exclude volume changes from collapsing seracs in the highly crevassed ice surface further upstream a mask around the glacier front was used. The mask is defined as a line along the front with a buffer of 20 pixels (approximately 75 m) on each side of the line (Fig. 4). All height changes outside the mask were ignored in the data processing.

For visualization the radar image pixels were mapped into cartesian coordinates. Since resampling is a possible source of error,
all calculations were performed in the radar geometry and only the final results were georeferenced. Nearest neighbour interpolation was used to resample the radar data to the cartesian UTM22N grid.

Next, we investigated whether the calving event sizes follow a size-frequency distribution. To test whether the measured calving volumes $V$ are explained by an exponential ($e^{-\beta V}$), a log-normal ($\frac{1}{V}\exp[-\frac{(\ln V - \mu)^2}{2\sigma^2}]$) or a power law ($V^{-\alpha}$) size-frequency distribution, a statistical analysis using the Python package powerlaw was applied (Alstott et al., 2014). The package
uses maximum-likelihood methods (Clauset et al., 2009) due to the non-linearity of the fitted curve and gives as result the log-likelihood ratio $R$, which is used to investigate which model fits the data better on a relative score, and the probability value $p$, which tells if one can trust the sign of $R$ (when p $\geq$ 0.1).

Ice flow velocities were calculated from consecutive interferograms of TRI acquisitions in one-minute intervals. To reduce noise, 120 interferograms (2 hours) were stacked before phase unwrapping with respect to a reflector on stable terrain. The
unwrapped phases can then be converted into line-of-sight displacement $\delta = \frac{-\lambda\phi}{4\pi}$ (Werner et al., 2008b), with a displacement measurement sensitivity smaller than 1 mm.

## 3.2 Pressure sensor data processing

The pressure sensor (PS; Fig.1) recorded the water pressure in the fjord opposite of the calving front, which can then be converted to water height and thus the amplitudes of the tides and calving waves are known. A high-pass filter with a pass frequency of 0.001 Hz was used to extract the calving waves which were then compared with the calving events detected by the TRI. The peaks of the calving waves were detected by using the peak detection algorithm detect_peaks (Duarte and Watanabe, 2018) with a minimum peak height of 0.01 m and a minimum peak distance of 300 samples. Similarly, the tides were extracted with a low-pass filter with a pass frequency of 0.001 Hz and are compared with the extracted calving events in order to identify a potential relationship between the tides and the calving events.

## 4 Results

### 4.1 DEM generation and calving event extraction

A DEM calculated with the TRI data and stacked over 60 minutes is presented in Figure 3. The glacier surface elevation above sea level is lower on the southern side (50 to 90 m), while at the northern side the elevation reaches up to 170 m.

To assess their uncertainty, the DEMs were compared to the Arctic DEM (Fig. S7). This comparison shows that on stable terrain, marked with a yellow box, the difference is around 5 m in flat areas, while it reaches about 15 m in steeper areas. The variability between the TRI-derived DEMs was investigated over time and space for the stable area marked in Figure 3. The mean height difference between the consecutive TRI-derived DEMs is between 1 and 2 m. The mean height difference as well as the standard deviation increases with distance and is higher in steeper areas (Fig. S2). The mean height difference of the stable terrain shows no clear trend over time (Figs. S3 and S4).

The calving events were extracted by using the height changes of the consecutive TRI-derived DEMs. In Figure 4 an example of unstacked height differences, of stacked height differences, and of the finally extracted calving event is given in radar geometry. It is clearly visible that the stacking improves the quality of the height-difference map. The same calving event is also traceable on the raw radar images as it generated waves (Fig. S5). The filtering methods used for the extraction of calving events reduce the number of calving events but also increase the signal-to-noise ratio. Comparing the amount of extracted events for a threshold of 1 and 5 m shows that with the threshold of 5 m 77% less events were extracted for both, the deep and the shallow sectors. The usage of the shape condition for events smaller than 40 pixels leads to 49% less events for the shallow and 54% less events for the deep sector.

To assess the distribution of the noise along the front, positive height changes were calculated using a minimum size of 10 pixels, a width of 3 pixels and the shape condition for all events (Fig. S6). The result shows that the shallow sector is likely more influenced by noise than the deep sector even after filtering. However, looking at unstacked and stacked height changes (Fig. 4) and the mean variabilities of the differentiated DEMs, the signal-to-noise ratio was increased considerably.

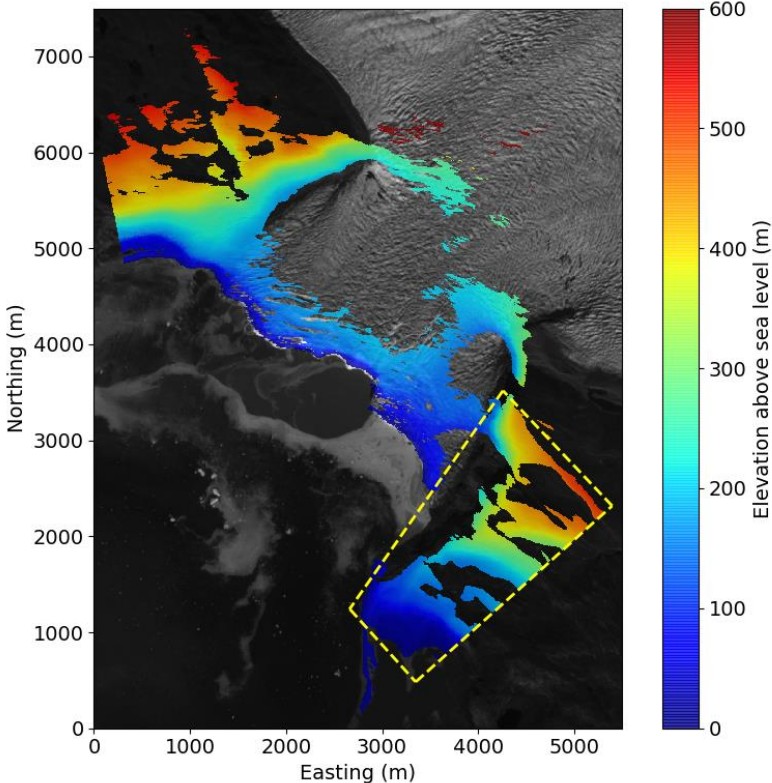

**Figure** 3**: TRI derived DEM stacked over 60 minutes. The yellow square marks the stable terrain area where the mean variability was investigated. The origin of the coordinate system corresponds to 527350 E / 7739550 N (UTM 22N). Background: Sentinel-2A scene from 3 August 2016 (from ESA Copernicus Science Hub: https://scihub.copernicus.eu).**

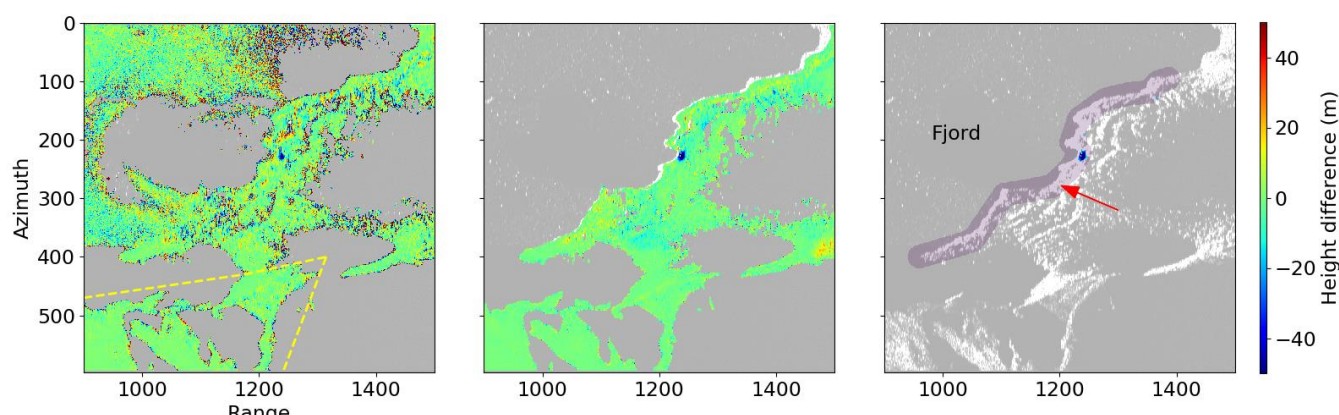

**Figure 4: Example of a calving event extraction on 20 August 16:40 UTC in radar geometry. The left image shows the elevation difference between two unstacked DEMs, while in the middle the difference is calculated between two stacked DEMs. The right image shows the final extracted calving event (blue colour). The red arrow indicates the general flow direction of the glacier, while the purple shaded area shows the front mask. The yellow marked area shows the stable terrain used for the uncertainty analysis.**

## 4.2 Flow velocities

Ice flow velocities from TRI measurements in vicinity of the calving front are presented in Figure 5a. Figure 5b shows the complete velocity field including the areas of radar line-of-sight shadow, which has been derived from repeated UAV surveys from August 2016 (Rohner et al. 2019). Speeds are increasing towards the calving front with highest values reaching 16 m d[-1]. Along the front the velocities are non-uniform, with two areas of high velocity separated by a frontal area where a bedrock ridge was visible during the field campaign (orange bar in Fig. 5; inset of Fig. 6). Further upstream the glacier velocity field is more uniform with generally higher velocities in the centre.

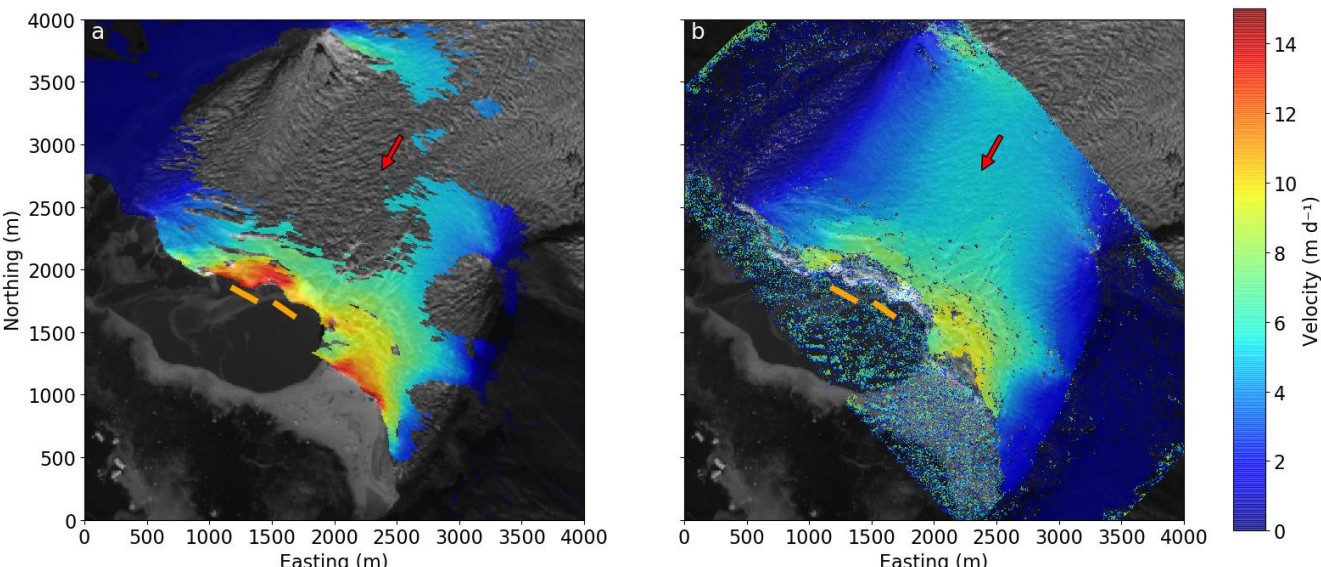

**Figure 5: The velocity field at the glacier front. (a) measured with the TRI (line of sight) on 19 August 2016 and (b) with a UAV (between 21 and 25 August 2016; Rohner et al., 2019). The red arrow indicates the general flow direction. The orange lines indicate an area where bedrock was observed at the foot of the front. The origin of the coordinate system corresponds to 528350 E / 7741550 N (UTM 22N). Background: Sentinel-2A scene from 3 August 2016 (from ESA Copernicus Science Hub: https://scihub.copernicus.eu).**

## 4.3 Magnitude and source area of calving events

During the field campaign 2016 a total of 906 calving events were identified within 6.12 days with a mean event volume of 17,686 m$^3$. Due to the distinctly different characteristics in cliff geometry and water depth the two front sectors were analysed separately. Within the shallow sector 725 events were found, whereas within the deep sector only 193 events were detected, which results in a mean calving activity of 4.9 and 1.3 events per hour, respectively. Note that 12 events were detected on the border of the two sectors and were thus counted for both sectors but only once for the total number of events. An overview of the number, volumes and event sizes is given in Table 1. The extracted calving event sizes are spread over four orders of magnitude and the total volume of all calving events detected in the deep sector is 5.8 times smaller than in the shallow sector.

Only small variations in the position of the calving front were observed with the TRI (Fig. S1) over the observation period, which implies that the ice loss by calving is compensated by the ice flow (Fig. 5).

Calving heights in each radar pixel (ca. 30 m$^2$ area) were added up over the measurement period and are referenced to as cumulative calving height. Figure 6 shows that within the shallow sector the cumulative calving height locally exceeds 350 m,
5  while it is considerably lower in the deep sector (D). Within the shallow sector variations in cumulative calving height are observable such that it can be divided into four sub-sectors named SL, SM, SR and M (Figs. 6 and 7). The highest cumulative calving heights are detected in sector SL, while sector M shows the lowest cumulative heights within the shallow sector. Sector SM has slightly lower values for the cumulative calving height than the sectors SR and SL. For sector D, the south-eastern part next to the mainland was not in sight of the radar as it is situated behind a moraine.

**Table 1: Detected calving events within each sector during the observation period of 6.12 days.**

|  | Whole front | Shallow sector (SL, SM, SR, M) | Deep sector (D) |
|---|---|---|---|
| **Total event number** | 906 | 725 | 193 |
| **Total event volume (m$^3$)** | 16,023,400 | 13,655,800 | 2,367,600 |
| **Event sizes** |  |  |  |
| Mean (m$^3$) | 17,700 | 18,800 | 12,300 |
| Median (m$^3$) | 11,600 | 12,900 | 8,500 |
| Minimum (m$^3$) | 660 | 660 | 2,115 |
| Maximum (m$^3$) | 275,700 | 275,700 | 108,900 |

Figure 7 shows the detailed record of calving activity along the different sectors of the calving front. Figure 7b presents how frontal height and velocity vary. The front height is fluctuating strongly along the front due to the highly crevassed surface.
15  The frontal cliff in the deeper sector D is mostly vertical and between 50 and 90 m high, while in the shallow sector the front is inclined at a slope of 50 degrees and reaching up to 170 m. In general, as shown in Figure 5, the velocities at the front increase from the margins towards the centre, with the exception of the area around the bedrock outcrop in sector M where velocities are slightly decreased.

Figure 7c summarizes the observed calving activity with event volumes and timing. The spatial pattern reflects the pattern
20  shown on the map of cumulative calving height (Fig. 6). In sector D, fewer and smaller events were observed than in the sectors SL, SM and SR. The four subsectors of the shallow front show well distinguishable calving event volume patterns throughout the observation period. In the central, very shallow sector M less calving events were observed, but several of them are significantly larger than those observed in the other sectors. Interestingly, the cumulative calving height in this area is

almost three times smaller than the shallow sectors SL and SR and similar to the values observed in sector D. Sector SL with the highest cumulative calving height also has a large number of events, but they are substantially smaller than for sector M. Figure 7c shows continuous calving activity without any obvious temporal pattern throughout the different sectors. The only visually observable cluster of calving events was detected on 26 August in the afternoon when a phase with many big events

5    in the sectors M and SR occurred. A strong spatial variation in observable calving volumes and fluxes along the front is visible in Figure 7d. The sectors SL and SR contribute the highest volumes, whereas only little calving was observed in sector D. Given the observations of Figure 7d the important question arises of how much ice mass loss at the calving front remained undetected by the TRI. Assuming similar mass fluxes over the front and a constant front position only about 10% of the mass loss is detected in sector D.

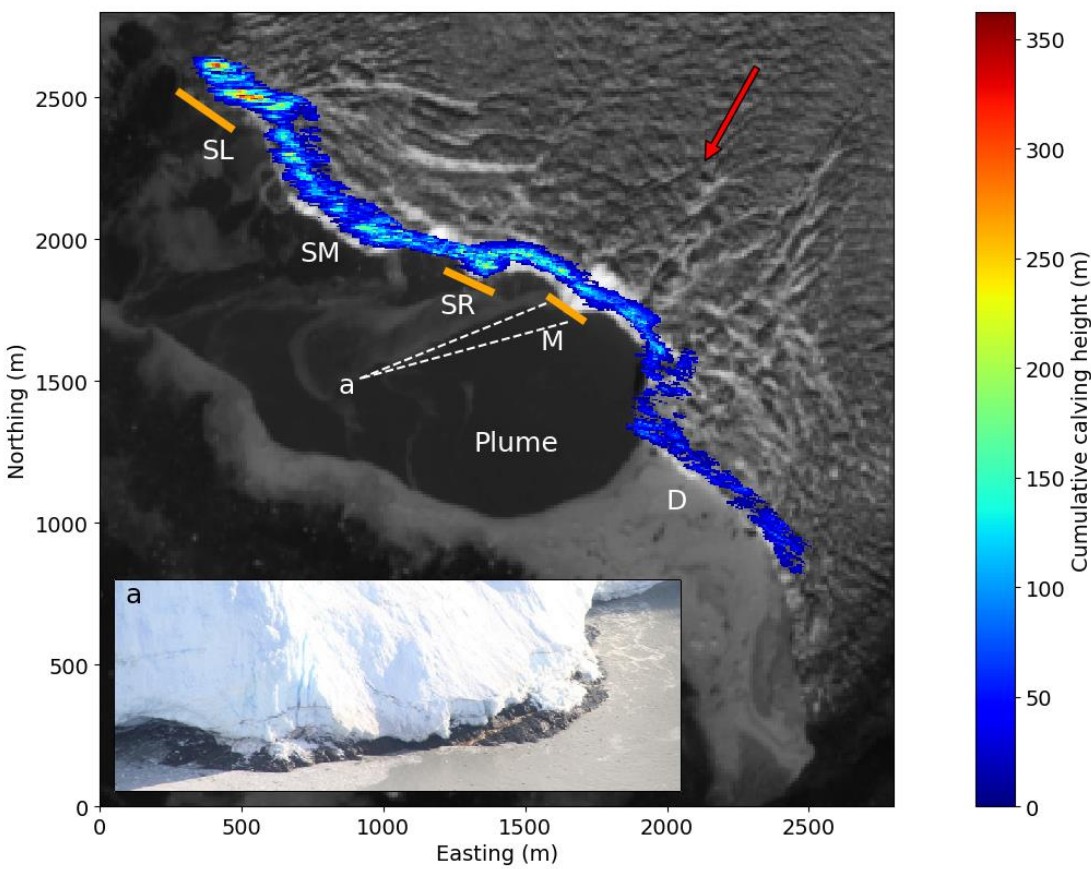

**Figure 6: Spatial distribution of cumulative calving height during the 6-day measurement period. The capital letters correspond to the sectors of the calving front (see also Fig. 7a). The deep sector (D) shows lower values than the shallow sector. Variations within the shallow sector were used to define the sectors SL, SM, SR and M. Orange lines indicate areas where bedrock was observed at**

15    **the base of the front; an example is shown in the inset (position and view angle of inset photograph is indicated by letter 'a' and dashed white lines, respectively). The meltwater plume due to subglacial discharge is well visible. Background: Sentinel-2A scene from 3 August 2016 (from ESA Copernicus Science Hub: https://scihub.copernicus.eu).**

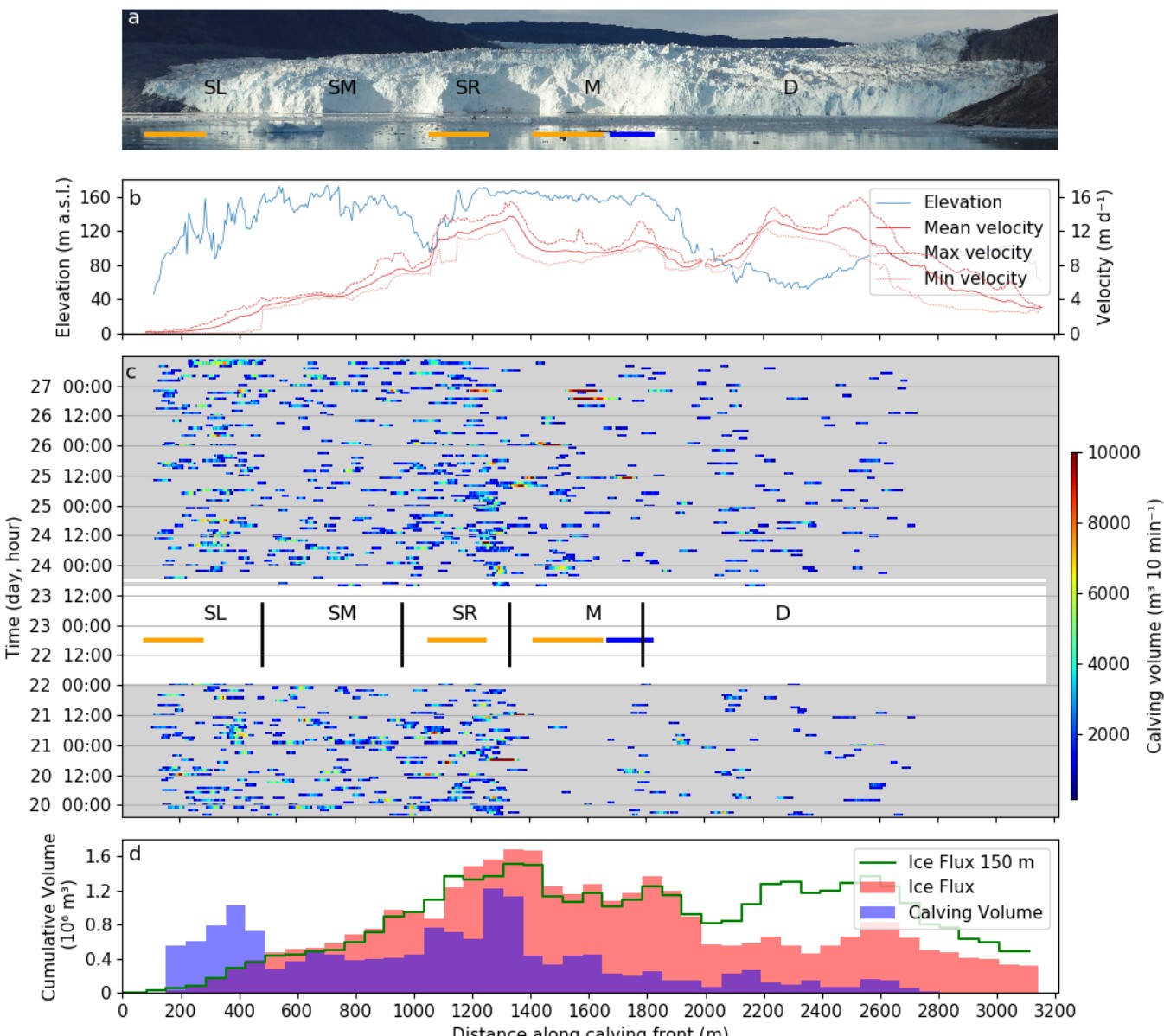

**Figure 7: The calving front of Eqip Sermia with all calving volume measurements. (a) The calving front with indication of sectors with specific calving behaviour. The differences in geometry between the sectors SL, SM, SR (steep) and sector D (flat) are well visible. (b) Elevation and velocity of the cliff top along the glacier front show strong variations. (c) Observed calving volumes in m³ along the front over time (20 to 27 August 2019). In the data gap (white area) the corresponding front sectors are marked. The orange lines indicate bedrock outcrops and the blue line represents the location of the meltwater plume. (d) Cumulative calving volume and ice flux (per bin width of 55 m) in m³ along the front. The ice flux is calculated with the corresponding front height above sea level and velocity and with an assumed ice thickness of 150 m (termed as 'Ice Flux 150 m').**

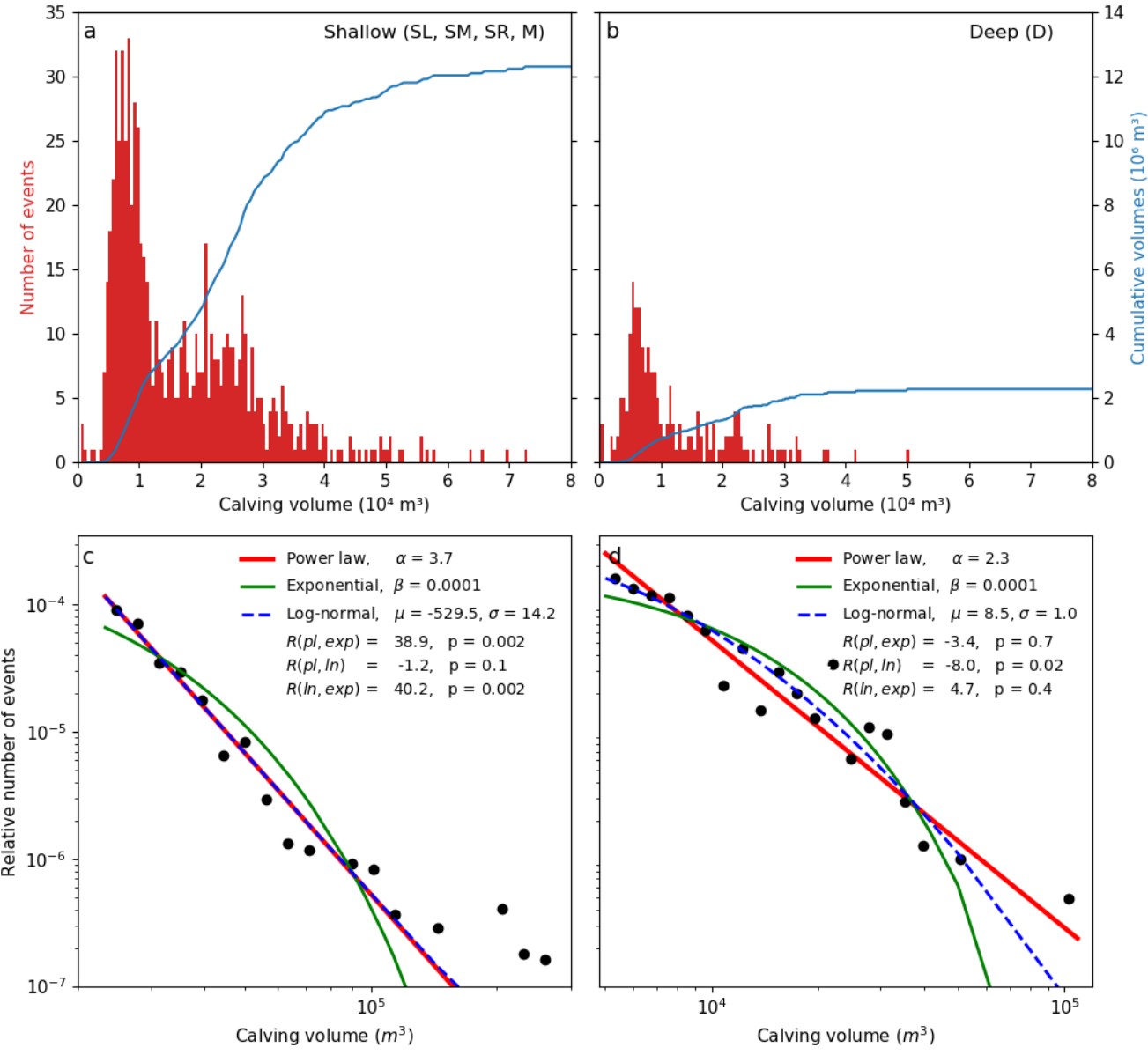

**Figure 8: Event size statistics of the observed calving events. (a) The size distribution of the calving events for the shallow sector (b) and the deep sector. (c) Distributions of calving event sizes for the shallow (d) and the deep sectors. Blue, red and green lines represent the best fit power law, exponential, and log-normal distribution.**

## 4.4 Calving event size distribution

The sizes of the calving events from the different sectors were analysed statistically with the methods described in section 3.1.

The calving event size distributions were compared with non-linear fitting models to investigate if a self-organised critical

system can be observed. The event size statistics were studied separately for the shallow sectors (SL, SM, SR, M) and the deep sector D and are shown in Figure 8. The distributions of the event sizes differ substantially between the shallow and the deep sector in the number of events (Fig. 8a and b), whereas the shapes of the event size distributions are similar. This results in a much lower cumulative volume of sector D, illustrated by the blue lines in Figures 8a and b. The result of the maximum-likelihood method is shown in the Figures 8c and d. The maximum-likelihood method uses the two values R and p to describe the best fit. The probability value p should be $\geq 0.1$ and tells if one can trust the sign of the log-likelihood ratio R. If R is positive the first model fits better, while if it is negative the second model is more likely. Both the power law and the log-normal model seem to explain well the event size distribution for both, the shallow and the deep sector. Comparing the different models to test, which model can describe the observed event size distribution better, results in a better fit of the log-normal model as compared to the power law model for the shallow sector (R = -1.2, p = 0.1) (Fig. 8c). The event size distribution of the deep sector is better represented by a log-normal model than by an exponential model (R = 4.7, p = 0.4), but comparing the power law and the log-normal model shows no significant better representation (R = -8.0, p = 0.02).

## 4.5 Pressure sensor records

Figure 9 shows the time series of short-term variations in the fjord water levels caused by calving events and recorded by the pressure sensor. The calving waves have an amplitude of up to 3.3 m and their duration ranged from several minutes up to about 50 min (Fig. S10). The wave events caused by larger calving events are recorded with a time delay of 3-4 min to the corresponding calving event. The calving-induced wave events are often difficult to attribute to single calving events due to reflection from fjord sides and superposition with subsequent events. Two types of wave oscillations can be observed: The first and most common type has a sharp onset in wave amplitudes, which are slowly damped (left inset of Fig. 9). The second type is more symmetric with a gradual increase and decrease of wave amplitude (right inset of Fig. 9).

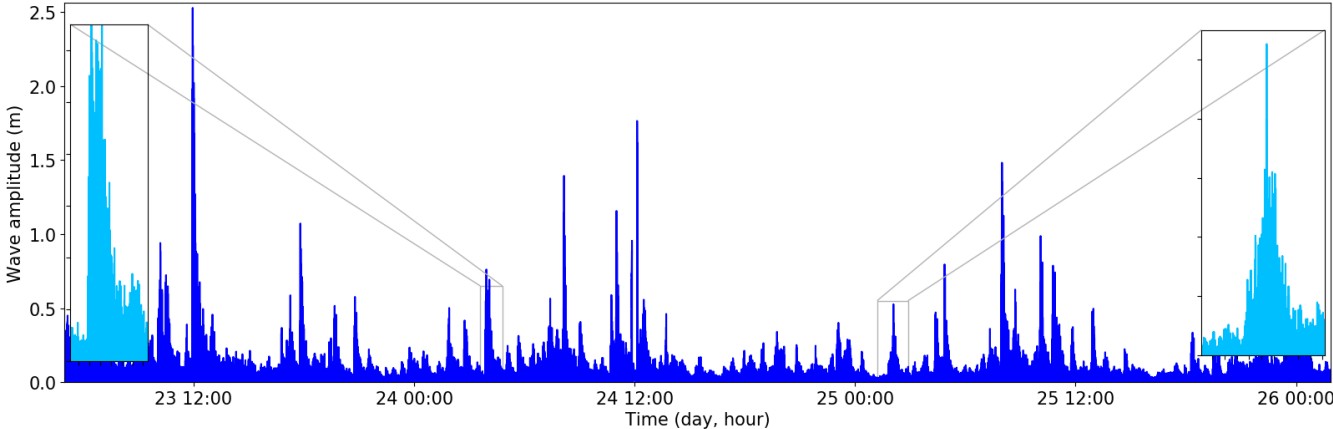

**Figure 9: Calving waves detected with a pressure sensor. The light blue inset panels show details of the two wave types due to calving events. The left one has a sharp onset, while the right one shows gradual increase and decrease of wave amplitude.**

## 5 Discussion

Using a terrestrial radar interferometer we established a detailed and continuous 6-day record of calving event volumes along the whole calving front. The detected calving event volumes were highly variable and ranged over four orders of magnitude, consistent with other studies of grounded tidewater glaciers (Chapuis and Tetzlaff, 2014; Pętlicki and Kinnard, 2016; Minowa et al., 2018). The observed calving events show no obvious temporal or spatial pattern, except for a series of bigger events on 26 August.

### 5.1 Relation to ice flux

The detected total calving volume is smaller than the ice fluxes estimated from the flow speeds and the frontal height except for sector SL, where the calving volume is too high (Fig. 7d). For the sectors SM and SR the detected cumulative calving volume is about 65% of the estimated ice flux, while for sector M the calving volume is only about 25%. For sector D the cumulative calving volume is about 15 % of the estimated aerial ice flux, while for an assumed total front thickness of 150 m the ice flux is 90% larger than the calving volume. Assuming a total ice thickness of 150 m for both sectors to calculate the total ice flux seems reasonable as this corresponds to the approximate height of the shallow sector and no signs of changes in the ice flux and ice thickness can be seen upstream of the glacier. This suggests that within the deep sector a large fraction of the ice removed at the terminus is missing from the TRI-calving detection. This missing calving volume of $17.7 \cdot 10^6$ m$^3$ can be explained by three main processes.

First, the missing volume may be removed by oceanic melt below the water line. The relatively warm saline water provides energy for ice melt where there is contact. Oceanic melt has been shown to be an important process in the mass balance of Greenland's glaciers with estimates of summer melt rates at Eqip Sermia of 0.7 m d$^{-1}$ for 2008 (Rignot et al., 2010). Assuming an ice thickness of 100 m below the water line for sector D this would result in a total oceanic melt volume of $0.47 \cdot 10^6$ m$^3$ during the observation period. However, Beaird et al. (2015) showed that this estimate is likely too small as they found a ratio of surface melt water derived to submarine melt of 26% within the fjord, which would result in higher submarine melt rates of 4 m d$^{-1}$ when considering the melt water discharge in summer of Rignot et al. (2010). This higher melt rate would over the observation period result in a total mass loss through oceanic melt of $2.7 \cdot 10^6$ m$^3$, which is however still substantially smaller than the estimate of the ice flux for the deep sector (Fig. 7d). For the shallow sector oceanic melt is likely less pronounced as the contact area exposed to ocean water is with a water depth between 0 and 20 m much smaller.

The second process explaining the missing volume is subaqueous calving, which cannot be detected with the TRI. In-situ observations by the authors and inspection of high-rate time-lapse camera imagery (Fig. 10) indicate that subaqueous calving is a frequent process but only occurs in sector D.

The third process is frequent calving of small volumes. Filtering of the TRI-data for event sizes smaller than 660 m$^3$ leads to a reduction of uncertainty but discards the potentially frequent small events below the detection limit. At the deep sector small, not detectable events are likely more frequent and contribute more to the cumulative volume due to undercutting of the calving

front caused by oceanic melt. If the missing volume is indeed dominated by undetected small calving events, our data would suggest that the calving style in the deep sector is dominated by very small but frequent calving events.

The calving at the southern side of sector M may also be affected by undercutting through enhanced submarine melt caused by the subglacial meltwater plume (blue bar in Figs. 6 and 7; Fried et al., 2015; Fried et al., 2019) and produce small and

undetectable but frequent calving events. Indeed, the TRI record only shows a few small events and several larger events on 20 and 25 August resulting in low total cumulative calving volumes (Fig. 7d).

In summary, for the deep sector the three processes of oceanic melt, subaqueous calving and calving events with small volumes provide together up to 90 % of the mass removal, while for the shallow sector calving of small volumes dominates and would explain the missing volume of about 35 – 40 %.

**5.2 Influence from cliff height and shape**

The shallow sector of the front with an inclined and higher ice cliff not only shows more but also larger calving events than the deep sector. This can be explained by the different geometries, which have an impact on the calving type as the stress regime is different. Mercenier et al. (2018) showed that an inclined ice cliff results in lower stresses, which can result in larger stable heights of the ice cliff and as a consequence at the shallow sector the calving events can release larger ice volumes. At

the vertical front of the deep sector therefore smaller calving events are expected, which consistent with the observations may not be detectable with the TRI. Further, our calving event record suggests that the geometry of the front (cliff height, slope and water depth) has an important control on the calving type. Calving events in the deep sector mostly occur as whole blocks or towers that fall into the water (visual observation by the authors). In contrast, for the sectors SL, SM, SR and M the calving events can be described mostly as avalanche like blocks or seracs that are shearing off.

The higher volumes and frequency detected for the sector SL (Fig. 7) can be explained by a rock ridge below the front of this sector. There, the water is very shallow and calving can be detected over almost the full frontal thickness. The strongly episodic but very large calving events in sector M (Fig. 7) might be related to a rock ridge over which the front is pushed (Fig. 6). Mercenier et al. (2018) found that for a smaller water level in front of the glacier stress maxima tend to reach further upstream and hence likely larger calving sizes occur.

**5.3 Calving event size distribution**

The size distribution of calving events for the shallow and the deep front are well represented by both a log-normal and a power law model. A comparison between the two models using the maximum-likelihood method indicates that the shallow sector is better represented by a log-normal model, while for the deep sector none of the two models fits significantly better than the other (Figs. 8c and d). The power law exponent of the deep sector is with $\alpha = 2.3$ in the range of other studies, which

found an exponent between 1.2 and 2.1 (Chapuis and Tetzlaff, 2014; Åström et al., 2014; Pętlicki and Kinnard, 2016).

As for the shallow sector the event size distribution can be better represented by a log-normal model, it is unlikely that this sector has the characteristics of a self-organized critical system. However, for the deep sector this cannot be excluded as neither

the log-normal nor the power law model is significantly better. Other studies found a clearer power-law distribution and concluded that the calving process shows characteristics of a self-organised critical system (Chapuis and Tetzlaff, 2014; Åström et al., 2014; Pętlicki and Kinnard, 2016), A potential difference between the shallow and the deep sector in the event size distribution leads to the suggestion that the dominant mechanisms of break-off are different. This suggestion seems

5    reasonable as for the shallow front the contact area exposed to sea water is small and thus submarine calving is less important. A study of Kirkham et al. (2017) supports those findings as they suggest by looking at size distribution of icebergs that a reduction of the number of mechanisms in their disintegration and thus a lower complexity leads to the transition from power law to log-normal distributions. To verify this suggestion and for a clear assignment of the deep sector to one of the proposed models more events would be needed. Also the event size distribution might change if a longer observation period is used as

10    the calving activity is not constant over time.

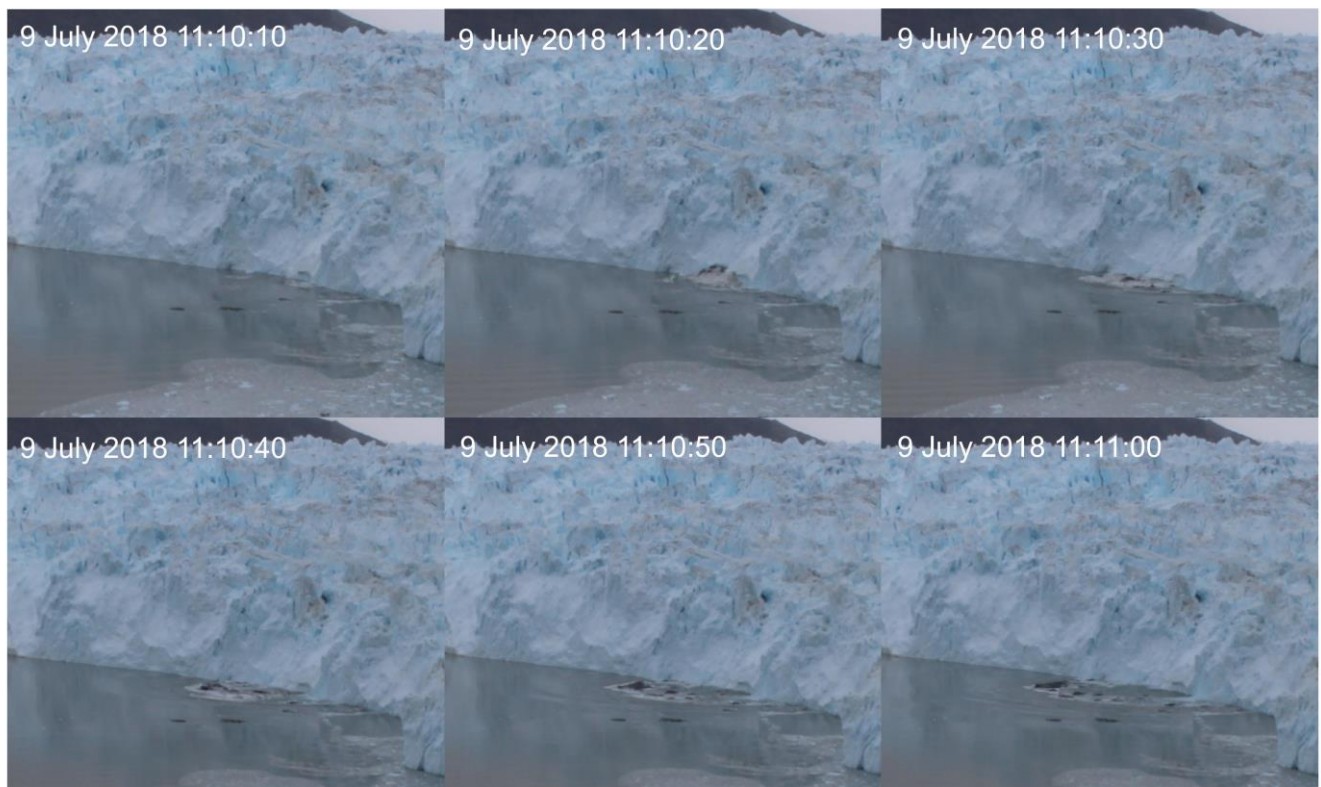

**Figure 10: An example of a subaquatic calving event recorded with a time-lapse camera in 2018. Pictures were taken every 10s.**

15    **5.4 Comparison with pressure sensor data**

Figure 11 shows a comparison of pressure sensor data and detected calving events during a 12-hour period. In addition, in Figure 12 peaks detected in the wave amplitudes are shown in comparison with the TRI derived calving events. Bigger events

are clearly visible in both data sets. In the pressure sensor data, those events mostly are of the first asymmetric type described in section 4.5 and displayed in Figure 9. The second symmetric type can be found in the pressure sensor data, but in general, they cannot be clearly assigned to a single event in the TRI dataset. These symmetric wave peaks, like the one at 2:00 on 25 August (Fig. 11), likely are due to larger subaqueous calving events in the deep sector as detected by the time-lapse camera

(Fig. 10) (Sect. 5.1) with big up-floating icebergs that cannot be detected by the TRI. These subaquatic calving events could explain parts of the missing calving volume. This reasoning is supported by other studies who found that aerial events have a gradually decreasing amplitude after the maximum wave amplitude, while subaqueous calving events showed no clear onset and a sudden drop of the amplitude after the maximum wave amplitude (Minowa et al., 2018). Also an experimental study showed that for aerial events the largest wave is earlier than for buoyancy driven events (Heller et al., 2019). For verification

of this distinction between subaquatic and aerial calving events additional observations, such as time-lapse cameras with a high temporal resolution, would be required.

In summary, the pressure sensor data together with the calving volume record (Fig. 11) indicate that large events can be well detected from pressure sensor data. Thus, pressure sensor observation could be exploited as a simple method to derive calving event numbers, volumes and potentially even calving style (aerial or subaqueous). However, the analysis of pressure sensor

data remains challenging as subglacial hydrological events, overturning of icebergs and superposition of reflected signals also produce waves and obstruct the recorded signal.

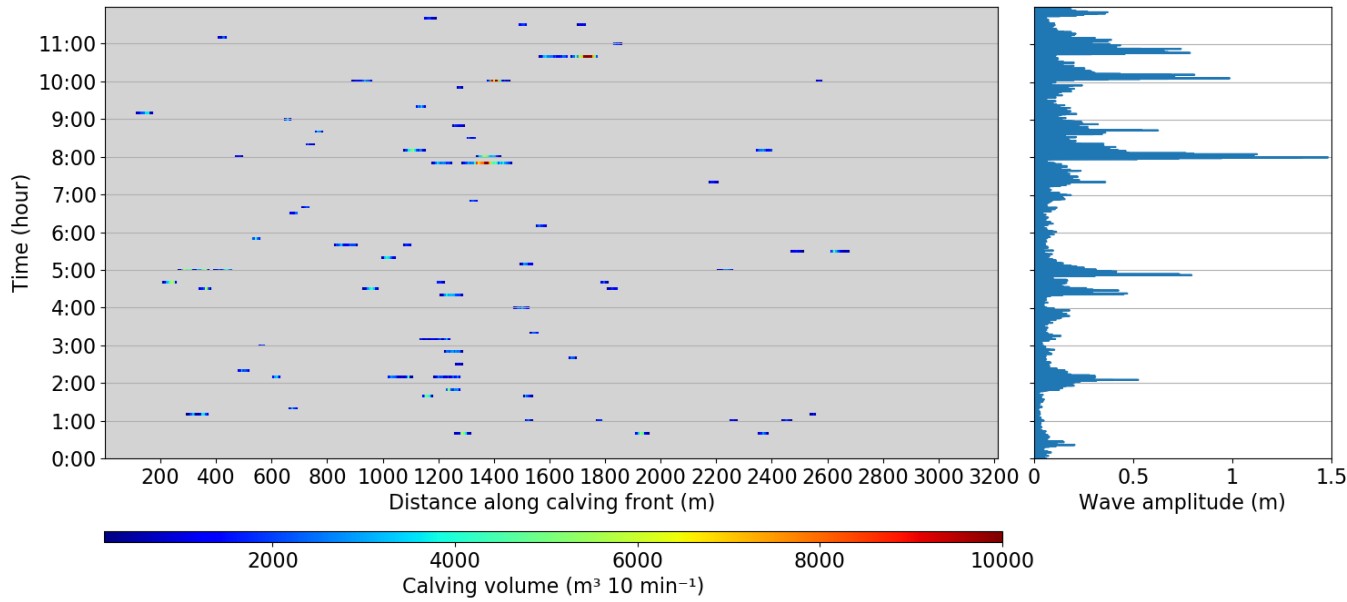

**Figure 11: Comparison between pressure sensor derived wave amplitudes (right) and detected calving events (left) for a 12 hour**

**period on 25 August. Big calving events are clearly visible in both data sets.**

## 5.5 Relation to external forcings

Calving activity has been hypothesized to be triggered by external forcings such as changes in stress state due to tides (Bartholomaus et al., 2015) and melt water accumulation in crevasses (Benn et al., 2007). Therefore, calving activity might be linked to high air temperatures and incoming radiation leading to surface melt.

Figure 12 compares air temperature, incoming shortwave radiation and tides with volume and number of calving events for the second part of the observation period (the first part is shown in Fig. S11). This comparison does not show any obvious relationship, but as the observation time of 6 days is rather short, we cannot exclude the influence of environmental forcings on calving activity. Consistent with our observations, Pętlicki and Kinnard (2016) and Chapuis and Tetzlaff (2014) also found that the calving activity during their observation period of a few days was not dependent on environmental forcings, while others found an influence of ocean temperature on calving activity over seasonal timescales (Luckman et al., 2015; Schild et al., 2018).

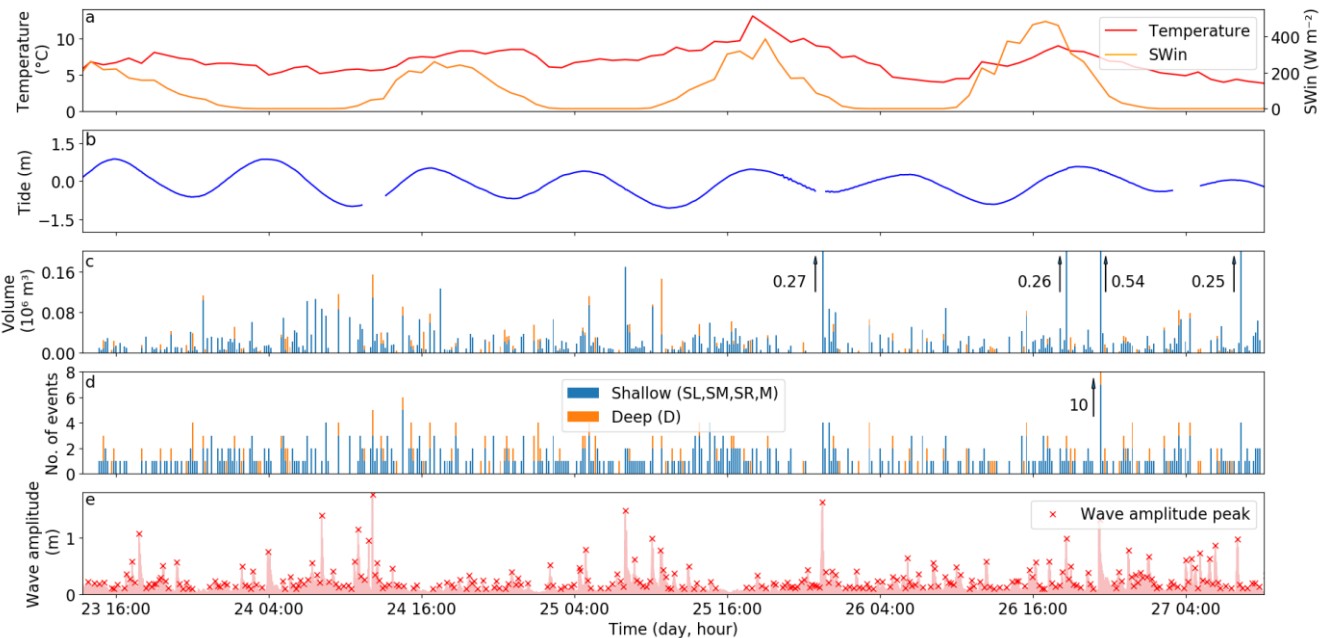

**Figure 12: Comparison between forcing and detected calving during a 3 day period. (a) Air temperature and incoming shortwave radiation from the AWS1. (b) Tides in meters. (c) Volume of calving events in m³ for the shallow and deep sectors. (d) Number of calving events. The calving events in the deep sector are plotted above those in the shallow sector. (e) Pressure sensor derived wave amplitudes and detected peaks.**

## 6 Conclusion

We developed a novel calving detection method applicable to high-rate TRI scans of glacier calving fronts. By differencing high-resolution DEMs generated from the TRI data, a detailed calving event catalogue was established, providing timing, source area and calving volume of aerial calving events.

The calving front of the observed glacier is characterized by sectors of different water depth and front height. The shallow sector features an inclined front and frequent calving events release larger ice volumes, whereas the deep sector produces less and smaller icebergs. A rock ridge in the centre of the calving front influences the calving activity there and leads to fewer but larger events.

During the 6-day observation period a total of 906 calving events were detected, of which 80% occurred in the shallow sector where mean calving volumes were 35% larger than in the deep sector. Since ice flux in both sectors is of similar magnitude, processes other than aerial calving seem to remove an important fraction of ice in the deep sector. Our analysis shows that the mass loss due to subaqueous calving, oceanic melt, and small aerial calving events contribute 90% to the total mass loss. Further, the event size distribution differs between the sectors, and fits a log-normal model in the shallow sector, whereas for

the deep sector both a log-normal and a power law model fit well but none significantly better. These differences in calving behaviour are clearly linked to basal topography and calving front geometry.

Comparison of the calving events with wave data registered with a pressure sensor shows that big events are clearly discernible in both data sets. Several events detected in the wave record, that do not occur in the TRI data, show a different wave characteristic, and likely correspond to subaqueous calving events. For the time span of the observations no obvious

relationship between the observed calving activity and environmental forcings, such as tides, temperature and incoming shortwave radiation, could be established.

This study shows the potential of detailed high-rate observations to elucidate the processes and forcings leading to iceberg calving from tidewater glaciers. The resulting statistics of calving event sizes in relation to geometry, bathymetry and external forcings are important benchmarks for calving models. Testing and calibrating such models with field data is mandatory for

the understanding of the delicate dynamics of outlet glaciers which control the evolution of large parts of the Greenland ice sheet.

*Code and data availability.* Data and codes are available from the authors upon request.

*Competing interests.* The authors declare that they have no conflict of interest.

*Acknowledgments.* We thank Diego Wasser and Rémy Mercenier for help during the field campaign at Eqip Sermia and Gamma Remote Sensing AG for the support during the data analysis. We acknowledge the reviews by Surui Xie and Pierre-Marie Lefeuvre and the additional comments by the scientific editor, which were of great help when revising the paper and have significantly improved its quality. We thank Christoph Rohner for providing the drone data for the velocity map. This work was funded by the Geographical Institute, University of Zurich and VAW, ETH Zurich, Switzerland. The field work was

supported through Swiss National Science Foundation grants 200021-156098 and 200021-153179/1.

*Author contributions.* A. Walter, M. P. Lüthi and A. Vieli designed the manuscript. A. Walter performed all analysis and wrote the draft of the manuscript. All authors contributed to the final version of the manuscript.

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
