# Peer review of "Calving event size measurements and statistics of Eqip Sermia, Greenland, from terrestrial radar interferometry"

_The Cryosphere, 2019_

## Referee Comment (RC1) · Surui Xie (Referee) · 29 May 2019

Summary:

This manuscript presents detailed observations of the glacier front of Eqip Seria, Greenland with terrestrial radar interferometry (TRI). Digital elevation models were derived from TRI measurements and used to generate calving event data. By analyzing calving event numbers/sizes and their temporal and spatial distributions, the authors concluded that the deep and shallow water sectors of the glacier front have different calving statistics and styles: 1) In the deep water sector (70-100 m), TRI observed calving events are less frequent, calved icebergs are smaller and the sizes follow a

power law distribution. Subaquatic calving is prominent here, combined with oceanic melt, they contribute up to 75% of the frontal mass loss. 2) In the shallow water sector (0-20 m), TRI observed calving events are more frequent, with much larger ice blocks (single volume larger than $5 \times 10^4$ m^3) occasionally calved, and the sizes of all calved volumes follow a log-normal model. Several possible reasons for the different characteristics in observed surface calving events were discussed, including subsurface melt, subaqueous calving, cliff height and shape, and bed topography. The authors concluded that subsurface melt and calving are the major contributors to mass loss in the deep water sector, but are trivial in the shallow water sectors. On the other hand, an inclined front geometry at the shallow water sectors results in a thicker and stable ice cliff, allowing larger potential ice volumes to calve. Besides, a rock ridge at the shallow water sectors caused less frequent but much larger calving events. In addition, several other types of data (i.e., air temperature, shortwave radiation and tide) were presented but no relationship with calving was established.

The manuscript is well organized, overall well written but can be improved. The topic meets the scope of The Cryosphere, the techniques of TRI and pressure sensor are suitable for this type of study, the references are appropriate. However, I do not think the current manuscript is suitable for publication in The Cryosphere. Below I list some major problems in the data analysis and interpretation. I encourage the authors to address these questions and consider a resubmission.

Major comments:

1) Uncertainty in TRI derived elevation models needs better assessment. The authors randomly choose 30 DEMs and computed the variability (its definition needs to be provided in the manuscript, please see my detailed comments) as a measure of the precision. Although the mean variability is 1 m, but the maximum variability is 5 m. Therefore, the DEMs are likely to have an uncertainty of ~1 m to several meters. Although a threshold of 5 m elevation decrease between adjacent DEMs is used to determine calving events, but note that even among only 30 DEMs there is a variability of

5 m between two DEMs. The calving statistics of this manuscript come from hundreds of DEMs, several large random errors (2-sigma or above) or outliers can significantly change the results. I suggest the authors to provide more details on error analysis.

2) Based on the calving data derived from TRI elevations, the authors concluded that surface calving is more frequent in the shallow water sectors, and the sizes are generally larger. This seems apparent if just looking at Figure 5c. However, due to lack of rigorous uncertainty analysis, I think this conclusion is hasty and may be flawed. In general, noise in TRI measurements increases with distance, and can increase rapidly at a distance of 4-6 km. Glacier front on the northwestern section (shallow water sectors in this manuscript) is further from the radar than the southeastern section (deep water sector), thus radar data on the northwestern section of the glacier should be noisier if all other conditions are similar. Besides, the northwestern section of the glacier front is crevassed heavier than the southeastern section (Figure 1), and elevation changes rapidly (inclined at a slope of 50 degrees according to the authors), both are more likely to induce phase unwrapping errors than a flat and less crevassed surface. These (i.e., increased noise with distance, phase unwrapping problems) could be some of the reasons why the identified calving volumes are more variable and the cumulative calving volumes are larger along the SL/SM/SR/M sectors. In Figure 5c, timing and sizes of calving events at different distances look random, but considering the characteristics of radar noise, it is important to examine if the observed pattern is due to noise or unwrapping errors. Here I suggest one possible method to test how much noise affected the distribution pattern in calving events: using the same analysis approach as presented in the manuscript, but apart from calculating calving volume based on pixels whose elevation decreased by >5 m, the authors can also calculate "increased volumes" by pixels whose elevations increased by >5 m. If a similar distribution pattern as in Figure 5c is seen, then the derived "calving volumes" are likely disturbed. The authors can probably add a plot of such "increased volumes" to the negative side of y-axis in Figure 5d (can used light blue color if the authors don't want it be distracting). A comparison figure of "detected increasing volume" similar as Figure 5c can also add

important information to the manuscript, and it can go to the supplement if the authors would like to save space in the main manuscript.

3) According to the manuscript, there was very little surface calving observed by TRI at sector D (the deep water sector), and mass loss due to subaqueous calving is dominant (50% or more, depends on the rate of oceanic melt) here. Limited evidence of subaqueous calving was shown in the manuscript. Even if substantial subaqueous calving events occurred and contributed significantly to the mass loss at the deep water sector, the manuscript failed to explain where the mass goes. I also think it is not adequate to simply assume that subaqueous calving is independent of TRI observed surface calving. If subaqueous calving would not cause surface elevation change by following the authors' logic, then what was there to fill the space left by the "subaqueous calving"? Besides, if TRI observed little calving at sector D, then glacier front at this section should advance, especially at the high velocity area. Speed in the middle of this sector is 16 m/day, in ∼7.65 days, ice front here can advance over 100 meters, much larger than the resolution of either Landsat/Sentinel satellites or TRI images so should be detectable. However, Supplementary Figure S1 rejected this.

4) The manuscript did not explain the method they used to choose the study area for calving detection well. Although on page 6 the authors mentioned that they applied a mask with ∼150 m wide across the glacier front, however, the glacier front was constantly moving, so a Lagrangian frame should be used. Whatever the reference frame was, according to the methods presented by the authors, areas with calving event detected (Figure 4) over the center of sector D should have the largest along-flow direction width. This is because glacier front at this location should advance (also see comment above), thus the test area should move. Whereas Figure 4 shows a different pattern.

I think the above comments can already warrant a rejection or resubmission. There are also many other problems with the current manuscript, such as inconsistent numbers and unjustified assumptions. However, I do believe that the authors are working

towards the right direction. Hopefully these comments can help.

Sincerely,

–Surui Xie

I attach detailed comments as a supplementary file.

Please also note the supplement to this comment:
https://www.the-cryosphere-discuss.net/tc-2019-102/tc-2019-102-RC1-supplement.pdf

**Supplement:**

**Reviewer:** Surui Xie

**Summary:** This manuscript presents detailed observations of the glacier front of Eqip Seria, Greenland with terrestrial radar interferometry (TRI). Digital elevation models were derived from TRI measurements and used to generate calving event data. By analyzing calving event numbers/sizes and their temporal and spatial distributions, the authors concluded that the deep and shallow water sectors of the glacier front have different calving statistics and styles: 1) In the deep water sector (70-100 m), TRI observed calving events are less frequent, calved icebergs are smaller and the sizes follow a power law distribution. Subaquatic calving is prominent here, combined with oceanic melt, they contribute up to 75% of the frontal mass loss. 2) In the shallow water sector (0-20 m), TRI observed calving events are more frequent, with much larger ice blocks (single volume larger than $5 \times 10^4$ m³) occasionally calved, and the sizes of all calved volumes follow a log-normal model. Several possible reasons for the different characteristics in observed surface calving events were discussed, including subsurface melt, subaqueous calving, cliff height and shape, and bed topography. The authors concluded that subsurface melt and calving are the major contributors to mass loss in the deep water sector, but are trivial in the shallow water sectors. On the other hand, an inclined front geometry at the shallow water sectors results in a thicker and stable ice cliff, allowing larger potential ice volumes to calve. Besides, a rock ridge at the shallow water sectors caused less frequent but much larger calving events. In addition, several other types of data (i.e., air temperature, shortwave radiation and tide) were presented but no relationship with calving was established.

The manuscript is well organized, overall well written but can be improved. The topic meets the scope of The Cryosphere, the techniques of TRI and pressure sensor are suitable for this type of study, the references are appropriate. However, I do not think the current manuscript is suitable for publication in The Cryosphere. Below I list some major problems in the data analysis and interpretation. I encourage the authors to address these questions and consider a resubmission.

**Major comments:**

1) Uncertainty in TRI derived elevation models needs better assessment. The authors randomly choose 30 DEMs and computed the variability (its definition needs to be provided in the manuscript, please see my detailed comments) as a measure of the precision. Although the mean variability is 1 m, but the maximum variability is 5 m. Therefore, the

DEMs are likely to have an uncertainty of ~1 m to several meters. Although a threshold of 5 m elevation decrease between adjacent DEMs is used to determine calving events, but note that even among only 30 DEMs there is a variability of 5 m between two DEMs. The calving statistics of this manuscript come from hundreds of DEMs, several large random errors ($2\sigma$ or above) or outliers can significantly change the results. I suggest the authors to provide more details on error analysis.

2) Based on the calving data derived from TRI elevations, the authors concluded that surface calving is more frequent in the shallow water sectors, and the sizes are generally larger. This seems apparent if just looking at Figure 5c. However, due to lack of rigorous uncertainty analysis, I think this conclusion is hasty and may be flawed. In general, noise in TRI measurements increases with distance, and can increase rapidly at a distance of 4-6 km. Glacier front on the northwestern section (shallow water sectors in this manuscript) is further from the radar than the southeastern section (deep water sector), thus radar data on the northwestern section of the glacier should be noisier if all other conditions are similar. Besides, the northwestern section of the glacier front is crevassed heavier than the southeastern section (Figure 1), and elevation changes rapidly (inclined at a slope of 50 degrees according to the authors), both are more likely to induce phase unwrapping errors than a flat and less crevassed surface. These (i.e., increased noise with distance, phase unwrapping problems) could be some of the reasons why the identified calving volumes are more variable and the cumulative calving volumes are larger along the SL/SM/SR/M sectors. In Figure 5c, timing and sizes of calving events at different distances look random, but considering the characteristics of radar noise, it is important to examine if the observed pattern is due to noise or unwrapping errors. Here I suggest one possible method to test how much noise affected the distribution pattern in calving events: using the same analysis approach as presented in the manuscript, but apart from calculating calving volume based on pixels whose elevation decreased by >5 m, the authors can also calculate "increased volumes" by pixels whose elevations increased by >5 m. If a similar distribution pattern as in Figure 5c is seen, then the derived "calving volumes" are likely disturbed. The authors can probably add a plot of such "increased volumes" to the negative side of y-axis in Figure 5d (can used light blue color if the authors don't want it be distracting). A comparison figure of "detected increasing volume" similar as Figure 5c can also add important information to the manuscript, and it can go to the supplement if the authors would like to save space in the main manuscript.

3) According to the authors, there was very little surface calving observed by TRI at sector D (the deep water sector), and mass loss due to subaqueous calving is dominant (50% or more, depends on the rate of oceanic melt) here. Limited evidence of subaqueous calving was shown in the manuscript. Even if substantial subaqueous calving events occurred and contributed significantly to the mass loss at the deep water sector, the manuscript failed to explain where the mass goes. I also think it is not adequate to simply assume that subaqueous calving is independent of TRI observed surface calving. If subaqueous calving would not cause surface elevation change by following the authors' logic, then what was there to fill the space left by the "subaqueous calving"? Besides, if TRI observed little calving at sector D, then glacier front at this section should advance, especially at the high velocity area. Speed in the middle of this sector is 16 m/day, in ~7.65 days, ice front here can advance over 100 meters, much larger than the resolution of either Landsat/Sentinel satellites or TRI images so should be detectable. However, Supplementary Figure S1 rejected this.

4) The manuscript did not explain the method they used to choose the study area for calving detection well. Although on page 6 the authors mentioned that they applied a mask with ~150 m wide across the glacier front, however, the glacier front was constantly moving, so a Lagrangian frame should be used. Whatever the reference frame was, according to the methods presented by the authors, areas with calving event detected (Figure 4) over the center of sector D should have the largest along-flow direction width. This is because glacier front at this location should advance (also see comment above), thus the test area should move. Whereas Figure 4 shows a different pattern.

I think the above comments can already warrant a rejection or resubmission. There are also many other problems with the current manuscript, such as inconsistent numbers and unjustified assumptions. However, I do believe that the authors are working towards the right direction. Hopefully these comments can help.

Sincerely,
—Surui Xie

**Detailed comments:**

Page 2, line 4: "was" —> "were", data should be plural.

Page 2, line 6: "style" —> "styles".

Page 2, line 8: "missing" —> "deficiency" ?

Page 2, lines 8-10: later in the manuscript, one conclusion was that that subaqueous calving and oceanic melt combined contribute ~75% to the frontal mass loss. However, here it seems that subaqueous calving itself contribute up to 75% of the frontal mass loss. Please clarify.

Page 2, line 18: It would be great if the authors can be more specific about "water masses in Greenland". Did the authors mean "water masses around Greenland", or "increase in surface water due to melt"?

Page 2, line 21: What are the major remaining limitations? A few examples briefly listed here would be helpful.

Page 2, line 24: I am not sure if calving controls tidewater glaciers's react to environmental condition changes, or verse visa? Please clarify.

Page 3, line 8: "was" —> "were".

Page 4, line 7: Figs. 1 and 2.

Page 5, line 13: "whole" —> "entire".

Page 5, line 18: "climate" —> "weather".

Page 6, lines 16-18: Are these differences RMS difference, Mean difference, or other types? These can be quite different because the difference between TRI-DEM and Arctic DEM can be systematically and/or randomly. Please clarify.

Page 6, line 17: Does "variability" mean "repeatability", or changes of topography? Please clarify.

Page 6, lines 17-18: How stable/random is stable/random? Maybe outline the test area in one of Figures 1, 2 or 3? And what does the "values" in line 18 mean? Values of a selected area (if

so, please outline it in Figure 1/2/3) or selected DEMs (if so, maybe mark the times in Figure 5c)?

Page 6, lines 23-24: Does the "10 pixels" mean "10-adjacent pixels"? I feel the two numbers "10 pixels" and "3 pixels" are confusing: Does noise needs to fulfill both "area<10 pixels" AND "width<3 pixels"? If so, how about a block with 3×3 (9 pixels, each pixel shown by an "O" below) shape like
"   OOO
    OOO
    OOO   ", or a 2×8 shape like
"   OOOOOOOO
    OOOOOOOO   ", or a shape like
"   OO
    OOOOO
      OOOO   "
Are these considered calving events if all "O" pixels have elevation decreases more than 5 m? Because many of the identified calving events are quite small, and shapes of these blocks may not be regular, I think it is important to clarify these settings here.

Page 6: Apart from using elevation changes to detect calving, it is also possible to identify calving from radar amplitude images. Including an example showing both changes in radar amplitude and elevation would be strong evidence that the method is reliable.

Page 7, lines 5-6: Did the authors mean that p should always be larger or equal to 0.1, or did they mean that one can only trust the sign of R when p≥0.1? If it is the latter, maybe rewrite the sentence to "which tells if one can trust the sign of R (when p≥0.1)"?

Page 7, line 15: Please also specify the low-pass frequency. Just as how the high-pass frequency was given.

Page 7, line 19: To avoid confusion for readers who are not familiar with radar, I suggest to add "line-of-sight" in front of "shadow".

Page 7, lines 26-28: I found that the number of total identified calving events is smaller than the sum of identified calving events in the shallow sector and the deep sector (1681 < 1403+289). Did I miss anything? Please also check numbers in Table 1, many of them are not consistent.

Page 8, line 2: Please ensure the minimum size of identified calving block fulfill the threshold defined for calving events (based on lines 21-24, page 6, I calculated a minimum volume $5×10×3.75×3.75=703$ m$^3$. Or did I misunderstand the "resolution"? — I picked it from line 13, page 6. Using the radar pixel specified in line 6 page 8 the minimum volume of identified block is even larger, i.e., 1500 m$^3$). If this paper aims to do statistics of calving event sizes, please ensure that the statistics are correct.

Page 8, lines 6-9: I think to calculate cumulative calving height a Lagrangian frame needs to be used, because ice at the front can move at 16 m/day (line 21 on page 7), which means a ~100 m displacement during the observation period. If the authors were referring to cumulative calving height from calved ice height at each pixel in each calving events, then line 6 needs to be rewritten, at least taken away "differences" because calved height was estimated from the difference between two DEMs.

Page 8, line 8: If cumulative calving height exceeds 300 m but not up to 300 m, please consider using "extend='max'" for the color bar of Figure 4.

Page 8: I think TRI-derived DEMs in this paper are very important data, however, there was no figure showing the DEM, neither in the main paper nor in the supplement. Please consider to include a TRI-derived elevation map in the manuscript. Maybe add a panel or two in Figure 3?

Page 9, Figure 4: Comparing this figure with the text makes me confused. If ice velocities on the two sides of sector M are the fastest (see Figure 3) but cumulative calving heights are not the largest (see colormap in Figure 4), shouldn't these two areas have the widest (along flow direction) spatial distribution of calving events? Please check data processing, and make sure the descriptions in lines 25-27 on page 6 are correct.

Page 9, Table 1: Please check calculations and make sure that numbers are consistent in the table and the main text. In the table, please at least ensure numbers in "Total event volume" equal to "Total event number" × "Mean event sizes", unless a different math was used.

Page 10, Figure 5: I like this figure! But I also have a few questions and suggestions. First, color changes from dark blue to light blue in Figure 5c is distracting, I suggest to mask out periods without calving using white or grey. In this way, calving characteristics will be more accessible. Consequently, the minimum value of the color bar can be changed to the smallest volume

detected based on the settings (lines 20-25 on page 6). Second, I am confused by the right axis of Figure 5d. Can the authors elaborate on this? Third, I could hardly read the superscripts in y-axis label of Figure 5d due to low resolution. Based on the manuscript I guess it was "$10^6$ m$^3$" in the bracket, is this correct? Please increase the figure resolution. Also, maybe add "Cumulative" to the y-axis label of Figure 5d to distinguish from the color bar label in Figure 5c. Fourth, I found that the further analysis separates these sectors, so it is necessary to show the exact along-distance ranges of different sectors. Maybe use vertical bars to mark the boundaries of different sectors? These bars can go between the annotations in Figure 5c, such as " |   SL |    SM    |    SR  |    M    |          D        |". Last, would it be possible to add a narrow column on the right of Figure 5c and show total calving volume along the entire calving front in color? Such a plot may provide useful information on calving volume changes with time.

Page 11, line 4: "already observed" —> "shown"

Page 11, line 10: It is probably not correct to say "only … was observed" because small calving volumes are also visible at sector M in Figure 5c. Maybe rewrite to something like "less calving events but several of them are significantly larger than those observed at the other shallow water sectors".

Page 11, lines 11-12: I think there is an ambiguity in"the most individual events". Maybe rewrite it to "the largest number of calving events".

Page 11, line 14: I agree that no clear temporal pattern can be seen throughout the different sectors, it looks pretty much like random noise. Please see my comments above.

Page 11, line 15-16: The "observable cluster of calving" is hard to tell from Figure 5c. Yes there are some big events, but since this manuscript does statics, in the sense of statics, do these relatively big events really clustered? Need more elaboration.

Page 11, lines 19-21: More detail of deriving the 25% needs to be provided. Is it an appropriate assumption of a constant front position? Here ice can move up to 16 m/day (Figure 3). And the assumption of a constant mass flux over the front also needs to be justified.

Page 11, line 30: A summary of the meaning of the log-likelihood ratio R would be helpful for understanding the statistics. It seems to be an important parameter describing the likelihood of

two different models. Also, were there any reasons to choose the three models here? Since one of the major conclusions came from statistics, more details should be provided.

Page 12, Figure 6: In the abstract I found "The size distribution of the deep sector follows a power law, while the shallow sector is likely represented by a log-normal model." From this figure, could we say that both can be represented by a log-normal model? The R and p values are identical for the power law and log-normal models in Figure 6d. Or did I miss anything?

Page 12, line 9: Can the authors provide the unfiltered water level during the entire observation period? So that readers will know the overall characteristics of local water level variations. It can either be in Figure 9, or go to the supplement, maybe add one panel to Figure S2.

Page 13, Figure 7: This figure is nice. I have a suggestion of another figure: plot one similar figure as Figure 7, but use the entire period (similar as Figure S2), and then plot calving volume of the entire glacier front (y-axis) versus time (x-axis) for comparison. This would help further discussion of potential relationship between calving and water level variation.

Page 13, line 14: "several clusters of events"? I thought the authors only observed one cluster of calving events (lines 14-16 on page 11). Please clarify.

Page 13, line 15: How did the authors reach to a conclusion that "no clear temporal pattern of tidal or diurnal recurrence could be detected"? Can the authors elaborate? If no evidence, I suggest to omit this sentence. Or at least admit that this is based on the impression of looking by eyes.

Page 13, line 18: I think some of these values can be calculated from the data. If the authors use values derived from real data, the further analysis would sound more reasonable. Also, maybe use "a front thickness" instead of "a front height" to avoid confusion?

Page 13, lines 18-23 and after: Please try to keep number of digits consistent.

Page 13, lines 20-21: Not sure if it is correct to say "This value should match up to ..."? Although the front position looks stable by eyes, but is this sufficient to support an assumption that total ice flux should match observed total calving volume? More rigorous analysis is needed.

Page 14, line 1: Remove "with", add "," before about.

Page 14, line 3: Please check if "160 m$^3$" is correct. And stacking does not likely contribute to the difference unless there are some errors in the data analysis.

Page 14, line 9: I bet the "0.47 · 10$^6$ m$^3$ " correspond to the deep sector? Why assuming an ice thickness of 100 m below the water line? Too many assumptions could result in significant bias. Since the authors have data of surface elevations (shown in Figure 5b), and have assumed a front thickness of 150 m (line 18 on page13, although I suggested to use real data instead of assumption), ice thickness below water line can be estimated.

Page 14, lines 16-18: If subaqueous calving cannot be detected with the TRI, how could it be detected by visual observations and time-lapse imagery? More details are needed. Maybe show some examples of images taken by the time-lapse camera.

Page 14, lines 19-12: I don't think the authors have shown enough evidence to reach this conclusion. If subaqueous calving account for ~50% of the mass removal from the deep sector, the average thickness removed at the glacier front could be calculate by doing simple math. I believe that will lead to significantly mass thickness removal and the deep sector may become afloat by the end. Plus, can the authors see icebergs coming out from subaqueous calving? If it accounts for ~50% of mass loss, then visual observations or time-lapse camera images (line 17) may be able to see icebergs coming out from subsurface. Please provide evidence to support the conclusion.

Page 14, lines 25-26: If ice cliff at the shallow sector can have larger but stable height, then why do calving events occur so frequently? Although ice here is thicker but calving should be less frequent or no calving at all because ice cliff can be stable (lines 25-26). May the authors were referring to the potential of calving so"a thick cliff CAN release larger ice volumes", but please note that here calving is quite frequent (also related to how "stable" was defined in this manuscript), while previous figures (e.g., Figure 5c) show that the shallow sectors calved more frequently at the surface. Even add subaquatic calving to the deep water sector, calving at the shallow sectors will still be more frequent than the deep water sector because the authors assumed the overall mass loss are similar in different sectors, plus the deep water sector has lost more mass due to melt.

Page 15, lines 4-7: Would enhanced submarine melt cause surface elevation decrease because ice becomes thinner? If it won't lead to surface elevation decrease, then what was there to support the upper part of the glacier? Would the empty chambers cause instability and calving? On the other hand, if it will lead to surface elevation decrease, then the TRI might be able to see the decrease. Please clarify.

Page 15, line 12: Please show the "Observed subaqueous calving events". Because the authors wrote that there were time-lapse cameras (line 17 on page 14). If no captured subaqueous calving events by instruments, then please provide more detail of the available observations by authors in the field.

Page 15, line 19: "acting" —> "dominant"?

Page 15, lines 19-24: The paragraph relies on the assumption that submarine calving in the deep sector is a major contributor to total calving volume. Please evaluate the assumption based on comments above.

Page 15, line 30: Please explain what is "big up-floating icebergs". If the icebergs are big, they might appear on the TRI amplitude images. An example image would be helpful.

Page 16, lines 4-6: Yes I agree that pressure sensor observations could be used to derive calving events, but challenges remain. One challenge is that subglacial hydrological events may cause similar signal as what has described as subaqueous calving in this manuscript. Need justification.

Page 16, lines 13-15: Perhaps this paragraph needs to be rewritten, because I don't get the logic of cause and effect. Sentences before and after "Therefore" seem to be out of place. What do tides do with air temperatures and radiation? Other readers may also be confused.

Page 17, Figure 9: In (d) and (e), are the calving events in the deep sector plotted above calving events in the shallow sector? Please note this in the caption, otherwise readers can assume that all these histograms start from 0 in the y-axis.

Page 17, line 10: Maybe "surface" should be added to the front of "calving event" because subaqueous calving was also discussed?

Page 17, lines 13-14: Or it may not have to be explained by other processes? I don't see the necessity of assuming similar ice flux in the two sectors.

Page 18, line 2: Here "center" was used, but at other places "centre" was also used. Please be consistent.

Supplement page 1, Figure S1: A more accurate job needs to be done if the front line was estimated visually, because this research studies many relatively small calving events. If there are satellite images at the beginning and the end of the campaign, then they should be plotted here. If no available satellite images at these times, TRI amplitude images can help.

Supplement page 1, Figure S2: As I commented above, please add a panel to show unfiltered water level.

Supplement page 2, Figure S3: I thought the tide data were heavily filtered, why there is a jump around 08:00 on the 20th? Please check.

---

## Referee Comment (RC2) · Pierre Marie Lefeuvre (Referee) · 2 Sep 2019

First of all, I apologise for the delay to the authors and the editor. Please contact me if you have any questions. Pierre-Marie Lefeuvre

This paper presents a detection and volume distribution of calving events at Eqip Sermia glacier in Greenland based on an eight-day campaign of Terrestrial Radar Interferometer (TRI). The authors identify calving events using elevation changes at the glacier front from TRI-derived, 10-minute-stack Digital Elevation Models (DEM). Calving volume is computed from cumulated vertical changes scaled to the area by a watershed algorithm. The volume distribution is found to change from a power law distribution (self-organised critical system) in the deep part of the calving front to a log-normal in the shallow part (less complex system). The authors infer that higher ice cliff and decrease in subaerial melting in shallower parts lead to a decrease in complexity associated with a lower number of calving mechanisms.

The topography from TRI is a relatively new technique to analyze calving at marine terminating glaciers and only applied to Ilulissat glacier (Xie et al, 2016, 2019) and Eqip Sermia (Lüthi et al, 2016). This new inventory and derived calving volumes are important to link individual calving processes to long-term calving rates as there is a general lack of high temporal direct observations. This type of quantitative dataset contributes to the theoretical description of calving as a self-critical system which is currently based on only qualitative estimates of calving volume. In the long run, the measured calving distribution will support the development of more physical calving models integrated to ice-sheet models and a better quantification of the contribution of the Greenland ice sheet to sea level rise in the future. I thus recommend its publication in the Cryosphere and this study will be of great interest for its readers.

Overall the paper is well written and structured. My major comments are listed below and I am confident that the authors can correct them.

1. DEM derivation of the glacier front from TRI

The critical part of the paper is the derivation of digital elevation models of the glacier front from terrestrial radar interferometer as developed by Strozzi et al. (2012). However, this method is known to be uncertain, although the large glacier size should help having a greater signal to noise ratio. I think that it is important to extend the paragraph on the error analysis and dedicate a specific figure with a map of the derived DEM(s), statistical distribution of the error in the discussed stable terrain. Assess the uncertainty of the glacier part with the UAV derived DEM too by replacing Figure 3 as the velocity comparison is done in Rohner et al, 2019.

I would like to have a Figure showing a study case of the detection and watershed algorithm to assess issues with signal to noise ratio and uncertainty in radar geometry or cartesian coordinates (in the main text or supplementary material).

2. Issues in determining best fit models for calving distribution

As seen on Figure 6c and 6d, it is not possible to distinguish between a power law and a log normal models as indicated by a low loglikelihood ratio $R$ between the two distributions and a poor significance value, $p>0.1$. The only evaluation possible of the power law is a comparison with other heavy-tailed distributions. The conclusion is that the shallow and deep part does not exhibit a transition in distribution from power law to log normal as they cannot be statistically differentiated from each other. Discuss instead whether the distribution over such a short period is representative.

3. Ice flux budget: bed topography and missing component

The paper bases its analysis on the depth of the fjord but no bed data is provided to support this description (just observations of surfacing rocks). Please use the BedMachine v3 to at least provide an idea of the fjord depth in front of the glacier to the reader. The shallow part may be constituted of

two bed pinning points beside a deep valley. Furthermore, simplify the subdivision of the shallow part to only the shallow part regrouping SL, SM and SR.

The simple ice flux calculation holds some caveats when identifying a missing volume due to the uncertain fluxgates and filtering of small events. The distribution of these small events may be related to calving mechanisms and ocean melting (undercutting). A more realistic flux can be derived by integrating the ice flux with your surface elevation and velocity data. See my minor comments to improve the understanding of section 5.1.

4. Better integration of calving wave dataset

The paper should integrate better the ocean wave data as an alternative dataset of calving events (including subaqueous ones?), explain this discrepancy and discuss other potential sources such as iceberg rolling. This better integration of the wave amplitude dataset with the TRI detected calving events would strengthen the discussion and conclusion of the paper.

**Minor comments:**

*page 2*

Abstract

Please mention that the study is based on derived digital elevation models. Focus on your findings right after your methods instead of following the paper structure: the characteristics of the shallow/deep part (l.5-7), then calving missing calving volume (l.8-10), self-critical system vs less complex model (l.10-11), Calving models vs front geometry (l.11-12) and finally lack of relation to air temperature and tides.

l.2: "in understanding **the processes of calving**"
l.4 "The **derived surface elevation** data with a **spatial resolution**". Specify the vertical resolution as well as it is key to the method..
l.5 can you find a better word than "source area"? "vertical front area"?

1. Introduction

It is well written and nicely placing the work in its context.

*page 3*

l.12 Add Kohler et al., 2015 in Polar Research [TOCHECK] as they produced the longest calving time series based on seismic records (20 years)
l.14 Delete "can only detect large events" as they can detect small events (even ice falling in a crevasse at the front, Kohler et al., 2019 in The Cryosphere Discussion) when placed close to the glacier front. Their main issue is the volume scaling.

2.1 Study area

l.30 Add that the glacier has been stable and even advanced since 2016 similarly than Jakobshavn Isbrae (based on Planet daily imagery).
l. 32 Indicate the time period when 16 m day-1 was obtained: "as measured over our two week period in 2016"? as the 2.5 and 5 m day-1 represents annually averaged velocities, correct?

*page 4*

l.1-7 Also present the glacier bed or bathymetry provided in the BedMachine as it covers an area that is now deglaciated (as they use an older surface elevation and glacier mask) and upstream bed geometry is also important to understand the glacier flux at the front.

*page 4-5*

The TRI and environmental data parts are complete and informative.

3.1 TRI data processing
l.15-17 Is the elevation difference just a shift in absolute elevation explained by a difference in geoid or geo-referencing problems of the Arctic DEM or your DEM? Also co-registering the two DEMs before differencing is useful to assess systematic errors outside obvious artefacts.

l.18-19 Please provide a sentence about precision change over time on stable terrain and also ice. You could plot this variability on stable terrain and some upper part of the glacier for the single DEMs and the stacked ones to appreciate the effect of atmospheric disturbances and the improvements from stacking. You could use this variation to provide first order error bars for your volume estimates on Figure 6. In the discussion could you compare your precision to what other studies found.

l.20-21 Indicate that the watershed algorithm uses elevation change as source image and merges calving events occurring within 10 minutes due to the stack. Can you define algorithm parameters like the number of start points or maximum points for reproducibility? Please plot an example of your watershed results (here or in supplementary materials) to assess the effect of noise on your segmentation. An error of 10 pixels in radar geometry already causes a volume error of 5625 m3 in range and 12000 m3 in azimuth using 150 m of ice thickness, that is the same order of magnitude than your calving volumes.

l.23-24 "10 pixels **in area**" Specify that in the context of your grid asymmetry, your area filter is more likely to remove events that are long instead of wide, thus you apply this second filter of 3 pixels. Add that 3 pixels is equal to 11.25 metres.

l.24 "When applying [...] are removed". The noise observed on stable terrain is not removed, but the signal to noise ratio is higher for the filtered events. Moreover, quantify the number or area of excluded events or give a percentage.

l.20-24 How do you deal with the zero elevation or water elevation when calving occurs along the entire ice column (i.e. column collapse)? Parts of the DEM covering the sea may have Not A Number values or problems with icebergs?

l.4-6 Rephrase the last line that explains what a "good" *p* value is and means. Specify that the maximum-likelihood methods are used because of the non-linearity of the fitted curve and that one implication is that the resulting log-likelihood is a relative score of how good two fitted models perform against each other instead of "an absolute score".

l.8 Add that 120 interferograms is approximately 2 hours.

l.7-10 Indicate the theoretical maximum velocity that the TRI measures with an interval of one minute (it should be of the order of 6 metre per day) as this will be useful to explain the differences with the UAV velocity data. [TAZIO equation]

3.2 Pressure sensor data processing
l.15 Indicate the frequency of the low pass filter or used methods.

4. Results
The key result of the paper is the TRI-derived DEMs but the velocity (4.1) is presented first instead. Add text and figure(s) specifically on the generated DEMs and signal improvement by stacking before section 4.2 on calving detection results or a comparison with the UAV DEM.

4.2 Magnitude and source of area of calving events
Abandon the subclassification of the shallow sector as it confuses the results
l.25 I suggest a simpler section title: Area and location of calving events?

l.29 Add the number of filtered/removed events for each sector as the filter may affect the number of calving events. I have a hunch that the deeper sector may have more small events, likely filtered out, due to the effect of higher submarine frontal ablation.

l.1 Replace "frequencies" by "number"
l.2 "**four** orders of magnitude".
l.2 Use the same order of magnitude i.e. 10ˆ3 for easier comparison, too.
l.2 I do no understand how you get a minimum volume of 160 m3. If you take a minimum area of 10 pixels with 30 m2 per pixel, you get a height of 0.53 m. This does not match your vertical change threshold of 5 m. So, I guess the comma is misplaced, it must be 1.6 10ˆ3 m and you were correct with "three orders of magnitude".
*Volume / (10x Pixel Area) = Height or 160/(10x3.75x 8) = 0.53*.
l.8-11 Delete the subdivision of the shallow sector it does not bring much to the comprehension of the calving distribution. Or just keep the rock part: M.

Table 1: Could this table be combined with Figure 6a and 6b by placing a text in the corner or a horizontal boxplot at the top? Rarely calving distribution are presented as a table, making it difficult to compare with other studies.

l.19-21 Develop the detail of your computation (numbers?) and how you obtain 25% in the text and on Figure 5d.

4.3 Calving statistics
l.23 Describe first what you want to achieve, meaning model the calving distribution with non-linear fitting models to assess whether you observe a "self organised critical system".
l.27-31 It is not clear to me what is the basis for selecting a log-normal against a power-law in both sectors as the results of the maximum likelihood (and visual inspection) show that the fitting models are as good and with similar parameters.

4.4 Pressure sensor records
In order to find the missing component presented in the discussion, it would really help to derive a rough calving catalogue based on a peak detector or even manual picking and neglecting other sources of wave oscillations such as iceberg rotation.

5. Discussion
l.14-15 "[...] no clear temporal pattern of tidal or diurnal recurrence [...]" comes to me as a surprise as it is not presented in the results (but should be, including Figure 9).

5.1 Relation to ice flux and other processes
l.16 Which "other processes"? Be specific. Here you want to close the "ice flux budget at the calving front" or find the "Missing volume in the deep sector"
l.17-23 Your simplification to compute the ice flux is fair but neglects variations in ice thickness and important processes such as submarine melt. It is thus not convincing that the total calving volume matches the computed flux. Try to obtain the ice flux by integrating the glacier velocity, height for aerial calving (ice thickness would be better assuming a certain bed topography) and the glacier discrete width (for each space unit) maybe even upstream of the front assuming constant front

position. Also, rephrase this part by first stating what are your assumptions and the missing elements.

l.1-6 I am confused here as you seem to compute the aerial ice flux using the ice height above water (150 m) and thus the missing aerial volume in the deep sector cannot be directly caused by oceanic melting, but indirect effect of the undercutting and thereby lower stress threshold for breakoff. Compare the volume you detect with the aerial ice flux in the deep sector with an ice height of 50 m. The estimated volume from the ice flux is then threefold lower than your previous estimate and is closer to your TRI calving volume estimate.

l.3-4 Before neglecting the role of filtered calving events in explaining the missing volume, could you check that the number of filtered calving events is proportionally the same in the shallow and deep sector assuming a homogeneous noise along the front? The effect of oceanic ice melt and undercutting in the deep sector may cause smaller blocks to fall at lower stress threshold than in the shallow part. This is coherent with your observation that few large calving events occur in the deep part.

l.8-15 Good discussion and comparison. I would just add the year when the oceanic melt was obtained as it depends on warm Atlantic water intrusion that has reached a peak in 2007 and has weakened since (hence the glacier advances in the region).

l.15 "the contact area [...] is much smaller" by how much?

l.20 Indicate that the 75% mass removal in the deep sector occurs only over two third of the calving front, showing a greater efficiency of melting in the submarine part of the front.

5.2 Influence from cliff height and shape

Overall, the discussion is good, but there is no discussion on the effect of undercutting on stress regime (see comments in 5.1).

l.22 Alternative title: "Effect of steeper and higher ice cliff " or "Role of front geometry on stress regime"

l.1 "decreasing water level" do you mean tides or specify what causes water level to decrease in crevasses.

l.3-7 Nice interpretation that could also be applied to the deep sector. You can verify your hypothesis by comparing front positions and the glacier retreat seen in Figure S.1 may be significant.

5.3 Calving statistics

Although the discussion is well written and nicely supported with recent studies, the low *p* values of the likelihood test on Figure 6 show me that both a power law and log-normal can explain your distribution in both sectors. Thus reformulate your question here as the power law distribution can also be attributed to the shallow sector.

l.8 Alternative title: "Self-organised critical system vs less complex systems" or "calving distribution and driving mechanisms" or "calving distribution"

l.12-13 All cited studies estimate aerial calving and neglects subaqueous events. The main reason for a too steep power law curve may be that the study period is too short and there were too many (or not enough) calving events larger than $10^5$ m3 (or between $10^4$ and $10^5$ m3). This would also explain the misfit on Figure 6c.

l.16 "[...] instability with **many small** events [...] events of **greater** magnitude"

l.17-18 I do not understand the point of that sentence. Rephrase or delete.

5.4 Comparison with pressure sensor data

Discuss whether you can identify waves of rolling iceberg from those of calving events. Subaqueous events can often be confused with icebergs rolling close to the glacier front as we can often only identify them from the produced waves.

l.25 Alternative title: "Comparison with calving wave signal"

5.5 Relation to external forcing

The air humidity is not present here (section 2.3) and would be useful to assess potential precipitation periods and atmospheric disturbance on the radar signal. Precipitation tends to mess up surface melt and tidal signals seen in calving event occurrence.

The temporal resolution used in Figure 9 for your calving events may be too high to find a relation that often occurs on hourly time scale. On Figure 9d-e, resample your calving data and apply a cutoff for the two-three largest events around 0.2 in 9d and 10 in 9e. The legend can be plotted once in between the two panels.

l.12 Alternative title "Absence of meteorological and tide effect"

l.14-15 The sentence on surface melt is not coherent with the sentence before that finishes on tide effects.

6. Conclusion

Update the conclusion after answering my comments.

l.10 "We developed a novel **detection** method **based on TRI DEM differencing** to establish [...]"

**---- Figures ----**

Figure 3

Remove the velocity comparison as a more thorough comparison is achieved in Rohner et al, 2019 (refer to their Figures). Instead show here a plot of the mean velocity for the entire period and beside or in another figure plot a comparison of the TRI DEM and Arctic DEM including a distribution of their difference (or do you have the UAV derived DEM as well?). Eventually, in a new figure plot the DEM before and after one selected calving event, the subtraction of the DEMs, mask of the glacier front and the watershed result. Condense the caption: "Velocity field at the glacier front measured a) with the TRI on 19 August 2016 and b) with a UAV"

Figure 4

Plot the rough location and view angle of the picture

Figure 5

*Panel a*

Could you use an image taken from the TRI position or the radar image in Figure 4 cropped and rotated to fit the format here and with the line you use to stack your data in c)? The current image is confusing as it is in a different orientation than the TRI (taken from a hike to front, I guess) and saturated (I cannot see the front texture).

*Panel b-d*

 extend the plot to the end of the right plot margin. If this was for the legend in b, change min max lines to a polygon (shadow) and place the legend horizontal at the bottom of the panel such as --- Elevation    ---- Mean velocity    |grey box| Min/max velocity. Indicate the location of the sectors above panel b) and delete them form panel c).

*Panel c*

Please use a linear colour scale. Your current palette highlights mostly areas with no calving volume change and the yellow parts of your spectrum (7000 m3) and a bit the red ones. Choose a linear colour scale from light yellow to red or blue. Write "data gap" between 22 and 23 Aug.

*Panel d*

Delete the red dashed line representing 50% and 100% of the mean calving volume in the shallow sectors. I don't understand which message it is supposed to convey. Do you mean that 50% is equal to ~0.4 $10^6$ m$^3$ of calving volume?

Figure 9
Stretch the vertical axis for all panels to highlight the variations of your parameters. Shift the shortwave data so that we see that the curve goes to zero during the night (there must still be some light at this latitude mid-august?). Use the same temporal resolution for the Volume and number of events than the two first panel for instance hourly. Cutoff the extreme value on the 26 Aug. evening and write its value on the figure with an arrow pointing to the top. (also see comments *page 16*)

---

## Author Comment (AC1) · 30 Sep 2019

General reply

We thank both reviewers, Surui Xie and Pierre Marie Lefeuvre, for reading through the manuscript and their critical comments and helpful ideas, and suggestions. We appreciate the invested time for the feedback. The reviewer's major concerns mainly referred to the issue of uncertainty in TRI derived elevation models, the ice flux budget estimation and the statistical analysis. We are thankful for these comments and we think we should be able to address those comments in our revisions and that they will considerably improve the manuscript. Here we reply to the more substantial concerns

raised by the reviewers and present our ideas for revisions. The main changes we plan are 1) a detailed error analysis of the stable terrain between the TRI derived DEMs 2) to compare the ArcticDEM with the TRI derived DEMs 3) and a more detailed analysis of the ice flux as comparison to the calculated calving volumes.

Most of the minor comments consider language or detailed content or are specific examples of the major concerns. All these more minor corrections will be addressed in the revised manuscript. We therefore mostly refer to the major comments here. We appreciate the opinion of the editor on the suggested revisions.

For our reply to the major concerns please look at the supplement pdf.

Please also note the supplement to this comment:
https://www.the-cryosphere-discuss.net/tc-2019-102/tc-2019-102-AC1-supplement.pdf

―――――――――――――――――

**Supplement:**

**Calving event size measurements and statistics of Eqip Sermia, Greenland, from terrestrial radar interferometry**

Andrea Walter, Martin P. Lüthi, Andreas Vieli

Submitted for review to The Cryosphere in May 2019

**Reply to reviewer's comments**

**General reply**

We thank both reviewers, Surui Xie and Pierre Marie Lefeuvre, for reading through the manuscript and their critical comments and helpful ideas, and suggestions. We appreciate the invested time for the feedback. The reviewer's major concerns mainly referred to the issue of uncertainty in TRI derived elevation models, the ice flux budget estimation and the statistical analysis. We are thankful for these comments and we think we should be able to address those comments in our revisions and that they will considerably improve the manuscript. Here we reply to the more substantial concerns raised by the reviewers and present our ideas for revisions. The main changes we plan are

- a detailed error analysis of the stable terrain between the TRI derived DEMs
- to compare the ArcticDEM with the TRI derived DEMs.
- and a more detailed analysis of the ice flux as comparison to the calculated calving volumes.

Most of the minor comments consider language or detailed content or are specific examples of the major concerns. All these more minor corrections will be addressed in the revised manuscript. We therefore mostly refer to the major comments here. We appreciate the opinion of the editor on the suggested revisions.

**Reply to major concerns Referee 1**

**R1:** Uncertainty in TRI derived elevation models needs better assessment. The authors randomly choose 30 DEMs and computed the variability (its definition needs to be provided in the manuscript) as a measure of the precision.
Although the mean variability is 1 m, but the maximum variability is 5 m. Therefore, the DEMs are likely to have an uncertainty of ~1 m to several meters. Although a threshold of 5m elevation decrease between adjacent DEMs is used to determine calving events, but note that even among only 30 DEMs there is a variability of 5 m between two DEMs. The calving statistics of this manuscript come from hundreds of DEMs, several large random errors ($2\sigma$ or above) or outliers can significantly change the results. I suggest the authors to provide more details on error analysis.

We agree that the uncertainty analysis was presented too vaguely. We will add a detailed section on error analysis, where we will present the variabilities on stable terrain over space and time. The test area on stable terrain will be indicated in Figure 1. This more detailed error analysis does however not substantially change our results and main findings stated in the paper. However, it will strengthen our methods and findings.

**R1:** Based on the calving data derived from TRI elevations, the authors concluded that surface calving is more frequent in the shallow water sectors, and the sizes are generally larger. This seems apparent if just looking at Figure 5c. However, due to lack of rigorous uncertainty analysis, I think this conclusion is hasty and may be flawed. In general, noise in TRI measurements increases with distance, and can increase rapidly at a distance of 4-6 km. Glacier front on the northwestern section (shallow water sectors in this manuscript) is further from the radar than the southeastern section (deep water sector), thus radar data on the northwestern section of the glacier should be noisier if all other conditions are similar. Besides, the northwestern section of the glacier front is crevassed heavier than the southeastern section (Figure 1), and elevation changes rapidly (inclined at a slope of 50 degrees according to the authors), both are more likely to induce phase unwrapping errors than a flat and less crevassed surface. These (i.e., increased noise with distance, phase unwrapping problems) could be some of the reasons why the identified calving volumes are more variable and the cumulative calving volumes are larger along the SL/SM/SR/M sectors. In Figure 5c, timing and sizes of calving events at different distances look random, but considering the characteristics of radar noise, it is important to examine if the observed pattern is due to noise or unwrapping errors. Here I suggest one possible method to test how much noise affected the distribution pattern in calving events: using the same analysis approach as presented in the manuscript, but apart from calculating calving volume based on pixels whose elevation decreased by >5 m, the authors can also calculate "increased volumes" by pixels whose elevations increased by >5 m. If a similar distribution pattern as in Figure 5c is seen, then the derived "calving volumes" are likely disturbed. The authors can probably add a plot of such "increased volumes" to the negative side of y-axis in Figure 5d (can used light blue color if the authors don't want it be distracting). A comparison figure of "detected increasing volume" similar as Figure 5c can also add important information to the manuscript, and it can go to the supplement if the authors would like to save space in the main manuscript.

We realized that in our paper the correction process to minimize errors with distance is not described. However, we used a correction factor to correct for systematic error sources. Those error sources can be caused by errors in the reference heights and instrumental geometry, baseline errors and errors caused by a not perfectly vertical mounting of the three antennas (Strozzi et al., 2012). We did that by comparing the calculated DEMs with the Arctic DEM and choosing control points on stable terrain at different distances from the radar. With the used correction factor we can minimize uncertainty in the height estimates. We will add details on computing the correction factor in the revised manuscript.
Considering the geometry of the calving front we agree that the shallow sector is more likely to induce errors than the deep part with the less steep front geometry. We will have a closer look at this by also investigating the uncertainty due to slope differences on stable terrain. For this we include an example of the analysis calculated on stable terrain with different slopes. We thank referee 2 for his suggestion of doing the whole analysis for positive height changes. We will have a look at that and include such a figure in our revised manuscript (most likely in appendix).

**R1:** According to the authors, there was very little surface calving observed by TRI at sector D (the deep water sector), and mass loss due to subaqueous calving is dominant (50% or more, depends on the rate of oceanic melt) here. Limited evidence of subaqueous calving was shown in the manuscript. Even if substantial subaqueous calving events occurred and contributed significantly to the mass loss at the deep water sector, the manuscript failed to explain where the mass goes. I also think it is not adequate to simply assume that subaqueous calving is independent of TRI observed surface calving. If subaqueous calving would not

cause surface elevation change by following the authors' logic, then what was there to fill the space left by the "subaqueous calving"? Besides, if TRI observed little calving at sector D, then glacier front at this section should advance, especially at the high velocity area. Speed in the middle of this sector is 16 m/day, in ~7.65 days, ice front here can advance over 100 meters, much larger than the resolution of either Landsat/Sentinel satellites or TRI images so should be detectable. However, Supplementary Figure S1 rejected this.

We agree that the explanation of the processes happening at the deep sector was not complete. We cannot see subaqueous calving events with the TRI data but we can add an image of a subaqueous calving event from the time-lapse camera installed in 2018. Unfortunately, no time-lapse camera was installed in 2016. As the flow field and surface slope further upstream is homogenous (smooth across-flow profile velocity profile) and thereby does not indicate substantial differences in ice thickness we assume that the ice thickness downstream (towards the front) is similar for the shallow and the deep sector also at the calving front. We argue that if this would not be the case, we would see it in the surface structure of the glacier by specific crevasses or flow velocity variations. Thus, if the ice thickness is similar, at the deep sector about 2/3 of the ice area are below the waterline. The assumption that at least 66 % of the mass loss are underwater is therefore justified. The remaining 10 % might be calved off above the water line through small calving events, which were filtered out during the analysis. Those small events can be caused by undercutting of the calving front due to ocean melt and calving below the waterline. Of course these are only relatively rough estimates which we will clarify in the text. We will rewrite this section and explain this in more detail.

**R1:** The manuscript did not explain the method they used to choose the study area for calving detection well. Although on page 6 the authors mentioned that they applied a mask with ~150 m wide across the glacier front, however, the glacier front was constantly moving, so a Lagrangian frame should be used. Whatever the reference frame was, according to the methods presented by the authors, areas with calving event detected (Figure 4) over the center of sector D should have the largest along-flow direction width. This is because glacier front at this location should advance (also see comment above), thus the test area should move. Whereas Figure 4 shows a different pattern.

In the revised manuscript we will add a figure showing the used front position mask as well as the front position over the observation period. The front position advanced and retreated only marginally but was always well within the mask. Thus we decided to use a simple constant mask. The fact that the centre of sector D does not have the largest along-flow direction width we explain with many small events, which might be triggered by undercutting of the calving front due to underwater calving and ocean melt. We will rewrite the corresponding section in the revised manuscript.

**Reply to major concerns Referee 2**

**R2:** DEM derivation of the glacier front from TRI
The critical part of the paper is the derivation of digital elevation models of the glacier front from terrestrial radar interferometer as developed by Strozzi et al. (2012). However, this method is known to be uncertain, although the large glacier size should help having a greater signal to noise ratio. I think that it is important to extend the paragraph on the error analysis and dedicate a specific figure with a map of the derived DEM(s), statistical distribution of the error in the discussed stable terrain. Assess the uncertainty of the glacier part with the UAV derived DEM too by replacing Figure 3 as the velocity comparison is done in Rohner et al, 2019. I would like to have a Figure showing a study case of the detection and watershed

algorithm to assess issues with signal to noise ratio and uncertainty in radar geometry or cartesian coordinates (in the main text or supplementary material).

We agree with the concern of referee 2 that the error analysis needs more elaboration. We will add a section, where we will investigate the error on stable terrain in more detail (see comments referee 1). We will compare a stacked TRI DEM with the Arctic DEM and investigate the variations between the TRI DEMs itself on the stable terrain over space and time.
A comparison with the UAV derived DEM to investigate the uncertainty of the glacier part is currently not possible as we do not have a georeferenced UAV DEM. To do this comparison we would need to georeference it and do a detailed error analysis, which is not part of this paper. Additionally, as we do not have ground control points on the glacier the UAV DEM might be warped in the centre of the glacier.
Figure 3 is not meant as a velocity comparison, but the UAV velocity is rather a completion of the TRI-derived velocity data to illustrate the flow field. However, we will rephrase the section to make this clearer.
We will add a figure showing a study case of the calving event detection before section 4.1. With this we will show how we can reduce errors by stacking and by using thresholds. An example of the detection of a calving event will be shown in the main text while an example of the stable terrain will be included in the supplementary material.

**R2:** Issues in determining best fit models for calving distribution
As seen on Figure 6c and 6d, it is not possible to distinguish between a power law and a log normal models as indicated by a low loglikelihood ratio R between the two distributions and a poor significance value, $p>0.1$. The only evaluation possible of the power law is a comparison with other heavy-tailed distributions. The conclusion is that the shallow and deep part does not exhibit a transition in distribution from power law to log normal as they cannot be statistically differentiated from each other. Discuss instead whether the distribution over such a short period is representative.

We agree that the significance value is poor and thus it is not really possible to distinguish between a power law and a log normal model. We will rephrase the section to make it clearer that the distinction is not significant but that there might be a difference. We also agree that the period is rather short to have a representative distribution and we will add a discussion section about that. Since we observed a big number of calving events we are convinced that a statistical analysis is meaningful and legitimate.

**R2:** *Ice flux budget: bed topography and missing component*
The paper bases its analysis on the depth of the fjord but no bed data is provided to support this description (just observations of surfacing rocks). Please use the BedMachine v3 to at least provide an idea of the fjord depth in front of the glacier to the reader. The shallow part may be constituted of two bed pinning points beside a deep valley. Furthermore, simplify the subdivision of the shallow part to only the shallow part regrouping SL, SM and SR.
The simple ice flux calculation holds some caveats when identifying a missing volume due to the uncertain fluxgates and filtering of small events. The distribution of these small events may be related to calving mechanisms and ocean melting (undercutting). A more realistic flux can be derived by integrating the ice flux with your surface elevation and velocity data. See my minor comments to improve the understanding of section 5.1.

We will add the data of BedMachine V3 to the supplement of the revised manuscript to give an idea about the fjord topography. However, as no direct measurements are available directly

at the calving front, the data of BedMachine V3 at the calving front should be considered with care. At the calving front and close to it the velocity is influenced by additional processes and the estimation of the ice thickness by inferring the surface elevation and velocity data cannot reproduce the bed topography correctly.

We will add a more detailed analysis of the ice flux to section 5.1. Therefore, we calculate the potential ice flux per bin in Figure 5d with our data on velocity and surface height per bin. We will plot it as additional bars in Figure 5d. To include the calving below waterline and the ocean melt we will assume a total ice thickness of 150 m and calculate the flux with this value, the surface height and the velocity per bin. This analysis will strengthen our findings regarding the importance of ocean melt and calving below the water line at the shallow sector. However, note that these flux estimates are only rough estimates to analyse the rough shares of different calving processes and not exact values of different calving fluxes.

We will keep our subdivision of the calving front as we think it is needed for the interpretation. The three parts of the shallow sector show different characteristics likely due to the bed topography and thus they cannot be seen as one homogenous sector.

**R2:** Better integration of calving wave dataset
The paper should integrate better the ocean wave data as an alternative dataset of calving events (including subaqueous ones?), explain this discrepancy and discuss other potential sources such as iceberg rolling. This better integration of the wave amplitude dataset with the TRI detected calving events would strengthen the discussion and conclusion of the paper.

We will add a peak detection analysis of the calving wave dataset for a comparison with the TRI detected calving events. However, in this paper we want to focus on the TRI dataset and the established methods. A more detailed analysis of the calving wave dataset is the topic of a follow-up paper.

**Reply to minor comments**

Page 6, lines 15-17:

R2: Is the elevation difference just a shift in absolute elevation explained by a difference in geoid or geo-referencing problems of the Arctic DEM or your DEM? Also co-registering the two DEMs before differencing is useful to assess systematic errors outside obvious artefacts.

R1: Are these differences RMS difference, Mean difference, or other types? These can be quite different because the difference between TRI-DEM and Arctic DEM can be systematically and/or randomly. Please clarify.

We will add a more complete comparison in the revised manuscript on the comparison of the Arctic DEM and our DEMs investigating the variabilities on stable terrain in more detail. The two DEMs were co-registered before comparing them. The elevation difference is likely not just a shift, it also depends on the slope of the terrain. We will outline the test area in Figure 1 or in an additional Figure and better explain the values in the extended error analysis section.

Page 6, lines18-19:

R2: Please provide a sentence about precision change over time on stable terrain and also ice. You could plot this variability on stable terrain and some upper part of the glacier for the

single DEMs and the stacked ones to appreciate the effect of atmospheric disturbances and the improvements from stacking. You could use this variation to provide first order error bars for your volume estimates on Figure 6. In the discussion could you compare your precision to what other studies found.

We thank referee 2 for the suggestion, we will add a section on the variability on stable terrain and use it as error estimates in the further analysis. However, as the glacier area measured with the TRI is highly crevassed and fast flowing, we do not think that an analysis of the variability there is meaningful.

Page 6, line 24:

R2: "When applying [...] are removed". The noise observed on stable terrain is not removed, but the signal to noise ratio is higher for the filtered events. Moreover, quantify the number or area of excluded events or give a percentage.

We will implement this suggestion and quantify the number and percentage of excluded events during the filtering process to assess also the uncertainty of the used calving detection method.

Page 6, lines 20-24:

R2: How do you deal with the zero elevation or water elevation when calving occurs along the entire ice column (i.e. column collapse)? Parts of the DEM covering the sea may have Not A Number values or problems with icebergs?

The water elevation is set to 0 also where there is Not A Number values. Thus, it does not influence the calculation if calving occurs along the entire column. The calving at Eqip Sermia happens mostly through ice avalanches, not resulting in big icebergs. Bigger icebergs remove the ice melange in front of the glacier, which results in a loss of coherence. Due to this loss of coherence the area including the iceberg is not included in the analysis. Icebergs further away are not included anymore as they are not within the glacier front mask.

Page 6, lines 23-24:

R1: Does the "10 pixels" mean "10-adjacent pixels"? I feel the two numbers "10 pixels" and "3 pixels" are confusing: Does noise needs to fulfill both "area<10 pixels" AND "width<3 pixels"? If so, how about a block with 3×3 (9 pixels, each pixel shown by an "O" below) shape, or a 2×8 shape? Are these considered calving events if all pixels have elevation decreases more than 5 m?
Because many of the identified calving events are quite small, and shapes of these blocks may not be regular, I think it is important to clarify these settings here.

We agree with referee 1 that the settings for the thresholds need to be clarified more to avoid confusion. We will add some examples in the supplement showing the used criteria's. Calving events need to fulfil both conditions, a size of 10 pixels and a width of 3 pixels. For the width it is enough if only one line within the calving event has 3 pixels. All the pixels need to have an elevation decrease of more than 5 m, except if some pixels with a lower decrease are

surrounded by pixels with a decrease of more than 5 m. Then those pixels are included in the calving event.

Page 7, lines 26-28:

R1: I found that the number of total identified calving events is smaller than the sum of identified calving events in the shallow sector and the deep sector (1681 < 1403+289). Did I miss anything? Please also check numbers in Table 1, many of them are not consistent.

The number of total identified calving events is smaller than the sum of identified calving events in the shallow and the deep sectors, because 11 events happened to be located on the boarder of the two sectors. Those events were counted only once for the total number but for both the number of the shallow and the number of the deep sector.

Page 7, line 29:

R2: Add the number of filtered/removed events for each sector as the filter may affect the number of calving events. I have a hunch that the deeper sector may have more small events, likely filtered out, due to the effect of higher submarine frontal ablation.

We will add the number and percentage of filtered events to see if the deeper sector has more small events. However, we will not change the filter threshold as for smaller thresholds it becomes hard to distinguish between noise and calving events.

Page 8, line 2:

R1: Please ensure the minimum size of identified calving block fulfill the threshold defined for calving events (based on lines 21-24, page 6, I calculated a minimum volume 5×10×3.75×3.75=703 m3. Or did I misunderstand the "resolution"? — I picked it from line 13, page 6. Using the radar pixel specified in line 6 page 8 the minimum volume of identified block is even larger, i.e., 1500 m3). If this paper aims to do statistics of calving event sizes, please ensure that the statistics are correct.

R2: I do no understand how you get a minimum volume of 160 m3. If you take a minimum area of 10 pixels with 30 m2 per pixel, you get a height of 0.53 m. This does not match your vertical change threshold of 5 m. So, I guess the comma is misplaced, it must be 1.6 10ˆ3 m and you were correct with "three orders of magnitude".
Volume / (10x Pixel Area) = Height or 160/(10x3.75x 8) = 0.53.

We thank the referees for spotting this. We realized that some of the events were considered as calving events even if they are collapsing seracs upstream the glacier. They were included as they were located exactly on the border of our glacier front mask. Those events will be excluded in the revised manuscript. The removing of those small events might influence or calving size distribution but all the other results will not be effected.
The algorithm takes also events into account where pixels of less than 5 m decrease are surrounded by pixels of more than 5 m decrease. Thus the events can be smaller than the calculated minimum size of a calving event.

Page 8:

R1: I think TRI-derived DEMs in this paper are very important data, however, there was no figure showing the DEM, neither in the main paper nor in the supplement. Please consider to include a TRI-derived elevation map in the manuscript. Maybe add a panel or two in Figure 3?

R2: The key result of the paper is the TRI-derived DEMs but the velocity (4.1) is presented first instead.

R2: Add text and figure(s) specifically on the generated DEMs and signal improvement by stacking before section 4.2 on calving detection results or a comparison with the UAV DEM.

We will add a section before 4.1 to show a generated DEMs as well as the signal improvement due to stacking and using thresholds (see comments above).

Page 10, Figure 5:

R1: I like this figure! But I also have a few questions and suggestions. First, color changes from dark blue to light blue in Figure 5c is distracting, I suggest to mask out periods without calving using white or grey. In this way, calving characteristics will be more accessible. Consequently, the minimum value of the color bar can be changed to the smallest volume detected based on the settings (lines 20-25 on page 6).

R2: Panel c: Please use a linear colour scale. Your current palette highlights mostly areas with no calving volume change and the yellow parts of your spectrum (7000 m3) and a bit the red ones. Choose a linear colour scale from light yellow to red or blue. Write "data gap" between 22 and 23 Aug.

We agree that the used color changes can be distracting. In the revised manuscript we will change the colors and use grey for the periods without calving while the data gap will be white.

R1: Second, I am confused by the right axis of Figure 5d. Can the authors elaborate on this?

R2: Panel d: Delete the red dashed line representing 50% and 100% of the mean calving volume in the shallow sectors. I don't understand which message it is supposed to convey. Do you mean that 50% is equal to ~0.4 106 m3 of calving volume?

We will better explain what the red dashed line means in the revised manuscript. The right axis shows the percentage of detected volume if the mean total calving volume of all bins in the shallow sector is assumed to be 100 %.

R1: Third, I could hardly read the superscripts in y-axis label of Figure 5d due to low resolution. Based on the manuscript I guess it was "106 m3" in the bracket, is this correct? Please increase the figure resolution. Also, maybe add "Cumulative" to the y-axis label of Figure 5d to distinguish from the color bar label in Figure 5c. Fourth, I found that the further analysis separates these sectors, so it is necessary to show the exact along-distance ranges of different sectors. Maybe use vertical bars to mark the boundaries of different sectors? These bars can go between the annotations in Figure 5c, such as " | SL | SM | SR | M | D |". Last,

would it be possible to add a narrow column on the right of Figure 5c and show total calving volume along the entire calving front in color? Such a plot may provide useful information on calving volume changes with time.

We will increase the figure resolution and add "cumulative" to the y-axis label of Figure 5d. We will not reduce the number of sectors (see comments above) but will use vertical bars to mark their boundaries. We will try to add such a column on the right of Figure 5c but we are not sure if the addition of more information in this already dense plot will not be distracting. Also this information is already provided in Figure 9.

Page 11, line 15-16:

R1: The "observable cluster of calving" is hard to tell from Figure 5c. Yes there are some big events, but since this manuscript does statics, in the sense of statics, do these relatively big events really clustered? Need more elaboration.

We think the observation time is too short to do a more statistical elaboration on that. We will delete this sentence.

Page 11, lines 19-21:

R1: More detail of deriving the 25% needs to be provided. Is it an appropriate assumption of a constant front position? Here ice can move up to 16 m/day (Figure 3). And the assumption of a constant mass flux over the front also needs to be justified.

R2: Develop the detail of your computation (numbers?) and how you obtain 25% in the text and on Figure 5d.

We will explain the deriving of the 25% better in the text and also in Figure 5d. The right axis shows the percentage of detected volume if the mean total calving volume of all bins in the shallow sector is assumed to be 100 %. If we calculate the mean total calving volume of all bins in the deep sector it results in less than 25% of the calculated mean total calving volume of the bins in the shallow sector.
We will also add a figure showing the used glacier front mask and the variation of the front over time to show that the changes are only marginally which makes it justifiable to assume a constant front position at least during our observation period. We agree that the assumption of a constant mass flux over the front is just an estimation. However, the velocity of the ice further upstream the glacier is homogenous. We argue that if the flux would not be constant anymore over the whole front we should be able to see signs of that in the surface structure of the glacier (crevasses). In the revised manuscript we will reformulate that section to make it clearer that this is an assumption and we do not have measurements to prove this assumption. However, due to bathymetric data we know that the fjord is deeper at the deep section (see chapter 2.1), which confirms that there the ice area below the waterline is larger.

Page 13, Figure 7:

R1: This figure is nice. I have a suggestion of another figure: plot one similar figure as Figure 7, but use the entire period (similar as Figure S2), and then plot calving volume of the entire

glacier front (y-axis) versus time (x-axis) for comparison. This would help further discussion of potential relationship between calving and water level variation.

We will add a panel to figure 9 with the water level variation and the detected peaks as comparison to the TRI detected calving events.

Page 13, line 18:

R1: I think some of these values can be calculated from the data. If the authors use values derived from real data, the further analysis would sound more reasonable. Also, maybe use "a front thickness" instead of "a front height" to avoid confusion?

R2: Your simplification to compute the ice flux is fair but neglects variations in ice thickness and important processes such as submarine melt. It is thus not convincing that the total calving volume matches the computed flux. Try to obtain the ice flux by integrating the glacier velocity, height for aerial calving (ice thickness would be better assuming a certain bed topography) and the glacier discrete width (for each space unit) maybe even upstream of the front assuming constant front position. Also, rephrase this part by first stating what are your assumptions and the missing elements.

We agree that the computation of the ice flux neglects some important parameters. We will therefore recalculate the ice flux with the available data of velocity and height (see major concerns referee 2).

Page 14, lines 1-6, 9:

R2: I am confused here as you seem to compute the aerial ice flux using the ice height above water (150 m) and thus the missing aerial volume in the deep sector cannot be directly caused by oceanic melting, but indirect effect of the undercutting and thereby lower stress threshold for breakoff. Compare the volume you detect with the aerial ice flux in the deep sector with an ice height of 50 m. The estimated volume from the ice flux is then threefold lower than your previous estimate and is closer to your TRI calving volume estimate.

R1: Why assuming an ice thickness of 100 m below the water line? Too many assumptions could result in significant bias. Since the authors have data of surface elevations (shown in Figure 5b), and have assumed a front thickness of 150 m (line 18 on page13, although I suggested to use real data instead of assumption), ice thickness below water line can be estimated.

We agree that for a comparison of the TRI calving volume and an estimated flux at the deep sector we need to do the calculations also with an ice height of 50 m. We will include that in the revised manuscript (see major concerns referee 2). If we assume the same ice front thickness for the deep part than we see for the shallow part using an ice thickness of 150 m is reasonable. This is the ice thickness of the shallow front where the water depth is about 0-20 m. In the revised manuscript we will calculate the flux above the waterline with the available velocity and surface height data and we will compute the total flux using the velocity, the surface height data and assuming an ice thickness of 150 m.

Page 14, lines 16-18:

R1: If subaqueous calving cannot be detected with the TRI, how could it be detected by visual observations and time-lapse imagery? More details are needed. Maybe show some examples of images taken by the time-lapse camera.

With the presented method of the TRI-DEM differencing it is only possible to detect height changes at the surface. In 2018 a time-lapse camera took images every 10 s. With this high temporal resolution it becomes possible to see icebergs coming up from under water. We will add a picture of such an event in the supplement.

Page 14, lines 25-26:

R1: If ice cliff at the shallow sector can have larger but stable height, then why do calving events occur so frequently? Although ice here is thicker but calving should be less frequent or no calving at all because ice cliff can be stable (lines 25-26). May the authors were referring to the potential of calving so"a thick cliff CAN release larger ice volumes", but please note that here calving is quite frequent (also related to how "stable" was defined in this manuscript), while previous figures (e.g., Figure 5c) show that the shallow sectors calved more frequently at the surface. Even add subaquatic calving to the deep water sector, calving at the shallow sectors will still be more frequent than the deep water sector because the authors assumed the overall mass loss are similar in different sectors, plus the deep water sector has lost more mass due to melt.

We agree that this section leads to some confusion. We will rewrite it to make it clearer that calving is more frequent at the shallow sector than at the deep sector. However, the calving events are generally larger at the shallow sector as compared to the deep sector, which can be explained with the inclined front.

Page 15, lines 4-7:

R1: Would enhanced submarine melt cause surface elevation decrease because ice becomes thinner? If it won't lead to surface elevation decrease, then what was there to support the upper part of the glacier? Would the empty chambers cause instability and calving? On the other hand, if it will lead to surface elevation decrease, then the TRI might be able to see the decrease. Please clarify.

With the TRI we do not see a surface elevation decrease. Due to undercutting at the deep sector caused by mass loss below the waterline small calving events, which were filtered out, might be triggered.

Page 15, lines 12-13:

R2: All cited studies estimate aerial calving and neglects subaqueous events. The main reason for a too steep power law curve may be that the study period is too short and there were too many (or not enough) calving events larger than 10^5 m3 (or between 10^4 and 10^5 m3). This would also explain the misfit on Figure 6c.

We agree that this could also be a possible explanation and we will add it to the revised manuscript.

Page 15, line 30:

R1: Please explain what is "big up-floating icebergs". If the icebergs are big, they might appear on the TRI amplitude images. An example image would be helpful.

Those icebergs are visible on the TRI amplitude images, however it is not possible to see the upcoming of the icebergs on the image as the temporal resolution is not high enough. We will add a time-lapse picture as example.

Page 16, lines 4-6:

R1: Yes I agree that pressure sensor observations could be used to derive calving events, but challenges remain. One challenge is that subglacial hydrological events may cause similar signal as what has described as subaqueous calving in this manuscript. Need justification.

R2: Discuss whether you can identify waves of rolling iceberg from those of calving events. Subaqueous events can often be confused with icebergs rolling close to the glacier front as we can often only identify them from the produced waves.

We agree that a more detailed study of the pressure sensor data is needed. Also turning over of icebergs can generate waves. However, in this study we focus on one method using the TRI dataset and the pressure sensor data will be investigated accurately in another study.

R2: In order to find the missing component presented in the discussion, it would really help to derive a rough calving catalogue based on a peak detector or even manual picking and neglecting other sources of wave oscillations such as iceberg rotation.

In the revised manuscript we will add a calving catalogue based on peak detection for comparison with the TRI-derived calving events.

R2: The air humidity is not present here (section 2.3) and would be useful to assess potential precipitation periods and atmospheric disturbance on the radar signal. Precipitation tends to mess up surface melt and tidal signals seen in calving event occurrence.

We thank referee 2 for this interesting suggestion. We will have a look at the air humidity but only add a subplot if we see that it is informative. Otherwise we will discuss it in the text.

R2: The temporal resolution used in Figure 9 for your calving events may be too high to find a relation that often occurs on hourly time scale. On Figure 9d-e, resample your calving data and apply a cutoff for the two-three largest events around 0.2 in 9d and 10 in 9e. The legend can be plotted once in between the two panels.

We thank referee 2 for this suggestion. We will have a look at this but only change it if we see that it is important. Additionally, we agree that a cut-off for the largest events might help to increase the readability of the graph.

Figure 9:

R2: Stretch the vertical axis for all panels to highlight the variations of your parameters. Shift the shortwave data so that we see that the curve goes to zero during the night (there must still be some light at this latitude mid-august?). Use the same temporal resolution for the Volume and number of events than the two first panel for instance hourly. Cutoff the extreme value on the 26 Aug. evening and write its value on the figure with an arrow pointing to the top. (also see comments page 16)

We thank referee 2 for this suggestions, we will adapt figure 9 by stretching the vertical axis and cut off the extreme value on the 26 August. However, we do not see why the volume and number of events should be plotted hourly, as we would rather loose information by doing this.

---

## Author Response (AR1)

**Calving event size measurements and statistics of Eqip Sermia, Greenland, from terrestrial radar interferometry**

Andrea Walter, Martin P. Lüthi, Andreas Vieli

5

Submitted for review to The Cryosphere in May 2019

**Reply to reviewer's comments**

10

**General reply**

We thank both reviewers, Surui Xie and Pierre-Marie Lefeuvre, for reading through the manuscript and their critical comments and helpful ideas, and suggestions. We appreciate the invested time for their feedback. The reviewer's major concerns mainly refer to the issue of uncertainty in TRI derived elevation models, the ice flux budget estimation and the statistical analysis. We are thankful for these comments and we addressed those comments in our revisions and think that they considerably improved our manuscript.

- 20 The main changes we undertook are:
  - using an additional threshold to reduce the distortion in the shallow sector.
  - a detailed error analysis of the stable terrain between the TRI derived DEMs.
  - a comparison of the ArcticDEM with the TRI derived DEMs.
  - a more detailed analysis of the ice flux as comparison to the calculated calving volumes.
- and all minor corrections and editing issues have been addressed.

**Reply to major concerns Referee 1**

30

Uncertainty in TRI derived elevation models needs better assessment. The authors randomly choose 30 DEMs and computed the variability (its definition needs to be provided in the manuscript) as a measure of the precision.

Although the mean variability is 1 m, but the maximum variability is 5 m. Therefore, the DEMs are likely to have an uncertainty of ~1 m to several meters. Although a threshold of 5m elevation decrease between adjacent DEMs is used to determine calving events, but note that even among only 30 DEMs there is a variability of 5 m between two DEMs. The calving statistics of this manuscript come from hundreds of DEMs, several large random errors ( $2\sigma$  or above) or outliers can significantly change the results. I suggest the authors to provide more details on error analysis.

40

We agree that the uncertainty analysis was presented too vaguely. We added section 4.1 on the resulting DEMs and error analysis. Additionally, we added Figures S2, S3 and S4 showing the variabilities on stable terrain over space and time. The test area on stable terrain is indicated now in Figure 3 as a yellow box. It is important to note, that this more detailed error analysis does however not

5 substantially change our results and main findings stated in the paper.

Based on the calving data derived from TRI elevations, the authors concluded that surface calving is more frequent in the shallow water sectors, and the sizes are generally larger. This seems apparent if just looking at Figure 5c. However, due to lack of rigorous uncertainty analysis, I think this conclusion is

- 10 hasty and may be flawed. In general, noise in TRI measurements increases with distance, and can increase rapidly at a distance of 4-6 km. Glacier front on the northwestern section (shallow water sectors in this manuscript) is further from the radar than the southeastern section (deep water sector), thus radar data on the northwestern section of the glacier should be noisier if all other conditions are similar.
- 15 Besides, the northwestern section of the glacier front is crevassed heavier than the southeastern section (Figure 1), and elevation changes rapidly (inclined at a slope of 50 degrees according to the authors), both are more likely to induce phase unwrapping errors than a flat and less crevassed surface. These (i.e., increased noise with distance, phase unwrapping problems) could be some of the reasons why the identified calving volumes are more variable and the cumulative calving
- volumes are larger along the SL/SM/SR/M sectors. In Figure 5c, timing and sizes of calving events at different distances look random, but considering the characteristics of radar noise, it is important to examine if the observed pattern is due to noise or unwrapping errors. Here I suggest one possible method to test how much noise affected the distribution pattern in calving events: using the same analysis approach as presented in the manuscript, but apart from calculating calving volume based on
- 25 pixels whose elevation decreased by >5 m, the authors can also calculate "increased volumes" by pixels whose elevations increased by >5 m. If a similar distribution pattern as in Figure 5c is seen, then the derived "calving volumes" are likely disturbed. The authors can probably add a plot of such "increased volumes" to the negative side of y-axis in

Figure 5d (can used light blue color if the authors don't want it be distracting). A comparison figure of "detected increasing values," similar on Figure 5a can also add important information to the

30 "detected increasing volume" similar as Figure 5c can also add important information to the manuscript, and it can go to the supplement if the authors would like to save space in the main manuscript.

We realized that in our paper the correction process to minimize errors with distance is not described.

- 35 However, we used a correction factor to correct for systematic error sources. Those error sources can be caused by errors in the reference heights and instrumental geometry, baseline errors and errors caused by a not perfectly vertical mounting of the three antennas (Strozzi et al., 2012). We did that by comparing the calculated DEMs with the Arctic DEM and choosing control points on stable terrain at different distances from the radar. With the used correction factor we can minimize uncertainty in the
- 40 height estimates. We added this information to the DEM generation methods (section 3.1). Considering the geometry of the calving front we agree that the shallow sector is more likely to induce errors than the deep part with the less steep front geometry. We thank referee 2 for his suggestion of doing the whole analysis for positive height changes. By investigating the positive changes we realized that the distortions are indeed higher for the shallow sector. Thus we redid the whole analysis and
- 45 added an additional condition in the watershed segmentation algorithm. As the noise has mostly an irregular shape, while the calving events are more homogenous the new extracted events have to fulfil

the condition (number of pixels of event \* 1.6)  $\geq$  (number of pixels in bounding box) if they are smaller than 40 pixels. This results in a smaller number of extracted calving events but it is less sensitive to noise. We included a positive volume change graph in the appendix (Fig. S6).

After our re-analysis as suggested by the referee 1 the main differences between the shallow and deep 5 part in terms of calving remain, even if less calving events were extracted.

According to the authors, there was very little surface calving observed by TRI at sector D (the deep water sector), and mass loss due to subaqueous calving is dominant (50% or more, depends on the rate of oceanic melt) here. Limited evidence of subaqueous calving was shown in the manuscript. Even

- 10 if substantial subaqueous calving events occurred and contributed significantly to the mass loss at the deep water sector, the manuscript failed to explain where the mass goes. I also think it is not adequate to simply assume that subaqueous calving is independent of TRI observed surface calving. If subaqueous calving would not cause surface elevation change by following the authors' logic, then what was there to fill the space left by the "subaqueous calving"? Besides, if TRI observed little calving
- 15 at sector D, then glacier front at this section should advance, especially at the high velocity area. Speed in the middle of this sector is 16 m/day, in ~7.65 days, ice front here can advance over 100 meters, much larger than the resolution of either Landsat/Sentinel satellites or TRI images so should be detectable. However, Supplementary Figure S1 rejected this.
- 20 We agree that the explanation of the processes happening at the deep sector was not complete. We cannot see subaqueous calving events with the TRI data but we added images of a subaqueous calving event from the time-lapse camera installed in 2018. Unfortunately, no time-lapse camera was installed in 2016. As the flow field and surface slope further upstream is homogenous (smooth across-flow profile velocity profile) and thereby does not indicate substantial differences in ice thickness we
- 25 assume that the ice thickness downstream (towards the front) is similar for the shallow and the deep sector also at the calving front. We argue that if this assumption would not hold, we would see it in the surface structure of the glacier by specific crevasses or flow velocity variations. Thus, if the ice thickness is similar, at the deep sector about 45 % 65 % of the ice area are below the waterline. The remaining missing volume might be calved off above the water line through small calving events, which
- 30 were filtered out during the analysis. Those small events can be caused by undercutting of the calving front due to ocean melt and calving below the waterline. We calculated now the flux with the available front height and velocity and also with an assumed ice thickness of 150m. This estimated flux is included in Figure 7.
- 35 The manuscript did not explain the method they used to choose the study area for calving detection well. Although on page 6 the authors mentioned that they applied a mask with ~150 m wide across the glacier front, however, the glacier front was constantly moving, so a Lagrangian frame should be used. Whatever the reference frame was, according to the methods presented by the authors, areas with calving event detected (Figure 4) over the center of sector D
- 40 should have the largest along-flow direction width. This is because glacier front at this location should advance (also see comment above), thus the test area should move. Whereas Figure 4 shows a different pattern.

We added a figure showing the used front position mask and an example from the watershed
algorithm. The front position advanced and retreated only marginally but was always well within the mask. Thus we decided to use a simple constant mask. The fact that the centre of sector D does not

have the largest along-flow direction width we explain with many small events, which might be triggered by undercutting of the calving front due to underwater calving and ocean melt.

**5 **Detailed comments:**

Page 2, line 4: "was" —> "were", data should be plural.

We adjusted this accordingly in the revised manuscript. Additionally, we changed data to plural in the whole manuscript.

Page 2, line 6: "style" -> "styles".

10 We corrected this in the revised manuscript.

Page 2, line 8: "missing" —> "deficiency" ?

We changed this in the revised manuscript.

Page 2, lines 8-10: later in the manuscript, one conclusion was that that subaqueous calving and oceanic melt combined contribute ~75% to the frontal mass loss. However, here it seems that

subaqueous calving itself contribute up to 75% of the frontal mass loss. Please clarify.

We rephrased this sentence and hope that it is clearer now.

Page 2, line 18: It would be great if the authors can be more specific about "water masses in Greenland". Did the authors mean "water masses around Greenland", or "increase in surface water due to melt"?

20 Straneo et al. (2013) suggested that a warming of the subpolar North Atlantic together with an increased runoff lead to enhance submarine glacier melting. So here we meant water masses around Greenland. We changed it to "water masses around Greenland".

Page 2, line 21: What are the major remaining limitations? A few examples briefly listed here would be helpful.

25 Examples would be that the link between atmospheric forcing and calving activity is not straightforward and that with the currently available resolution in models and observations small scale processes like subglacial hydrology are not resolvable or that short term and long term observations are often not available. We included some examples in the revised manuscript.

Page 2, line 24: I am not sure if calving controls tidewater glaciers's react to environmental condition changes, or verse visa? Please clarify.

We agree that the sentence was confusing. We changed it in the revised manuscript to: "Calving is a crucial process for the dynamic behaviour of tidewater glaciers, but the detailed mechanisms and

relation to environmental forcing are not well understood (Joughin et al. 2004; Thomas, 2004; Nick et al., 2009)."

Page 3, line 8: "was" -> "were".

We corrected this in the revised manuscript.

5 Page 4, line 7: Figs. 1 and 2.

We changed this in the revised manuscript.

Page 5, line 13: "whole" -> "entire".

We corrected this in the revised manuscript.

Page 5, line 18: "climate" -> "weather".

10 We corrected this in the revised manuscript.

Page 6, lines 16-18: Are these differences RMS difference, Mean difference, or other types? These can be quite different because the difference between TRI-DEM and Arctic DEM can be systematically and/or randomly. Please clarify.

The differences calculated here are mean differences but we added a more complete comparison of the TRI-DEM and the Arctic DEM in the revised manuscript in section 4.1 and Figure S7.

Page 6, line 17: Does "variability" mean "repeatability", or changes of topography? Please clarify.

We extended this section to a more detailed error analysis in section 4.1, where we investigate the variabilities on stable terrain. With variability we mean the difference between two DEMs for points on stable terrain.

20 Page 6, lines 17-18: How stable/random is stable/random? Maybe outline the test area in one of Figures 1, 2 or 3? And what does the "values" in line 18 mean? Values of a selected area (if so, please outline it in Figure 1/2/3) or selected DEMs (if so, maybe mark the times in Figure 5c)?

We extended this analysis in section 4.1. The test area on stable terrain is marked in Figure 3.

Page 6, lines 23-24: Does the "10 pixels" mean "10-adjacent pixels"? I feel the two numbers "10 pixels"
and "3 pixels" are confusing: Does noise needs to fulfill both "area<10 pixels" AND "width<3 pixels"? If so, how about a block with 3×3 (9 pixels, each pixel shown by an "O" below) shape like</li>

" 000

000

OOO ", or a 2×8 shape like

30 " 0000000

00000000 ", or a shape like

" OO

00000

0000 "

5 Are these considered calving events if all "O" pixels have elevation decreases more than 5 m? Because many of the identified calving events are quite small, and shapes of these blocks may not be regular, I think it is important to clarify these settings here.

We agree with referee 1 that the settings for the thresholds need to be clarified more to avoid confusion. Calving events need to fulfil both conditions, a size of 10 adjacent pixels and a width of 3

- 10 pixels. For the width it is enough if the bounding box has 3 pixels. All the pixels need to have an elevation decrease of more than 5 m, except if some pixels with a lower decrease are surrounded by pixels with a decrease of more than 5 m. Then those pixels are included in the calving event. So a 3x3 and a 8x2 shape are not considered as calving event but the third shape you mention is extracted as calving event.
- 15 In the revised manuscript we changed the description accordingly to: "Additionally, calving events smaller than 10 adjacent pixels and with a bounding box width smaller than 3 pixels were excluded as noise. Thus only calving events with both, ≥ 10 adjacent pixels and a bounding box width larger than 3 pixels, were extracted."

Page 6: Apart from using elevation changes to detect calving, it is also possible to identify

20 calving from radar amplitude images. Including an example showing both changes in radar amplitude and elevation would be strong evidence that the method is reliable.

Instead of including an example of identifying calving from radar amplitude images we added in the supplement an example of a calving event which is visible on the multi-look radar images (Fig. S5).

Page 7, lines 5-6: Did the authors mean that p should always be larger or equal to 0.1, or did they mean that one can only trust the sign of R when  $p \ge 0.1$ ? If it is the latter, maybe rewrite the sentence to "which tells if one can trust the sign of R (when  $p \ge 0.1$ )"?

We changed the sentence to "which tells if one can trust the sign of R (when p≥0.1)."

Page 7, line 15: Please also specify the low-pass frequency. Just as how the high-pass frequency was given.

30 The pass frequency for the low-pass filter was 0.001 Hz. We added that to the revised manuscript.

Page 7, line 19: To avoid confusion for readers who are not familiar with radar, I suggest to add "line-of-sight" in front of "shadow".

**We added "line-of-sight" in front of "shadow".**

Page 7, lines 26-28: I found that the number of total identified calving events is smaller than the sum of identified calving events in the shallow sector and the deep sector (1681 < 1403+289). Did I miss anything? Please also check numbers in Table 1, many of them are not consistent.

5 The number of total identified calving events is smaller than the sum of identified calving events in the shallow and the deep sectors, because 12 events happened to be located on the boarder of the two sectors. Those events were counted only once for the total number but for both the number of the shallow and the number of the deep sector (double counting).

We added an explanation to this in the revised manuscript on page 8, line 6-7: "Note that 12 events
were detected on the border of the two sectors and were thus counted for both sectors but only once for the total number of events."

Page 8, line 2: Please ensure the minimum size of identified calving block fulfill the threshold defined for calving events (based on lines 21-24, page 6, I calculated a minimum volume 5×10×3.75×3.75=703 m3. Or did I misunderstand the "resolution"? — I picked it from line 13, page 6. Using the radar pixel

15 specified in line 6 page 8 the minimum volume of identified block is even larger, i.e., 1500 m3). If this paper aims to do statistics of calving event sizes, please ensure that the statistics are correct.

We thank referee 1 for spotting this. We realized that some of the events were considered as calving events even if they are collapsing seracs upstream on the glacier surface. They were included as they were located exactly on the border of our glacier front mask. Those events are excluded in the revised

20 manuscript. The removal of those small events influences the calving size distribution but all the other results and the main conclusions are not affected.

The algorithm takes also events into account where pixels of less than 5 m decrease are surrounded by pixels of more than 5 m decrease. Thus the events can be smaller than the calculated minimum size of a calving event.

- Page 8, lines 6-9: I think to calculate cumulative calving height a Lagrangian frame needs to be used, because ice at the front can move at 16 m/day (line 21 on page 7), which means a ~100 m displacement during the observation period. If the authors were referring to cumulative calving height from calved ice height at each pixel in each calving events, then line 6 needs to be rewritten, at least taken away "differences" because calved height was estimated from the difference between two DEMs.
- 30 We added a figure in chapter 4.1 showing an example of the watershed algorithm and also the used front mask. In figure S1 it becomes visible that the front is changing its position only marginally. Thus using a constant mask seems to be appropriate.

We changed the sentence to "Calving heights in each radar pixel.." in the revised manuscript.

Page 8, line 8: If cumulative calving height exceeds 300 m but not up to 300 m, please

consider using "extend='max" for the color bar of Figure 4.

**We changed the Figure accordingly and used extend=max in the color bar.**

Page 8: I think TRI-derived DEMs in this paper are very important data, however, there was no figure showing the DEM, neither in the main paper nor in the supplement. Please consider to include a TRI-5 derived elevation map in the manuscript. Maybe add a panel or two in Figure 3?

**We included Figure 3 of a TRI-derived DEM in section 4.1.**

Page 9, Figure 4: Comparing this figure with the text makes me confused. If ice velocities on the two sides of sector M are the fastest (see Figure 3) but cumulative calving heights are not the largest (see colormap in Figure 4), shouldn't these two areas have the widest (along flow direction) spatial

10 distribution of calving events? Please check data processing, and make sure the descriptions in lines 25-27 on page 6 are correct.

The calving front only changed marginally during the observation period. We explain the missing volume for sector M with small calving events, which we cannot detect with our method. We added this explanation to the discussion in section 5.1.

15 Page 9, Table 1: Please check calculations and make sure that numbers are consistent in the table and the main text. In the table, please at least ensure numbers in "Total event volume" equal to "Total event number" × "Mean event sizes", unless a different math was used.

The numbers in table 1 are rounded as a more detailed number makes no sense due to the uncertainty. So the "Total event volume" equals to "Total event number" x "Mean event sizes".

- 20 Page 10, Figure 5: I like this figure! But I also have a few questions and suggestions. First, color changes from dark blue to light blue in Figure 5c is distracting, I suggest to mask out periods without calving using white or grey. In this way, calving characteristics will be more accessible. Consequently, the minimum value of the color bar can be changed to the smallest volume detected based on the settings (lines 20-25 on page 6). Second, I am confused by the right axis of Figure 5d.
- 25 Can the authors elaborate on this? Third, I could hardly read the superscripts in y-axis label of Figure 5d due to low resolution. Based on the manuscript I guess it was "106 m3" in the bracket, is this correct? Please increase the figure resolution. Also, maybe add "Cumulative" to the y-axis label of Figure 5d to distinguish from the color bar label in Figure 5c. Fourth, I found that the further analysis separates these sectors, so it is necessary to show the exact along-distance ranges of different
- 30 sectors. Maybe use vertical bars to mark the boundaries of different sectors? These bars can go between the annotations in Figure 5c, such as "| SL | SM | SR | M | D |". Last, would it be possible to add a narrow column on the right of Figure 5c and show total calving volume along the entire calving front in color? Such a plot may provide useful information on calving volume changes with time.
- 35 We thank referee 1 for the suggestions. We changed the colours in the Figure 7c. We added cumulative to the y-axis of Figure 7d and we increased the resolution. Also the vertical bars to mark the boundaries of different sectors was included. We did not add a narrow column on the right of Figure 7c as we think this would be too much information for one figure. Also this information is already included in Figure 12.

Page 11, line 4: "already observed" -> "shown"

We changed this in the revised manuscript.

5

Page 11, line 10: It is probably not correct to say "only ... was observed" because small calving volumes are also visible at sector M in Figure 5c. Maybe rewrite to something like "less calving events but several of them are significantly larger than those observed at the other shallow water sectors".

We rephrased the sentence to: "In the central, very shallow sector M less calving events were observed but several of them are significantly larger than those observed at the other sectors."

Page 11, lines 11-12: I think there is an ambiguity in "the most individual events". Maybe rewrite it to "the largest number of calving events".

10 We rewrote the sentence to "Sector SL with the highest cumulative calving height also has a large number of events,...."

Page 11, line 14: I agree that no clear temporal pattern can be seen throughout the different sectors, it looks pretty much like random noise. Please see my comments above.

With the additional condition we excluded more noise but still have no clear temporal pattern.

15 Page 11, line 15-16: The "observable cluster of calving" is hard to tell from Figure 5c. Yes there are some big events, but since this manuscript does statics, in the sense of statics, do these relatively big events really clustered? Need more elaboration.

We think the observation period is too short to do more statistics. But we will rewrite that this observable cluster is only investigated by looking at it.

20 Page 11, lines 19-21: More detail of deriving the 25% needs to be provided. Is it an appropriate assumption of a constant front position? Here ice can move up to 16 m/day (Figure 3). And the assumption of a constant mass flux over the front also needs to be justified.

We added in Figure 7c two ice flux estimates along the front and adjusted this sentence accordingly. The front position only changed marginally over the whole observation period.

25 Page 11, line 30: A summary of the meaning of the log-likelihood ratio R would be helpful for understanding the statistics. It seems to be an important parameter describing the likelihood of two different models. Also, were there any reasons to choose the three models here? Since one of the major conclusions came from statistics, more details should be provided.

We added an explanation on how to understand R in this section. We used the same models as others
before as we wanted it to be comparable to other studies. Also this three models are widely used in natural science.

Page 12, Figure 6: In the abstract I found "The size distribution of the deep sector follows a power law, while the shallow sector is likely represented by a log-normal model." From this figure, could we say

that both can be represented by a log-normal model? The R and p values are identical for the power law and log-normal models in Figure 6d. Or did I miss anything?

With the re-calculated dataset the shallow front follows a log-normal model ( $p \ge 0.1$ ), while for the deep front log-normal and power law model fit well, but none of them significantly better.

5 Page 12, line 9: Can the authors provide the unfiltered water level during the entire observation period? So that readers will know the overall characteristics of local water level variations. It can either be in Figure 9, or go to the supplement, maybe add one panel to Figure S2.

The unfiltered pressure sensor dataset is added to the appendix in Figure S9.

Page 13, line 14: "several clusters of events"? I thought the authors only observed one cluster of calving events (lines 14-16 on page 11). Please clarify.

We thank referee 1 for spotting this. We changed the sentence to "The observed calving events show no temporal or spatial pattern, except for a series of bigger events on 26 August."

Page 13, line 15: How did the authors reach to a conclusion that "no clear temporal pattern of tidal or diurnal recurrence could be detected"? Can the authors elaborate? If no evidence, I suggest to omit

15 this sentence. Or at least admit that this is based on the impression of looking by eyes.

We changed the sentence to "The observed calving events show no obvious temporal or spatial pattern, except for a series of bigger events on 26 August"

Page 13, line 18: I think some of these values can be calculated from the data. If the authors use values derived from real data, the further analysis would sound more reasonable. Also, maybe use "a front thickness" instead of "a front height" to avoid confusion?

We calculated the ice flux in the revised manuscript by using the available elevation and velocity. The calculated flux is added to Figure 7d.

Page 13, lines 18-23 and after: Please try to keep number of digits consistent.

We changed this accordingly.

25 Page 13, lines 20-21: Not sure if it is correct to say "This value should match up to …"? Although the front position looks stable by eyes, but is this sufficient to support an assumption that total ice flux should match observed total calving volume? More rigorous analysis is needed.

We rewrote the whole section as we calculated now the ice flux from the available data.

Page 14, line 1: Remove "with", add "," before about.

30 We rewrote the sentence in the revised manuscript.

Page 14, line 3: Please check if "160 m3" is correct. And stacking does not likely contribute to the difference unless there are some errors in the data analysis.

As we changed our data analysis this values changed in the new manuscript.

Page 14, line 9: I bet the " $0.47 \cdot 106 \text{ m3}$ " correspond to the deep sector? Why assuming an ice thickness of 100 m below the water line? Too many assumptions could result in significant bias. Since the authors have data of surface elevations (shown in Figure 5b), and have assumed a front thickness of 150 m (line 18 on page 13, although I suggested to use real data instead of assumption), ice

5 thickness below water line can be estimated.

We changed that section as we calculate now the ice flux with the available data over the front. Thus by assuming a total ice thickness of 150 m we can calculate also the corresponding ice flux below the waterline. All this information is added to Figure 7d.

Page 14, lines 16-18: If subaqueous calving cannot be detected with the TRI, how could it be detected by visual observations and time-lapse imagery? More details are needed. Maybe show some examples of images taken by the time-lapse camera.

We added an example of a subaqueous calving event recorded with the time-lapse camera in 2018.

Page 14, lines 19-12: I don't think the authors have shown enough evidence to reach this conclusion. If subaqueous calving account for ~50% of the mass removal from the deep sector, the average

- 15 thickness removed at the glacier front could be calculate by doing simple math. I believe that will lead to significantly mass thickness removal and the deep sector may become afloat by the end. Plus, can the authors see icebergs coming out from subaqueous calving? If it accounts for ~50% of mass loss, then visual observations or time-lapse camera images (line 17) may be able to see icebergs coming out from subsurface. Please provide evidence to support the conclusion.
- 20 With the new ice flux calculation this value changed now and it is now 45 60 % mass loss is due to oceanic melt and subaqueous calving. We added an example from the time-lapse camera.

Page 14, lines 25-26: If ice cliff at the shallow sector can have larger but stable height, then why do calving events occur so frequently? Although ice here is thicker but calving should be less frequent or no calving at all because ice cliff can be stable (lines 25-26). May the authors were referring to the

- 25 potential of calving so"a thick cliff CAN release larger ice volumes", but please note that here calving is quite frequent (also related to how "stable" was defined in this manuscript), while previous figures (e.g., Figure 5c) show that the shallow sectors calved more frequently at the surface. Even add subaquatic calving to the deep water sector, calving at the shallow sectors will still be more frequent than the deep water sector because the authors assumed the overall mass loss are similar in different sectors, plus
- 30 the deep water sector has lost more mass due to melt.

40

We changed that sector to explain it better. The vertical front of the deep section might results in smaller events, which are not detectable with the TRI.

Page 15, lines 4-7: Would enhanced submarine melt cause surface elevation decrease because ice becomes thinner? If it won't lead to surface elevation decrease, then what was there to support the

35 upper part of the glacier? Would the empty chambers cause instability and calving? On the other hand, if it will lead to surface elevation decrease, then the TRI might be able to see the decrease. Please clarify.

With the TRI there is no visible decrease in surface elevation due to subaquatic calving. But the subaquatic calving can lead to instabilities and thus to more calving of smaller volumes, which are not detectable with the TRI.

11

Page 15, line 12: Please show the "Observed subaqueous calving events". Because the authors wrote that there were time-lapse cameras (line 17 on page 14). If no captured subaqueous calving events by instruments, then please provide more detail of the available observations by authors in the field.

We added an example of a subaqueous calving event recorded with the time-lapse camera in 2018.

Page 15, line 19: "acting" —> "dominant"? 5

We agree with referee 1 that "dominant mechanisms" is more appropriate and changed it accordingly.

Page 15, lines 19-24: The paragraph relies on the assumption that submarine calving in the deep sector is a major contributor to total calving volume. Please evaluate the assumption based on comments above.

We added an example of such a subaqueous calving event. The size distribution of the deep sector 10 can be represented by a log-normal and a power law model but no model is significantly better. Thus we reformulated this paragraph and phrased it less certain.

Page 15, line 30: Please explain what is "big up-floating icebergs". If the icebergs are big, they might appear on the TRI amplitude images. An example image would be helpful.

15 We could not find a nice example on the TRI images as the icebergs often fall apart after they emerged. Thus we added an example from the time-lapse camera.

Page 16, lines 4-6: Yes I agree that pressure sensor observations could be used to derive calving events, but challenges remain. One challenge is that subglacial hydrological events may cause similar signal as what has described as subaqueous calving in this manuscript.

Need justification. 20

25

We added a sentence about the challenges with pressure sensors. However, a more detailed analysis of the pressure sensor data is not the scope of this study and will be done in further work.

Page 16, lines 13-15: Perhaps this paragraph needs to be rewritten, because I don't get the logic of cause and effect. Sentences before and after "Therefore" seem to be out of place. What do tides do with air temperatures and radiation? Other readers may also be confused.

We rewrote this paragraph to make it clear that the surface melt can influence the water level in the crevasses or the glacier dynamics through the subglacial hydrology.

Page 17, Figure 9: In (d) and (e), are the calving events in the deep sector plotted above calving events in the shallow sector? Please note this in the caption, otherwise readers can assume that all

these histograms start from 0 in the y-axis. 30

We added the sentence "The calving events in the deep sector are plotted above those in the shallow sector." to the caption.

Page 17, line 10: Maybe "surface" should be added to the front of "calving event" because subaqueous calving was also discussed?

We added aerial in front of calving event as this is the term used in the rest of the manuscript.

Page 17, lines 13-14: Or it may not have to be explained by other processes? I don't see the necessity of assuming similar ice flux in the two sectors.

If the ice flux would be very different for both sectors, we should see this in the flow field and surface

5 characteristics (e.g. crevasses), but higher up the velocities are very similar for both sides of the glacier. We clarified this in the revised manuscript.

Page 18, line 2: Here "center" was used, but at other places "centre" was also used. Please be consistent.

We changed center to centre so that it is consistent in the whole manuscript.

10 Supplement page 1, Figure S1: A more accurate job needs to be done if the front line was estimated visually, because this research studies many relatively small calving events. If there are satellite images at the beginning and the end of the campaign, then they should be plotted here. If no available satellite images at these times, TRI amplitude images can help.

We redid the visualization of the calving front position more accurate.

15 Supplement page 1, Figure S2: As I commented above, please add a panel to show unfiltered water level.

We added a figure with unfiltered water level data in the supplements (Fig S9).

Supplement page 2, Figure S3: I thought the tide data were heavily filtered, why there is a jump around 08:00 on the 20th? Please check.

20 We thank referee 1 for spotting this, the jump is removed in the revised manuscript.

**Reply to major concerns Referee 2**

**25**

DEM derivation of the glacier front from TRI The critical part of the paper is the derivation of digital elevation models of the glacier front from terrestrial radar interferometer as developed by Strozzi et al. (2012). However, this method is known to be uncertain, although the large glacier size should help having a greater signal to noise ratio. I think

- 30 that it is important to extend the paragraph on the error analysis and dedicate a specific figure with a map of the derived DEM(s), statistical distribution of the error in the discussed stable terrain. Assess the uncertainty of the glacier part with the UAV derived DEM too by replacing Figure 3 as the velocity comparison is done in Rohner et al, 2019. I would like to have a Figure showing a study case of the detection and watershed algorithm to assess issues with signal to noise ratio and uncertainty in radar
- 35 geometry or cartesian coordinates (in the main text or supplementary material).

We agree with the concern of referee 2 that the error analysis needs more elaboration. We added section 4.1 and Figures S2, S3 and S4, where we investigated the error on stable terrain over time and space. We also compared a stacked TRI DEM with the Arctic DEM.

A comparison with the UAV derived DEM to investigate the uncertainty of the glacier part is currently

5 not possible as we do not have a georeferenced UAV DEM for this time period. To do this comparison we would need to georeference it and do a detailed error analysis, which is not part of this paper. Additionally, as we do not have ground control points on the glacier the UAV DEM might be warped in the centre of the glacier.

Figure 3 is not meant as a velocity comparison, but the UAV velocity is rather a completion of the TRI-

derived velocity data to illustrate the flow field (which we also use to estimate and discuss ice fluxes 10 later in the manuscript).

We add a figure in section 4.1 showing a study case of the calving event detection.

Issues in determining best fit models for calving distribution

- 15 As seen on Figure 6c and 6d, it is not possible to distinguish between a power law and a log normal models as indicated by a low loglikelihood ratio R between the two distributions and a poor significance value, p>0.1. The only evaluation possible of the power law is a comparison with other heavy-tailed distributions. The conclusion is that the shallow and deep part does not exhibit a transition in distribution from power law to log normal as they cannot be statistically differentiated from each other.
- Discuss instead whether the distribution over such a short period is representative. 20

With the changed calving event size distribution we find that the shallow sector is following a power law with a p value  $\geq 0.1$ . For the deep sector both the lognormal and power law models fit well, but none of them fits significantly better. We also agree that the period is rather short to have a representative

distribution and we will add a sentence about that. Since we observed a big number of calving events 25 we are convinced that a statistical analysis is meaningful and legitimate.

Ice flux budget: bed topography and missing component

The paper bases its analysis on the depth of the fiord but no bed data is provided to support this description (just observations of surfacing rocks). Please use the BedMachine v3 to at least provide an 30 idea of the fjord depth in front of the glacier to the reader. The shallow part may be constituted of two bed pinning points beside a deep valley. Furthermore, simplify the subdivision of the shallow part to only the shallow part regrouping SL, SM and SR.

- The simple ice flux calculation holds some caveats when identifying a missing volume due to the uncertain fluxgates and filtering of small events. The distribution of these small events may be related 35 to calving mechanisms and ocean melting (undercutting). A more realistic flux can be derived by integrating the ice flux with your surface elevation and velocity data. See my minor comments to improve the understanding of section 5.1.
- We added the data of BedMachine V3 to the supplement (Fig S8) of the revised manuscript to give an 40 idea about the fiord topography. However, as no direct measurements are available directly at the calving front, the data of BedMachine V3 at the calving front should be considered with care. At and near the calving front the velocity is influenced by additional processes and the estimation of the ice thickness by inferring the surface elevation and velocity data cannot reproduce the bed topography correctly.
- 45

We added a more detailed analysis of the ice flux and show it in Figure 7d. We calculated the potential ice flux per bin with our data on velocity and surface height per bin. To include the calving below waterline and the ocean melt we assumed a total ice thickness of 150 m and calculated the flux with this value and the velocity per bin. However, note that these flux estimates are only rough estimates to

5 analyse the rough shares of different calving processes and not exact values of different calving fluxes (but they are consistent with the observed flow fields in Fig. 5). We will keep our subdivision of the calving front as we think it is needed for the interpretation. The three parts of the shallow sector show different characteristics likely due to the bed topography and thus they cannot be seen as one homogenous sector.

10

**Better integration of calving wave dataset**

The paper should integrate better the ocean wave data as an alternative dataset of calving events (including subaqueous ones?), explain this discrepancy and discuss other potential sources such as iceberg rolling. This better integration of the wave amplitude dataset with the TRI detected calving events would strengthen the discussion and expelusion of the paper.

15 events would strengthen the discussion and conclusion of the paper.

We added a peak detection analysis of the calving wave dataset for a comparison with the TRI detected calving events in Figure 12. However, in this paper we want to focus on the TRI dataset and the established methods. A more detailed analysis of the calving wave dataset is beyond the scope of this paper and the topic of a follow-up paper.

**Reply to minor comments**

25

page 2

**Abstract**

Please mention that the study is based on derived digital elevation models. Focus on your findings right after your methods instead of following the paper structure: the characteristics of the shallow/deep part (I.5-7), then calving missing calving volume (I.8-10), self-critical system vs less complex model

30 (I.10-11), Calving models vs front geometry (I.11-12) and finally lack of relation to air temperature and tides.

We thank referee 2 for the suggestions. We added that the study is done by using digital elevation models. Otherwise we think we already have the suggested structure or did we misunderstood it?

**I.2: "in understanding the processes of calving"**

35 We adjusted this in the revised manuscript.

1.5 can you find a better word than "source area"? "vertical front area"?

We find source area the more appropriate term as it implies that this is the location where the calving event occurred.

1. Introduction

It is well written and nicely placing the work in its context.

page 3

5

I.12 Add Kohler et al., 2015 in Polar Research [TOCHECK] as they produced the longest calving time series based on seismic records (20 years)

We added Köhler et al. 2016 and we thank referee 2 for the hint.

10 I.14 Delete "can only detect large events" as they can detect small events (even ice falling in a crevasse at the front, Kohler et al., 2019 in The Cryosphere Discussion) when placed close to the glacier front. Their main issue is the volume scaling.

We deleted this part of the sentence.

15 2.1 Study area

I.30 Add that the glacier has been stable and even advanced since 2016 similarly than Jakobshavn Isbrae (based on Planet daily imagery).

Eqip Sermia had a fairly stable front position during the last years with only a small retreat at the southern margin. We added this to the section. We specifically checked satellite images and we cannot see the advance of the front position mentioned by the referee.

I. 32 Indicate the time period when 16 m day-1 was obtained: "as measured over our two week period in 2016"? as the 2.5 and 5 m day-1 represents annually averaged velocities, correct?

We included: "...measured over the observation period in 2016...".

25

page 4

I.1-7 Also present the glacier bed or bathymetry provided in the BedMachine as it covers an area that is now deglaciated (as they use an older surface elevation and glacier mask) and upstream bed geometry is also important to understand the glacier flux at the front.

We added the bathymetry provided in the BedMachine v3 to the supplement (Fig S8) and added a sentence in the study side section.

page 4-5

The TRI and environmental data parts are complete and informative.

page 6

**3.1 TRI data processing**

35 I.15-17 Is the elevation difference just a shift in absolute elevation explained by a difference in geoid or geo-referencing problems of the Arctic DEM or your DEM? Also co-registering the two DEMs before differencing is useful to assess systematic errors outside obvious artefacts. We added a more complete comparison in the revised manuscript between the Arctic DEM and our DEMs and investigating there the variabilities on stable terrain in more detail. The result is presented in Figure S7. The two DEMs were co-registered before comparing them. The elevation difference is likely not just a shift, it also depends on the slope of the terrain.

- 5 I.18-19 Please provide a sentence about precision change over time on stable terrain and also ice. You could plot this variability on stable terrain and some upper part of the glacier for the single DEMs and the stacked ones to appreciate the effect of atmospheric disturbances and the improvements from stacking. You could use this variation to provide first order error bars for your volume estimates on Figure 6. In the discussion could you compare your precision to what other studies found.
- 10 We thank referee 2 for the suggestion, we added a section on the variability between the stacked DEMs on stable terrain over time and space. However, as the glacier area measured with the TRI is highly crevassed and fast flowing, we do not think that an analysis of the variability there is meaningful. We only looked at the variability for the stacked DEMs, however as an example for the watershed algorithm we also included a plot from a non-stacked DEM. We do not think that first order error bars
- 15 due to the improvements by stacking are meaningful, because it is not only the stacking but also the used threshold which influences the increased signal to noise ratio.

I.20-21 Indicate that the watershed algorithm uses elevation change as source image and merges calving events occurring within 10 minutes due to the stack. Can you define algorithm parameters like the number of start points or maximum points for reproducibility? Please plot an example of your

- 20 watershed results (here or in supplementary materials) to assess the effect of noise on your segmentation. An error of 10 pixels in radar geometry already causes a volume error of 5625 m3 in range and 12000 m3 in azimuth using 150 m of ice thickness, that is the same order of magnitude than your calving volumes.
- We added this information to the method part. Number of start points and maximum points are not needed for the algorithm. The only needed parameters are a maximum and minimum value, which are used for the markers in the algorithm. The minimum value is 5 m as we define this as background, while for the maximum value we used 15 m. We added the maximum value to the methods part. We added an example of the watershed results to section 4.1.

1.23-24 "10 pixels in area" Specify that in the context of your grid asymmetry, your area filter is more
30 likely to remove events that are long instead of wide, thus you apply this second filter of 3 pixels. Add that 3 pixels is equal to 11.25 metres.

**We added this information to the manuscript.**

I.24 "When applying [...] are removed". The noise observed on stable terrain is not removed, but the signal to noise ratio is higher for the filtered events. Moreover, quantify the number or area of excluded
events or give a percentage.

We changed that sentence to: "When applying these filtering thresholds, the signal to noise ratio is higher on stable terrain than for the non-filtered events." We added a percentage of excluded events to section 4.1.

I.20-24 How do you deal with the zero elevation or water elevation when calving occurs along the entire ice column (i.e. column collapse)? Parts of the DEM covering the sea may have Not A Number values or problems with icebergs?

The water elevation is set to 0 also where there are Not A Number values. Thus, it does not influence the calculation if calving occurs along the entire column. The calving at Eqip Sermia happens mostly through ice avalanches, not resulting in big icebergs. Bigger icebergs remove the ice melange in front of the glacier, which results in a loss of coherence. Due to this loss of coherence the area including the iceberg is not included in the analysis. Icebergs further away are not included anymore as they are not within the glacier front mask.

**10**

**page 7**

I.4-6 Rephrase the last line that explains what a "good" *p* value is and means. Specify that the maximum-likelihood methods are used because of the non-linearity of the fitted curve and that one implication is that the resulting log-likelihood is a relative score of how good two fitted models perform against each other instead of "an absolute score".

15 We added this information to the manuscript and changed the last line according to referee 1.

I.8 Add that 120 interferograms is approximately 2 hours.

We added this information to the manuscript.

I.7-10 Indicate the theoretical maximum velocity that the TRI measures with an interval of one minute (it should be of the order of 6 metre per day) as this will be useful to explain the differences with the
 UAV velocity data. [TAZIO equation]

We did not add this information as the UAV velocity data is not meant for comparison but only as an additional information (for example for the flux estimation) due to the limited area of the TRI velocity field.

**3.2 Pressure sensor data processing**

25 I.15 Indicate the frequency of the low pass filter or used methods.

We added the pass frequency of the low pass filter, which is the same as for the high pass filter.

**4. Results**

The key result of the paper is the TRI-derived DEMs but the velocity (4.1) is presented first instead. Add text and figure(s) specifically on the generated DEMs and signal improvement by stacking before section 4.2 on calving detection results or a comparison with the UAV DEM.

We added a section about the derived TRI DEMs and the variations before the velocities as section 4.1.

4.2 Magnitude and source of area of calving events

Abandon the subclassification of the shallow sector as it confuses the results

We will keep the subclassification as the three parts of the shallow part are not homogenous.

I.25 I suggest a simpler section title: Area and location of calving events?

We thank referee 2 for this suggestion. However, we think area is misleading as it is more the size of the events which is discussed here.

5 I.29 Add the number of filtered/removed events for each sector as the filter may affect the number of calving events. I have a hunch that the deeper sector may have more small events, likely filtered out, due to the effect of higher submarine frontal ablation.

We added the sentences: "Comparing the amount of extracted events for a threshold of 1 m and of 5 m shows that with the threshold of 5 m 77% less events were extracted for both, the deep and the

10 shallow sectors. The usage of the shape condition for events smaller than 40 pixels leads to 49% less events for the shallow and 54% less events for the deep sector." to section 4.1.

**page 8**

I.1 Replace "frequencies" by "number"

We changed that accordingly.

15 I.2 "four orders of magnitude".

We thank referee 2 for spotting this. We changed it in the manuscript.

I.2 Use the same order of magnitude i.e. 103 for easier comparison, too.

We changed the numbers to the same order of magnitude.

I.2 I do no understand how you get a minimum volume of 160 m3. If you take a minimum area of 10
 pixels with 30 m2 per pixel, you get a height of 0.53 m. This does not match your vertical change threshold of 5 m. So, I guess the comma is misplaced, it must be 1.6 103 m and you were correct with "three orders of magnitude".

Volume / (10x Pixel Area) = Height or 160/(10x3.75x 8) = 0.53.

We thank the referees for spotting this. We realized that some of the events were considered as calving events even if they are collapsing seracs upstream the glacier. They were included as they were located exactly on the border of our glacier front mask. Those events are excluded in the revised manuscript.

The algorithm takes also events into account where pixels of less than 5 m decrease are surrounded by pixels of more than 5 m decrease. Thus the events can be smaller than the calculated minimum eize of a solving event.

30 size of a calving event.

I.8-11 Delete the subdivision of the shallow sector it does not bring much to the comprehension of the calving distribution. Or just keep the rock part: M.

We will keep the subdivision as the shallow sector is not homogenous. Sector SM has less events than the other two sectors, which is likely caused by differences in the bed topography.

**page 9**

Table 1: Could this table be combined with Figure 6a and 6b by placing a text in the corner or a horizontal boxplot at the top? Rarely calving distribution are presented as a table, making it difficult to

**compare with other studies. 5**

We understand the concerns of referee 2, but also other studies used a table and we find it clearer like this.

**page 11**

1.19-21 Develop the detail of your computation (numbers?) and how you obtain 25% in the text and on 10 Figure 5d.

We exchanged the right axis of Figure 7d and provide now calculated ice fluxes over the front. The calculation on the percent are done in the text.

**4.3 Calving statistics**

1.23 Describe first what you want to achieve, meaning model the calving distribution with non-linear 15 fitting models to assess whether you observe a "self organised critical system".

We added the sentence: "The calving distribution was compared with non-linear fitting models to investigate if a self-organised critical system can be observed."

1.27-31 It is not clear to me what is the basis for selecting a log-normal against a power-law in both sectors as the results of the maximum likelihood (and visual inspection) show that the fitting models are as good and with similar parameters. 20

Due to the new condition used in the filtering of calving events, the event size distribution changed. The shallow sector follows now a log-normal model with a p value = 0.1, while for the deep sector both a power law and a log-normal model fit, but none significantly better.

page 12

**4.4 Pressure sensor records 25**

In order to find the missing component presented in the discussion, it would really help to derive a rough calving catalogue based on a peak detector or even manual picking and neglecting other sources of wave oscillations such as iceberg rotation.

We added the detected peaks of the pressure sensor data to Figure 12. Thus a comparison with the 30 TRI derived calving events becomes possible.

**5. Discussion**

1.14-15 "[...] no clear temporal pattern of tidal or diurnal recurrence [...]" comes to me as a surprise as it is not presented in the results (but should be, including Figure 9).

We deleted this part of the sentence here as it comes later in the discussion.

5.1 Relation to ice flux and other processes

1.16 Which "other processes"? Be specific. Here you want to close the "ice flux budget at the calving front" or find the "Missing volume in the deep sector"

We changed the subtitle to "Relation to ice flux".

- 1.17-23 Your simplification to compute the ice flux is fair but neglects variations in ice thickness and 5 important processes such as submarine melt. It is thus not convincing that the total calving volume matches the computed flux. Try to obtain the ice flux by integrating the glacier velocity, height for aerial calving (ice thickness would be better assuming a certain bed topography) and the glacier discrete width (for each space unit) maybe even upstream of the front assuming constant front.
- We added to Figure 7 an ice flux estimate along the front derived from the available velocity and 10 surface height data. Additionally, we produced an ice flux estimate that uses a constant front thickness of 150 m to also account for the ice flux underwater. We assume that the front thickness is similar for both sections as we cannot see changes in surface characteristics (e.g. crevasses) on the surface of the glacier.
- 15

page 14

1.1-6 I am confused here as you seem to compute the aerial ice flux using the ice height above water (150 m) and thus the missing aerial volume in the deep sector cannot be directly caused by oceanic melting, but indirect effect of the undercutting and thereby lower stress threshold for breakoff. Compare the volume you detect with the aerial ice flux in the deep sector with an ice height of 50 m. The

estimated volume from the ice flux is then threefold lower than your previous estimate and is closer to 20 your TRI calving volume estimate.

The 150 m is not used to calculate the aerial ice flux but the total ice flux (with subaguatic ice flux). So in Figure 7d we calculated now the ice flux with the available front height and additionally with an ice thickness of 150 m to represent the total ice flux.

- 25 I.3-4 Before neglecting the role of filtered calving events in explaining the missing volume, could you check that the number of filtered calving events is proportionally the same in the shallow and deep sector assuming a homogeneous noise along the front? The effect of oceanic ice melt and undercutting in the deep sector may cause smaller blocks to fall at lower stress threshold than in the shallow part. This is coherent with your observation that few large calving events occur in the deep
- part. 30

We added a discussion part about the calving of small volumes at the end of this section.

I.8-15 Good discussion and comparison. I would just add the year when the oceanic melt was obtained as it depends on warm Atlantic water intrusion that has reached a peak in 2007 and has weakened since (hence the glacier advances in the region).

We added that the summer melt rates that were measured in 2008. 35

I.15 "the contact area [...] is much smaller" by how much?

We added the water depth of the shallow sector as a reminder.

I.20 Indicate that the 75% mass removal in the deep sector occurs only over two third of the calving front, showing a greater efficiency of melting in the submarine part of the front.

We deleted this number as we do not know how much is actually explained by the subaquatic mass loss and how much with small calving events.

5 5.2 Influence from cliff height and shape

Overall, the discussion is good, but there is no discussion on the effect of undercutting on stress regime (see comments in 5.1).

We added this to the chapter before to the section with calving of small events.

I.22 Alternative title: "Effect of steeper and higher ice cliff " or "Role of front geometry on stress

10 We thank referee 2 for this suggestion. However, in this section not only the cliff height and steepness but also the front shape including the rock ridge is discussed.

**page 15**

I.1 "decreasing water level" do you mean tides or specify what causes water level to decrease in crevasses.

15 Here we mean not that the process of decreasing water level causes the crevasses further upstream. The water level in front of the glacier influences where the crevasses open. If the water level is small then the crevasses open higher upstream. We rephrased it and hope it is clearer now.

I.3-7 Nice interpretation that could also be applied to the deep sector. You can verify your hypothesis by comparing front positions and the glacier retreat seen in Figure S.1 may be significant.

20 Thank you for the suggestion. We agree that in the deep sector undercutting due to oceanic melt may lead to small calving events, which cannot be detected with the TRI. We moved this section to the chapter before and added the undercutting due to oceanic melt at the deep sector.

5.3 Calving statistics

Although the discussion is well written and nicely supported with recent studies, the low *p* values of the likelihood test on Figure 6 show me that both a power law and log-normal can explain your distribution in both sectors. Thus reformulate your question here as the power law distribution can also be attributed to the shallow sector.

Due to the new filtering condition the statistics changed and we updated this section.

I.8 Alternative title: "Self-organised critical system vs less complex systems" or "calving distribution anddriving mechanisms" or "calving distribution"

We changed the title to "Calving event size distribution".

I.12-13 All cited studies estimate aerial calving and neglects subaqueous events. The main reason for a too steep power law curve may be that the study period is too short and there were too many (or not enough) calving events larger than 105 m3 (or between 104 and 105 m3). This would also explain the miefit on Figure 6c.

35 the misfit on Figure 6c.

We thank referee 2 for this hint but we had to change this paragraph due to the new results.

I.16 "[...] instability with many small events [...] events of greater magnitude"

We had to delete this sentence in the revised manuscript.

I.17-18 I do not understand the point of that sentence. Rephrase or delete.

5 We deleted this sentence.

5.4 Comparison with pressure sensor data

Discuss whether you can identify waves of rolling iceberg from those of calving events. Subaqueous events can often be confused with icebergs rolling close to the glacier front as we can often only identify them from the produced waves.

10 We added a sentence about the challenges of analysis wave data at the end of the section. A more detailed analysis of the pressure sensor data will be done in a follow up study.

I.25 Alternative title: "Comparison with calving wave signal"

page 16

5.5 Relation to external forcing

- 15 The air humidity is not present here (section 2.3) and would be useful to assess potential precipitation periods and atmospheric disturbance on the radar signal. Precipitation tends to mess up surface melt and tidal signals seen in calving event occurrence. The temporal resolution used in Figure 9 for your calving events may be too high to find a relation that often occurs on hourly time scale. On Figure 9d-e, resample your calving data and apply a cutoff for the two-three largest events around 0.2 in 9d and 10 in 9a. The leagend can be pletted once in between the two papels.
- in 9e. The legend can be plotted once in between the two panels.

We could not find a relation with the air humidity. Also Eqip Sermia is a very dry place with not a lot of precipitation. Thus we did not add the air humidity. We could not find a different pattern by using hourly timescale. We kept the higher temporal resolution as we would lose information by lowering the resolution. We added a cut-off to the high values and plotted the legend only once.

25 I.12 Alternative title "Absence of meteorological and tide effect"

We thank referee 2 for this suggestion. However, we think that we have not enough data to ensure that the meteorological and tide effect is absent.

I.14-15 The sentence on surface melt is not coherent with the sentence before that finishes on tide effects.

30 We changed that part and hope it is now clearer.

page 17

6. Conclusion

Update the conclusion after answering my comments.

I.10 "We developed a novel detection method based on TRI DEM differencing to establish [...]"

We changed that sentence in the revised manuscript.

**---- Figures ----**

5

Figure 3

Remove the velocity comparison as a more thorough comparison is achieved in Rohner et al, 2019 (refer to their Figures). Instead show here a plot of the mean velocity for the entire period and beside or in another figure plot a comparison of the TRI DEM and Arctic DEM including a distribution of their

10 difference (or do you have the UAV derived DEM as well?). Eventually, in a new figure plot the DEM before and after one selected calving event, the subtraction of the DEMs, mask of the glacier front and the watershed result. Condense the caption: "Velocity field at the glacier front measured a) with the TRI on 19 August 2016 and b) with a UAV"

Fig 5 in revised version: The velocity of the UAV is not meant as a comparison but to get an idea of the
 velocity field over a larger area (e.g. for ice fluxe estimates) as the TRI velocity field is only available at
 the front. We added a plot of a TRI derived DEM, a comparison between TRI DEM and Arctic DEM
 (supplement), a differentiated DEM (unstacked and stacked) and the watershed result. We condensed
 the caption.

**Figure 4**

20 Plot the rough location and view angle of the picture

Fig 6 in revised version: We plotted the location and view angle of the picture roughly.

Figure 5

**Panel a**

Could you use an image taken from the TRI position or the radar image in Figure 4 cropped and rotated to fit the format here and with the line you use to stack your data in c)? The current image is confusing as it is in a different orientation than the TRI (taken from a hike to front, I guess) and saturated (I cannot see the front texture).

Fig 7 in revised version: We exchanged the image with an image taken close to the radar position but further away.

30 Panel b-d

extend the plot to the end of the right plot margin. If this was for the legend in b, change min max lines to a polygon (shadow) and place the legend horizontal at the bottom of the panel such as --- Elevation ---- Mean velocity |grey box| Min/max velocity. Indicate the location of the sectors above panel b) and delete them form panel c).

35 Fig 7 in revised version: The plot does not go to the end of the margin to make it comparable with the image in panel b which is slightly wider.

**Panel c**

Please use a linear colour scale. Your current palette highlights mostly areas with no calving volume change and the yellow parts of your spectrum (7000 m3) and a bit the red ones. Choose a linear colour scale from light yellow to red or blue. Write "data gap" between 22 and 23 Aug.

5 Fig 7 in revised version: We changed the colours here, no calving is now grey which makes the calving events better visible. We wrote data gap in the caption.

**Panel d**

Delete the red dashed line representing 50% and 100% of the mean calving volume in the shallow sectors. I don't understand which message it is supposed to convey. Do you mean that 50% is equal to 0.4.106 m2 of calving volume?

10 ~0.4 106 m3 of calving volume?

Fig 7 in revised version: We deleted this and added the ice flux to this graph.

**Figure 9**

Stretch the vertical axis for all panels to highlight the variations of your parameters. Shift the shortwave data so that we see that the curve goes to zero during the night (there must still be some light at this

15 latitude mid-august?). Use the same temporal resolution for the Volume and number of events than the two first panel for instance hourly. Cutoff the extreme value on the 26 Aug. evening and write its value on the figure with an arrow pointing to the top. (also see comments *page 16*)

Fig 12 in revised version: We changed the y-axis to make the variations better visible. The shortwave data is now shifted. We cut-off the extreme values and added arrows pointing to the top. We did not
change the temporal resolution as we do not see a different result with a lower resolution and we would lose information in doing so.

**Calving event size measurements and statistics of Eqip Sermia, Greenland, from terrestrial radar interferometry**

Andrea Walter1,2, Martin P. Lüthi1, Andreas Vieli1

1Institute of Geography, University of Zurich, Zurich, Switzerland

5 2Laboratory of Hydraulics, Hydrology and Glaciology, ETH Zurich, Zurich, Switzerland

Correspondence to: Andrea Walter (andrea.walter@geo.uzh.ch)

Abstract. Calving is a crucial process for the recently observed dynamic mass loss changes of the Greenland ice sheet. Despite its importance for global sea level change, major limitations in understanding the process of calving process-remain. This study presents high resolution calving event data and statistics recorded with a terrestrial radar interferometer at the front of Eqip Sermia, a marine terminating outlet glacier in Greenland. The derived digital elevation models with a spatial resolution of

- 5 several meters recorded at one-minute intervals were was processed to provide source areas and volumes of 9061700 individual calving events during a 6 day period. The calving front can be divided into sectors ending in shallow and deep water with different calving statistics and styles. For the shallow sector, characterised by an inclined and very high front, calving events are more frequent and larger than for the vertical ice cliff of the deep sector. We suggest that the calving volume deficiency of 90% missing in our observations of the deep sector is removed by oceanic melt, -and-subaquatic calving and small aerial
- 10 calving events., Assuming a similar ice thickness for both sectors which implies that subaqueous mass loss must be substantial for this sector with a contribution of up to 675 % to the frontal mass loss. The size distribution of the shallowdeep sector is represented by a log-normal model, follows a power law, while for the deep shallow sector the log-normal and power-law model fit well, but none of them is significantly better, is likely represented by a log normal model. Variations in calving activity and style between within 
[revised manuscript text omitted]

---

## Referee Report (RR1)

The revision has significantly improved the quality of the paper, thank you the authors!

Here are a few fairly minor comments that might help clarify the paper:

1) I am glad to see that the authors have now detailed the TRI-DEM uncertainty assessment in section 4.1. Figures S2-S4 show mean height difference over the stable terrain. I am not surprised to see the relatively small variabilities, because these values are the averages of many stationary points. I think a figure similar to Figure S7, but showing the difference between two TRI derived DEMs over the stable terrain area would be more informative.

2) Figure 4. For the purpose of this study, we are really interested in areas with elevation difference larger than 5 m. But the color scale makes it difficult to interpret the spatial variability. The authors may consider using a different color scheme, or reduce the ~-75— 75 m elevation range.

3) Line 15, page 7: What does "the signal to noise ratio" refer to? I though the signal-to-noise ratio on stable terrain would be 0. How could it be higher than for the calving events?

4) References: Please consider using the final version of Xie et al. (2018) published in TC, not the discussion version published in TCD.

Sincerely,

—Surui Xie

---

## Referee Report (RR2)

I would like to thank the authors for undertaking significant revisions of the manuscript. The paper has been re-written in large parts following my comments and is now supported by five edited figures and three new ones. The corrections have strengthened its message and improved its clarity, especially since a reorganization of the text introduces now, in the order of importance, the DEM methods/results/figures before the velocity ones.

I use the structure of my earlier major concerns and an assessment of the corrections to recommend this paper for publication. Its conclusions highlight the dominant role of oceanic melting on the glacier frontal ablation as the ice is in greater contact with ocean and on the type, size and frequency of calving events. A major result of this high resolution campaign (in space and time) is that calving is less important than earlier thought for the frontal ablation budget and it is in lines with recent literature.

1. DEM derivation of the glacier front from TRI

The reliability of the measurements and the improvements due to DEM stacking are now extended in both the methods and results. The authors have re-processed their entire datasets after applying a correction factor to improve systematic error in the GPRI positioning and a shape conditions to account for pixel anisotropy (in radar space) while filtering small calving events. These improvements lead to new estimates of calving events and volumes (mostly filtering out small, Table 1) and thus new results for the distribution fits.

The new error assessment in space and time reinforces my confidence in their results as the signal to noise ratio is high, constant over their field period and a bias towards the shallow sector is clearly stated. Moreover, the threshold to detect calving events is higher than the noise (5 m) and coherent across the old and new manuscript. This is now clearly supported and illustrated for the stacked absolute elevations in the novel Figure 3 and the noise reduction of the difference through stacking in the added Figure 4. Figure 4 is particularly stunning to show the spatial distribution of noise and a clear example of the workflow used here.

2. Issues in determining best fit models for calving distribution

The new manuscript explains now in clearer details the interpretation of the maximum likelihood assessment (the ratio R values) in the results and the difficulty to prove the dominance of a particular fit. The authors extended the discussion as well. They state that a longer observation period is necessary to demonstrate the transition from a power law (self-organised critical system) to a log-normal distribution (complex system). Since the reprocessing of the calving event dataset, the retrieved power law exponent is also closer to published estimates. These corrections provide a more balanced and interesting scientific discussion answering my earlier concern.

3. Ice flux budget: bed topography and missing component

The added mention of the BedMachine data to estimate the bed topography, the inclusion of the ice flux comparison in Figure 7 and simplification of the text in the discussion lifted my doubts regarding your simplified flux estimates. The change from absolute ice flux into percentage makes your work easier to read, more relevant to other scientists and comparable to other glaciers. In section 5.1, the new paragraphs on the missing volume from small calving events in the total ice flux brings a more general view on the processes occurring at the front and the limits of our remote sensing instruments as I recommended earlier.

4. Better integration of calving wave dataset

A peak detection is now applied to the wave dataset providing a first order estimate of calving events although, as the authors added, some other wave sources may contaminate their estimate such as iceberg rolling. The integration of this dataset is also better through its use in Figure 12 that tries to link environmental forcing to calving variations.

Minor comments:
Abstract
l. 8, Precise what is the "deficiency" relative to

p.15
l. 6: replace "ration" by "ratio"
l.9: "better fit of the log-normal" compared to what exponential or power law.

Figure 8:
Can you place back on the figure the maximum likelihood scores as in your earlier manuscript? Those scores were very clear and useful for the reader to interpret your results.

---

## Author Response (AR2)

**Calving event size measurements and statistics of Eqip Sermia, Greenland, from terrestrial radar interferometry**

Andrea Walter, Martin P. Lüthi, Andreas Vieli

Submitted for review to The Cryosphere in May 2019

**Reply to reviewer's comments**

**General reply**

We thank both reviewers, Surui Xie and Pierre-Marie Lefeuvre, for their invested time for reading through the manuscript and their suggestions. We addressed their comments in our revisions as described in the corresponding answers.

**Reply to concerns Referee 1**

1) I am glad to see that the authors have now detailed the TRI-DEM uncertainty assessment in section 4.1. Figures S2-S4 show mean height difference over the stable terrain. I am not surprised to see the relatively small variabilities, because these values are the averages of many stationary points. I think a figure similar to Figure S7, but showing the difference between two TRI derived DEMs over the stable terrain area would be more informative.

We thank reviewer 1 for this suggestion. However, we think this information is already presented in Figure 4. We additionally marked in Figure 4 the stable terrain used for the uncertainty analysis.

2) Figure 4. For the purpose of this study, we are really interested in areas with elevation difference larger than 5 m. But the color scale makes it difficult to interpret the spatial variability. The authors may consider using a different color scheme, or reduce the ~-75— 75 m elevation range.

We agree that a reduction of the elevation range helps to better see the spatial variability. Thus in the revised manuscript the elevation range is reduced.

3) Line 15, page 7: What does "the signal to noise ratio" refer to? I though the signal-to-noise ratio on stable terrain would be 0. How could it be higher than for the calving events?

A high signal to noise ratio means that the signal is stronger than the noise, which is also the case for the stable terrain. As it is stable the signal is no movement. We rewrote this sentence to: When applying these filtering thresholds, the signal-to-noise ratio is higher on stable terrain than without filtering.

4) References: Please consider using the final version of Xie et al. (2018) published in TC, not the discussion version published in TCD.

We apologize for this and changed it in the revised manuscript.

5  **Reply to concerns Referee 2**

Abstract, l. 8: Precise what is the "deficiency" relative to

We changed this sentence so that it should be clearer, to: We suggest that the calving volume deficiency of 90%
10  relative to the estimated ice flux in our observations of the deep sector is removed by oceanic melt, subaquatic calving and small aerial calving events.

p.15, l. 6: replace "ration" by "ratio"

15  We changed that in the revised manuscript.

p.15, l.9: "better fit of the log-normal" compared to what exponential or power law.

We clarified this in the revised manuscript.
20
Figure 8: Can you place back on the figure the maximum likelihood scores as in your earlier manuscript? Those scores were very clear and useful for the reader to interpret your results.

We thank reviewer 2 for this suggestion and placed back the scores on the figure.

[revised manuscript text omitted]

---

## Author Response (AR3)

**Calving event size measurements and statistics of Eqip Sermia, Greenland, from terrestrial radar interferometry**

Andrea Walter, Martin P. Lüthi, Andreas Vieli

Submitted for review to The Cryosphere in May 2019

**Reply to editors' comments**

**General reply**

We thank the editor Evgeny Podolskiy for the suggested corrections and the invested time for reading through the manuscript. We addressed all the technical corrections and added the thresholds used in the peak detection algorithm to the methods section.

[revised manuscript text omitted]